# Instance-Optimal Private Density Estimation in the Wasserstein Distance

**Vitaly Feldman**
Apple

**Audra McMillan**
Apple

**Satchit Sivakumar**[*]
Boston University

**Kunal Talwar**
Apple

## Abstract

Estimating the density of a distribution from samples is a fundamental problem in statistics. In many practical settings, the Wasserstein distance is an appropriate error metric for density estimation. For example, when estimating population densities in a geographic region, a small Wasserstein distance means that the estimate is able to capture roughly where the population mass is. In this work we study differentially private density estimation in the Wasserstein distance. We design and analyze instance-optimal algorithms for this problem that can adapt to easy instances.

For distributions $P$ over $\mathbb{R}$, we consider a strong notion of instance-optimality: an algorithm that uniformly achieves the instance-optimal estimation rate is competitive with an algorithm that is told that the distribution is either $P$ or $Q_P$ for some distribution $Q_P$ whose probability density function (pdf) is within a factor of 2 of the pdf of $P$. For distributions over $\mathbb{R}^2$, we use a different notion of instance optimality. We say that an algorithm is instance-optimal if it is competitive with an algorithm that is given a constant-factor multiplicative approximation of the density of the distribution. We characterize the instance-optimal estimation rates in both these settings and show that they are uniformly achievable (up to polylogarithmic factors). Our approach for $\mathbb{R}^2$ extends to arbitrary metric spaces as it goes via hierarchically separated trees. As a special case our results lead to instance-optimal private learning in TV distance for discrete distributions.

## 1 Introduction

Distribution estimation is a fundamental problem in statistics. In this work, we focus on the problem of learning the density of a distribution over a low-dimensional real space. Our motivation for studying this problem comes from practical problems such as estimating the population density in a geographical area (defined by bounded two dimensional space, for e.g. $[0, \ell]^2$), learning the distribution of accuracy of a machine learning model (i.e. a distribution over $[0, 1]$), estimating the average temperature across latitude, longitude, and altitude (i.e. a distribution over $[0, \ell]^3$) etc.

In this work, we are interested in the *non-parametric* version of this question, where we make no assumptions on the form of the distribution we are learning. This is frequently of interest in practice, where population densities for example may change over time (become more or less concentrated), and it is difficult to specify a meaningful parametric class that will simultaneously capture all densities of interest. Given estimation is often done using sensitive data (for e.g. health data), our interest in this question is in, and consequently all our results are for, the differentially private version of this question. While we believe our results in the non-private setting are also novel and interesting, we view the private results as our main contribution.

Any statistical algorithm learning from samples is inexact. The appropriate gauge to measure the (in)accuracy of a density estimation algorithm depends on how this density estimate is used. In this work, we focus on the *Wasserstein* distance between the original distribution and the learnt

38th Conference on Neural Information Processing Systems (NeurIPS 2024).

distribution as our measure of accuracy. Known by many names (Earthmover distance, Kantorovich distance, Optimal Transport distance), this distance is defined over any distance metric $d$ as the minimum over all couplings $\pi$ from $P$ to $Q$ of the quantity $\mathbb{E}_{x \sim P}[d(x, \pi(x))]$. It is arguably one of the most natural ways to define distances between distributions over a metric space and has been extensively studied (see Appendix C) . We note that Wasserstein distance is particularly salient in many practical applications of density estimation where the geometry of the space is significant. As a simple example, when creating population density estimates, if the population is concentrated in a few cities, then outputting a distribution concentrated close to these cities (even if not exactly at the cities) is intuitively better than outputting a distribution that is more spread out. Metrics such as TV distance that do not incorporate the geometry of the space do not capture this nuance. Additionally, Wasserstein distance is versatile and can be adapted to the setting of interest by varying the metric. In the case of the metric being a discrete metric with $d(x, y) = \mathbf{1}(x \neq y)$, it reduces to the commonly used total variation (TV) distance. Our focus in this work is on the case of Euclidean distance metric on $[0, 1]$ or $[0, 1]^2$, though our results apply to both to higher-dimensional Euclidean space as well as to any finite metric. In the $[0, 1]$ case (with the standard Euclidean metric), the Wasserstein distance is equivalent to the total area between the cumulative distribution functions.

The problem of learning a distribution under Wasserstein distance has a long history, starting with [Dud69] proving worst-case bounds on the rate of convergence of the Wasserstein distance between the empirical distribution $\hat{P}_n$ and the target distribution $P$ over $\mathbb{R}^d$. Similarly, this question for the case of the discrete metric ($d(x, y) = \mathbf{1}(x \neq y)$) has been very well studied. However, most known results for this problem look at it from the point of view of worst-case analysis. This can paint a rather pessimistic picture. For example, the minimax rate of $\varepsilon$-privately learning a discrete distribution over $\{0, \ldots, k\}$ in TV distance (i.e. Wasserstein with the discrete metric described above) scales linearly with $k$, which can be prohibitive for large support size $k$. For Wasserstein distance with $\ell_2$ norm, the rate of convergence of the empirical distribution suffers a curse of dimensionality, with the worst-case error between the distribution and the empirical distribution being $\Theta(n^{-\frac{1}{d}})$ for distributions over $[0, 1]^d$. For the differentially private version of this question, recent works [BSV22, HVZ23] have shown that the optimal Wasserstein minimax error between the sample and the private estimate is $\tilde{\Theta}((\varepsilon n)^{-\frac{1}{d}})$. This worst-case analysis viewpoint fails to distinguish between algorithms that perform very differently on the types of instances one may see in practice. In particular, many practical distributions may be more feasible to estimate than suggested by the minimax rate. As an example, Figure 1 shows the cumulative distribution function of a bimodal distribution on $[0, 1]$ with very sparse support, and the cdf learnt by a minimax optimal algorithm, as well as an algorithm we present in this work (See Appendix F for details on this experiment). As is clear from the figure, the minimax optimal algorithm is easily outperformed. This phenomenon only gets worse in higher dimensions. Similarly, if the distribution in $\mathrm{Re}^d$ lies on a $k$-dimensional subspace, the worst-case error scaling with $\tilde{O}((\varepsilon n)^{-\frac{1}{d}})$ is significantly larger than our algorithm's scaling of $\tilde{O}((\varepsilon n)^{-\frac{1}{k}})$.

This motivates the problem of viewing this question through the lens of *instance optimality*. [2] Briefly, instance optimal algorithms are those that on any given instance of the problem, are able to perform competitively with what any algorithm can do on this instance. Let $\mathcal{M}$ be a class of algorithms of interest (e.g. all $(\varepsilon, \delta)$-differentially private algorithms) and $cost(\cdot, P)$ be a cost measure for an instance $P$. In our setting, we have a distribution $P$ over a metric space, and given a set $\hat{P}_n$ of $n$ samples from $P$, we want to learn an estimate $\mathcal{A}(\hat{P}_n)$ for the distribution. Our measure of performance is the Wasserstein distance $\mathcal{W}$, so $cost(\mathcal{A}, P) = \mathbb{E}[\mathcal{W}(P, \mathcal{A}(\hat{P}_n))]$. We would ideally like to say that an algorithm $\mathcal{A}$ is $\alpha$-instance optimal in a class $\mathcal{M}$ if for all instances $P$, and all $\mathcal{A}' \in M$,

$$cost(\mathcal{A}(\hat{P}_n)), P) \leq \alpha \cdot cost(\mathcal{A}'(\hat{P}_n)), P). \qquad \text{(InstanceOptimality-Ideal)}$$

The reader would have noticed that this definition is however impossible to achieve except for trivial classes $\mathcal{M}$. The algorithm $\mathcal{A}'$ that ignores its input and always outputs $P$ makes the right hand side 0. However, this algorithm performs poorly on any distributions far from $P$ and so is not a reasonable benchmark. A common approach in many works is to measure the performance of the competing algorithm $\mathcal{A}'$ not just on the given instance, but on a small neighborhood around it. Thus we say

---

[2]c.f. related work section for discussion of other beyond worst case analysis approaches for this question and Appendix B for a more in-depth discussion of our approach.

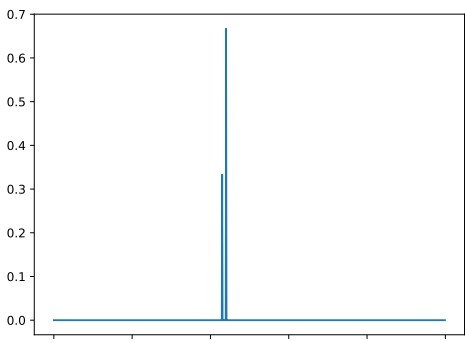
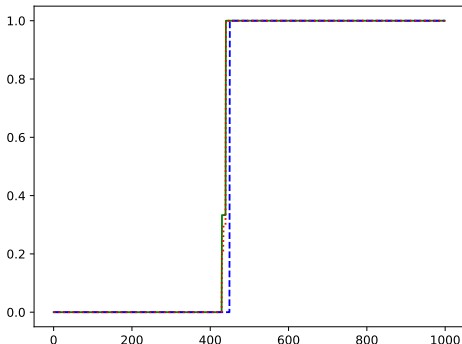

Figure 1: (Left) A sparsely supported distribution on integers [0,999] (pdf). (Right) CDF for the same distribution (green, solid line), along with a (non-private) minimax optimal learnt distribution (blue, dashed line), as well as 1-DP instance-optimal algorithm (red, dotted), both learnt from the same 1600 samples. The $W_1$ error for the minimax optimal algorithm is 13.4, whereas the DP estimated distribution has $W_1$ error of 0.86. While this example is artificial, it demonstrates the large potential gap between minimax optimal and instance optimal algorithms on specific instances.

that that an algorithm $\mathcal{A}$ is $\alpha$-instance optimal amongst a class $\mathcal{M}$ with respect to a neighborhood function $\mathcal{N}$ if for all instances $P$, and all $\mathcal{A}' \in M$

$$cost(\mathcal{A}(\hat{P}_n)), P) \leq \alpha \cdot \sup_{P' \in \mathcal{N}(P)} cost(\mathcal{A}'(\hat{P'}_n)), P').$$

In other words, the benchmark we evaluate against is the cost of the best algorithm for a neighborhood $\mathcal{N}(P)$ *that knows this neighborhood*. We would like our algorithm $\mathcal{A}$, that is not tailor-made for $\mathcal{N}(P)$, to nevertheless be competitive against this benchmark.

This definition is general, and captures most notions of instance optimality that have been studied in the literature. The set $\mathcal{N}(P)$ must be carefully defined for this notion to be meaningful; we can always define $\mathcal{N}(P)$ to be the set of all instances whence this notion reduces to worst-case analysis. In many previous works, this neighborhood map has been defined to capture the belief that any natural algorithm must not have significantly different performance on different members of $\mathcal{N}(P)$. For example, [FLN01, ABC17, VV16, OS15, GKN20] include in $\mathcal{N}(P)$ appropriate renamings of $P$ to capture some kind of permutation invariance of natural algorithms. In statistics, one often enforces that the cardinality of $\mathcal{N}(P)$ is 2, often called the *hardest one-dimensional subproblem* [CL15, AD20, DLSV23]. Some recent works in privacy [HLY21, DKSS23] have defined instance optimality w.r.t. neighboring datasets obtained by deleting a small number of data points. Any reasonable definition of instance optimality for a problem must justify its choice of the neighborhood map; similar choices must be justifiable in every other notion of beyond worst case analysis [Rou21]. In instance-optimality definitions, this choice of neighborhood is what encapsulates what class of domain-specific algorithms our algorithm competes against. A good definition thus depends on the context and on the kind of domain knowledge we imagine an expert designing a custom algorithm for an application may have. Ideally, the definition is broad (i.e. the neighborhoods $\mathcal{N}(P)$ are sufficiently contained) so that in a large class of applications, we expect the domain knowledge to not be enough to rule out any member of $\mathcal{N}(P)$. We discuss this general definition of instance optimality further in Appendix B. We remark that for reasonable neighborhood maps, this is an extremely strong requirement: an instance-optimal algorithm must simultaneously do well on every single input, in fact as well as any other algorithm that is given this neighborhood $\mathcal{N}(P)$ in advance!

Instance optimality guarantees are most useful when there is a big difference between achievable utility guarantees for typical cases and the worst-case utility guarantees. Wasserstein estimation is an example of such a problem. We will see that achievable utility bounds for, for example, concentrated distributions are a lot better than worse case distributions. Our definition of instance optimality is particularly suitable for metric spaces, and our notion of neighborhood allows the target utility bound to adapt to the distribution. We note that for estimation in Wasserstein distance with practically important metrics such as $\ell_1$ and $\ell_2$ norms, it is unclear if existing instance optimality

definitions (using notions of neighborhood discussed above) capture this. For example, for discrete distributions, setting the neighborhood to be all permutations of the distribution destroys all structure of the distribution (for e.g. concentration), and hence performance on this neighborhood may not capture the relative ease of estimation of a concentrated distribution. Similar problems apply to other previously studied definitions of instance optimality, which are not well-suited to density estimation with error metrics that incorporate the geometry of the metric space. See Appendix B and Appendix C for further discussion on the inadequacy of existing instance optimality definitions for our setting of interest.

Our notion of neighborhood will correspond to small balls in one of the strictest notions of distance between distributions. Recall that for distributions $P, Q$ on $X$, $D_\infty(P, Q)$ is defined as $\sup_{x \in X} \max \left( \ln \frac{P(x)}{Q(x)}, \ln \frac{Q(x)}{P(x)} \right)$. Our neighborhood map $\mathcal{N}$ will have the property that for all $P$, and for all $Q \in \mathcal{N}(P)$, $D_\infty(P, Q) \leq \ln 2$. This corresponds to the benchmark algorithm $\mathcal{A}'$ being given as auxiliary input a multiplicative constant factor approximation to the probability density function $P(x)$ (and we can replace the constant 2 by any constant). In particular, an algorithm that knows the support of the distribution $P$ will not be able to do much better than our algorithm that gets no such information. Notice that this implicitly implies that our algorithm is able to exploit sparsity in the data distribution since it is competitive with an algorithm that is told the support. In the one-dimensional real case we can achieve an even stronger notion of instance-optimality. In this case $\mathcal{N}(P)$ is defined to be $\{P, Q\}$ where $Q$ is a distribution with $D_\infty(P, Q) \leq \ln 2$. This is a strengthening of the rate defined by the *hardest one-dimensional subproblem*.

We also give a definition that captures another aspect of instance optimality, related to the notion of super efficiency, that we term local minimality in Appendix B. Informally, local minimality says that if any comparator algorithm does better than $\mathcal{A}$ on $P$, then there is a distribution $Q$ in the neighborhood of $P$ where $\mathcal{A}$ does better than the comparator. Approximate local minimality relaxes the latter condition to being better than some constant times the comparator. The two definitions of approximate local minimality and instance optimality are in general incomparable (see Appendix B) but for suitable smooth algorithms, we show that these definitions are equivalent. Our algorithms, both for the 1-dimensional and the case of general metric spaces approximately satisfy both these definitions.

In order to show that the instance optimality definition is achievable, we give both algorithmic upper bounds and matching, up to logarithmic factors, theoretical lower bounds. The algorithms we use in our upper bounds are built largely from ingredients previously used for similar problems. We see this as an asset since these algorithms are implementable in practice. A key ingredient that we do introduce is the use of randomised HST approximation of finite metric spaces. This replaces deterministic hierarchical decompositions that were used in prior work, allowing us to gain tighter utility guarantees. Our main conceptual contribution is to introduce what we believe to the right notion of instance optimality for this problem, including the definition of a meaningful neighbourhood function. The main technical challenge is in the lower bounds, which require carefully building nets of distributions within each neighbourhood $\mathcal{N}(P)$ that allow us to use a slight generalisation of DP Assoud's Lemma to give a lower bound on the target estimation rate for each distribution $P$.

**Preliminaries:** First, we define differential privacy. Further discussion on differential privacy can be found in Appendix A.

**Definition 1.1** (Differential Privacy [DMNS17, DKM$^+$06])**.** *A randomized algorithm $\mathcal{A} : \mathcal{X}^n \to \mathcal{Y}$ is $(\varepsilon, \delta)$-differentially private if for every pair of datasets $\mathbf{x}, \mathbf{x}' \in \mathcal{X}^n$ that differ in at-most one data entry, and for all events $Y \subseteq \mathcal{Y}$,*

$$\Pr[\mathcal{A}(\mathbf{x}) \in Y] \leq e^\varepsilon \cdot \Pr[\mathcal{A}(\mathbf{x}') \in Y] + \delta.$$

Given an estimation algorithm $\mathcal{A} : \mathcal{X}^n \to \mathcal{M}$, the estimation rate of $\mathcal{A}$ for distribution $P$ is:

$$R_{\mathcal{A},n}(P) = \inf_{t \in \mathbb{R}} \{t : \text{w.p.} \geq 0.75 \text{ over } \mathbf{x} \sim P^n \text{ and the randomness of the algorithm, } \mathcal{W}_d(\mathcal{A}(\mathbf{x}), P) \leq t\}. \tag{1}$$

## 1.1 Our Results

We start by stating an informal version of our result in the one-dimensional real case.

**Theorem 1.2** (Informal 1-dimensional result). *Let $\varepsilon, \gamma \in (0,1]$. There is an $\varepsilon$-differentially private algorithm $\mathcal{A}$ such that, for all distributions $P$ supported in $[0,1]$, for all natural numbers $n > \frac{\text{polylog } 1/\gamma}{\varepsilon}$, there exists a distribution $Q$ (with $D_\infty(P, Q) \leq \ln 2$) such that the following is satisfied.*

*For any $\varepsilon$-DP algorithm $\mathcal{A}'$, with probability at least $0.75$ over the randomness of $\mathbf{x} \sim P^n$ and additional randomness of the algorithm,*

$$\mathcal{W}(\mathcal{A}(\hat{P}_n), P) \leq \text{polylog } n \cdot \sup_{P' \in \{P, Q\}} R_{\mathcal{A}', n'}(P') + \gamma$$

*where $n' \approx \frac{n}{\text{polylog } n/\gamma}$*

In this one-dimensional case, our algorithm is based on DP quantile estimation. The additive $\gamma$ term can be made polynomially small. The lower bound is based on (differentially private) simple hypothesis testing where for each distribution $P$, we find a distribution in $\mathcal{N}(P)$ that is indistinguishable from $P$ given $n$ samples but also sufficiently far from $P$ in Wasserstein distance.

Extending the quantiles based approach from the one dimensional setting to even the two dimensional setting is challenging, as there is no "right" way to generalize quantiles to dimensions 2 or beyond. Several previous works on Wasserstein density estimation (e.g. [BNNR09]) have used a hierarchical decomposition approach to address this question. A hierarchical approach has also been used in various more practical works on private density estimation (e.g. [CB22, QYL12, BKM$^+$21, MJT$^+$22, ZXX16]). These works focus on practical performance and do not offer tight theoretical bounds. A hierarchical approach was also used by [GHK$^+$23], who proved theoretical bounds for a related problem, but not through the lens of instance optimality. We compare our results to theirs in more detail later in this section.

The use of deterministic hierarchical decompositions in all these papers means that some points that are very close (but on opposite sides of the boundaries of the hierarchical decomposition) get mapped to relatively far points, resulting in high distortion factors that are not appropriate for instance optimality.

Inspired by the above approaches but noting their constraints, we use a randomized embedding into hierarchically separated trees instead of a deterministic one. We define our algorithm on any hierarchically separated tree metric and use the fact that there is a randomized embedding of $[0,1]^2$ on a hierarchically separated tree metric space with low distortion. This, along with some other important technical modifications (such as truncating low values to $0$), allows us to analyze a variant of the above practical algorithms theoretically and show that it satisfies our strong notion of instance optimality, up to polylogarithmic factors in the number of samples.

**Theorem 1.3** (Informal two-dimensional result). *There is a polynomial time $\varepsilon$-differentially private algorithm $\mathcal{A}$ that for any distribution $P$ on $[0,1]^2$, any integer $n$, and any $\varepsilon$-DP algorithm $\mathcal{A}'$ with probability at least $0.75$, satisfies*

$$\mathcal{W}_2(\mathcal{A}(\hat{P}_n), P) \leq (\log n)^{O(1)} \sup_{P': D_\infty(P, P') \leq \ln 2} \mathbb{E}[\mathcal{W}_2(\mathcal{A}'(\hat{P}'_{n'}), P')],$$

*where $n' \approx \frac{n}{\text{polylog } n}$. Here, the expectation is taken over the internal coin tosses of $\mathcal{A}$ as well as over the choice of the i.i.d. samples $\hat{P}_n$.*

In fact, since our algorithm is defined on any hierarchically separated tree metric space, it has the added bonus of giving instance optimality results for any finite metric space (since powerful results [Bar96, FRT03] show that any finite metric space can be embedded in a hierarchically separated tree metric space with a distortion factor at most logarithmic in the size of the metric space).

**Theorem 1.4** (Informal finite metric result). *Let $(\mathcal{X}, d)$ be an arbitrary metric space with diameter $1$. There is a polynomial time $\varepsilon$-differentially private algorithm $\mathcal{A}$ such that for any distribution $P$ on $\mathcal{X}$ any integer $n$ and any $\varepsilon$-DP algorithm $\mathcal{A}'$ with probability at least $0.75$, satisfies*

$$\mathcal{W}(\mathcal{A}(\hat{P}_n), P) \leq (\log |\mathcal{X}| \cdot \log n)^{O(1)} \sup_{P': D_\infty(P, P') \leq \ln 2} \mathbb{E}[\mathcal{W}(\mathcal{A}'(\hat{P}'_{n'}), P')],$$

*where $n' \approx \frac{n}{\text{polylog } n}$. Here, the expectation is taken over the internal coin tosses of $\mathcal{A}$ as well as over the choice of the i.i.d. samples $\hat{P}_n$.*

Our lower bound result is actually slightly stronger than stated in Theorem 1.4 since it holds not only for $\varepsilon$-DP, but also for $(\varepsilon, \delta)$-DP. At this point, we also compare specifically to the paper of [GHK$^+$23] who give an algorithm for obtaining two-dimensional heatmaps and analyze it theoretically. They focus on the empirical version of a variant of this problem as opposed to the population version, and aim to compete with the best $k$-sparse distribution. Their algorithm takes the sparsity parameter $k$ as input in order to set parameters and achieves additive error $\sqrt{k}/n$ (and a constant multiplicative factor). On the other hand, our algorithm also performs better for sparse distributions but is *automatically adaptive* to the sparsity (and hence doesn't need to take it as an input). Additionally the additive term in our work can be made polynomially small (for any polynomial) in $n$ at a logarithmic cost to the multiplicative error (regardless of the sparsity of the distribution). On the other hand, for large $k$ their results have additive error that scales with $1/\sqrt{n}$. Their use of a deterministic hierarchical decomposition makes their algorithm unsuitable for our notion of instance optimality (as discussed earlier), and it is unclear if their algorithm can be directly extended to all finite metric spaces.

Note that instance optimality for all finite metric spaces implies instance optimality results for a wide variety of applications not addressed in prior work. For example, our results immediately extend to other low-dimensional real spaces with arbitrary metrics (for e.g. $\ell_p$ norms). They also give non-trivial improvements on worst-case analysis for higher-dimensional spaces that are not the main focus of our work (for $[0, 1]^d$, we can use a fine grid of size $(1/(\eta/\sqrt{d}))^d$ at an additive cost of $\eta$ in the Wasserstein distance in order to create a finite metric space to apply our result on. Since the dependence on $|\mathcal{X}|$ in the result above is logarithmic, this translates to a $d \log \frac{d}{\eta}$ multiplicative overhead term replacing the $\log |\mathcal{X}|$ factor above. While this is still a significant overhead, all previous results on density estimation in the Wasserstein distance (in both the private and non-private literature) are worst case, where the sample complexity is exponential in $d$. Since our results only have a polynomial dependence in $d$ over the optimal error, this is a non-trivial improvement over worst-case error, even when $d$ is large.

Another immediate application of our results is to give (to the best of our knowledge) new bounds for private estimation of discrete distributions in TV distance. Generally, for learning a discrete distribution defined by probabilities $\{p_1, \ldots, p_k\}$, our results lead to a rate (up to polylogarithmic factors) of $\sum_i \min \left\{ p_i(1 - p_i), \sqrt{\frac{p_i(1-p_i)}{n}} \right\} + \sum_i \min \left( p_i, (1 - p_i), \frac{1}{\varepsilon n} \right)$.

This can give significant improvements over the worst case bounds for practically important distributions. The minimax rate is linear in the support size $k$, namely $\Theta(k/\varepsilon n)$ (for sufficiently small $\varepsilon$). Now, consider the following power-law distribution over support size $k$: $p(i) \propto i^{-2}$. (Power law distributions arise frequently in practice for e.g. frequencies of family names, sizes of power outages etc. all follow power law distributions.) Applying our result above gives a bound that is $\tilde{O} \left( \min\{\frac{k}{\varepsilon n}, \frac{1}{\sqrt{\varepsilon n}}\} \right)$, which is much better than the worst case bound for large support distributions.

Our result also applies to other practically important settings such as building lists of popular sequences such as n-grams over words. We leave open the questions of designing instance-optimal algorithms for other practically important questions in private learning and statistics, and of designing better instance optimal algorithms for higher dimensional spaces. We also leave open the question of removing the polylogarithmic factors in our instance optimality bounds.

## 1.2 Techniques

### 1.2.1 Distributions over $\mathbb{R}$:

We start by describing the rate we obtain for distributions $P$ over $\mathbb{R}$. In order to state the rate, we will use $q_\alpha$ to represent the $\alpha$-quantile of the distribution $P$ and use $P|_{a,b}$ to define a certain restricted distribution described below. The rate consists of three terms and roughly looks as follows— we suppress logarithmic factors in $n$.

$$R_{\mathcal{A},n}(P) = \tilde{O} \left( \mathbb{E} \left[ \mathcal{W} \left( P, \hat{P}_n \right) \right] + \frac{1}{\varepsilon n} \left( q_{1 - \frac{1}{\varepsilon n}} - q_{\frac{1}{\varepsilon n}} \right) + \mathcal{W}(P, P|_{q_{\frac{1}{\varepsilon n}}, q_{1 - \frac{1}{\varepsilon n}}}) \right),$$

The first term is $\mathbb{E}[\mathcal{W}(P, \hat{P}_n)]$, the expected Wasserstein distance between the true distribution and the empirical distribution over $n$ samples, and is the non-private term. The remaining two terms

represent the cost of privacy- the first is a specific interquantile distance, roughly $\frac{1}{\varepsilon n}(q_{1-\frac{1}{\varepsilon n}} - q_{\frac{1}{\varepsilon n}})$, and the second can be thought of as capturing the weight of the tails- represented by the Wasserstein distance between $P$ and a 'restricted' version of $P$ with its tails chopped off (i.e. the cumulative distribution function is 0 below $q_{1/\varepsilon n}$ and 1 above $q_{1-1/\varepsilon n}$ and identical to $P$ otherwise). Observe that all 3 of the terms above are smaller for distributions with small support or greater concentration, and hence the rate adapts to the hardness of the distributions.

**Upper Bounds:** The upper bound involves estimating roughly $\varepsilon n$ equally spaced quantiles of the empirical distribution differentially privately (using a known private CDF estimation algorithm), and placing roughly $1/\varepsilon n$ mass at each of the estimated quantile points. For the analysis, the intuition for each of the terms is as follows: since we only have access to the empirical distribution, the non-private term $\mathbb{E}[\mathcal{W}(P, \hat{P}_n)]$ comes from that. Next, if the quantile estimates are good, then the pointwise CDF differences between the empirical distribution and the estimated distribution are at most $1/\varepsilon n$ (due to the discretization), and so we will pay $1/\varepsilon n$ multiplied by the interquantile distance of the empirical distribution. This aligns with the accuracy of state-of-the-art DP quantile estimation algorithms. Finally, since the distribution is restricted to the estimated quantiles, the distribution is 0 before the first estimated quantile and 1 above the last estimated quantile and so we pay the Wasserstein distance between the empirical distribution and a restricted version of the empirical distribution. Some care needs to be taken while reasoning about expectation versus high probability (for various terms), and in relating population quantities to empirical quantities (which we do using various concentration inequalities). Details can be found in Section E.2.

**Lower Bounds:** We prove that the private and non-private terms are lower bounds separately. Both proofs follow the same framework. The idea is that given knowledge of two distributions $P$ and $Q$, we can use a (private) Wasserstein estimation algorithm to construct a hypothesis test distinguishing $P$ from $Q$. If the (private) estimate for $P$ and $Q$ with $n$ samples gives error smaller than $\frac{1}{2}\mathcal{W}(P, Q)$, we can use this to distinguish $P$ from $Q$. This would give a contradiction if $P$ and $Q$ are (privately) indistinguishable with $n$ samples. Hence, this would give a lower bound of $\frac{1}{2}\mathcal{W}(P, Q)$ on the error of the Wasserstein estimation algorithm on $P$ or $Q$.

Thus the task reduces to constructing a distribution $Q$ that satisfies three properties: 1) it is (privately) indistinguishable from $P$ given $n$ samples, 2) the Wasserstein distance between $P$ and $Q$ is sufficiently large, 3) $D_\infty(P, Q) \le \ln 2$. The main technical work is in identifying a distribution $Q$ that satisfies these properties.

For the privacy term, we construct the distribution $Q$ by taking half the mass from the first $1/\varepsilon n$-quantile of $P$ (scaling the density function by half) and moving it to the last $1/\varepsilon n$-quantile of $P$ (scaling the density function by $3/2$). The third property is satisfied by definition, so we reason about the other two. Intuitively, since the Wasserstein distance captures how hard it is to 'move' $P$ to $Q$, this mass needs to move at least the interquantile distance to change $P$ to $Q$. This implies that the Wasserstein distance is at least the interquantile distance scaled by $1/\varepsilon n$, as described in the rate. Additionally, mass that is further out in the tail needs to move more, which is captured by the Wasserstein distance between the distribution $P$ and its 'restriction'. Hence, the Wasserstein distance between $P$ and $Q$ is lower bounded by these two terms of interest. The intuition behind Property 2 is that it is hard for any $\varepsilon$-DP algorithm to pinpoint the location of an $\frac{1}{\varepsilon n}$-fraction of the points in the dataset. Overall, this shows the privacy lower bound.

The non-private lower bound requires a more careful construction of $Q$. We divide $P$ into various scales and carefully adjust them differently to obtain the desired properties. Formally, to construct $Q$ from $P$, we consider $q_{1/2}$ and all quantiles of the form $q_{1/2^i}$ and $q_{1-1/2^i}$ for $i > 1$. For $1 \le i < \log n$, we add mass to $[q_{1/2^{i+1}}, q_{1/2^i})$, by setting the density $f_Q$ to be $(1 + \sqrt{2^i/n})f_P$ and balance out the extra mass by setting $f_Q$ to be $(1 - \sqrt{2^i/n})f_P$ between $[q_{1-1/2^i}, q_{1-1/2^{i+1}})$. For $i \ge \log n$ (i.e. the tail), we add mass to $[q_{1/2^{i+1}}, q_{1/2^i})$, by setting $f_Q$ to be $(1 + \frac{1}{2})f_P$ and balance out the extra mass by setting $f_Q$ to be $(1 - \frac{1}{2})f_P$ between $[q_{1-1/2^i}, q_{1-1/2^{i+1}})$.

The third property is again trivially satisfied. For the first property, observe that to 'move' $P$ to $Q$ the extra $\frac{1}{\sqrt{2^i n}}$ mass between $[q_{1/2^{i+1}}, q_{1/2^i})$ has to 'travel' between $q_{1/2^i}$ and $q_{1-1/2^i}$, and so the Wasserstein distance between $P$ and $Q$ can be lower bounded by a sum of various scaled interquantile distances. We attempt to upper bound the expected Wasserstein distance between $P$ and $\hat{P}_n$ by a similar term. It is more intuitive to reason about this using an alternative (equivalent) for-

mulation of Wasserstein distance as the area between the CDF curves of $P$ and $Q$. The intuition is that the expected pointwise CDF difference between $P$ and $\hat{P}_n$ in the interval $[q_{1/2^{i+1}}, q_{1/2^i})$ would be roughly $\frac{1}{\sqrt{2^i n}}$ (by properties of a Binomial) and hence the contribution of this interval to the area would be roughly $\frac{1}{\sqrt{2^i n}}\left(q_{1/2^i} - q_{1/2^{i+1}}\right)$ and similarly for the corresponding interval $[q_{1-1/2^i}, q_{1/2^{i+1}})$. Hence, the expected Wasserstein distance would be a sum of these scaled quantile interval distances. We formalize this intuition using a result of Bobkov and Ledoux [BL19] that character­izes the expected Wasserstein distance between $P$ and $\hat{P}_n$ as an integral of a function of the CDF of $P$. We now have a bound in terms of the sum of scaled quantile interval distances, but we want to bound it by a sum of scaled *interquantile* distances. We can telescope the sum to indeed bound it by a sum of scaled *interquantile* distances. This establishes that $\mathcal{W}(P, Q) \geq \mathbb{E}[\mathcal{W}(P, \hat{P}_n)]$. Next, we show that $P$ is indistinguishable from $Q$ by analyzing the KL divergence between $P$ and $Q$. The main idea is that high density intervals are modified by a small multiplicative factor of roughly $1 + \frac{1}{\sqrt{n}}$, but low density intervals (with mass less than $1/n$) are modified by a constant multiplicative factor, so overall the contribution of each interval to the KL divergence is sufficiently small. This establishes indistinguishability with $n$ samples. For formal details we refer the reader to Section E.1.

### 1.2.2 Distributions on HSTs

Since the main technical challenge of proving Theorem 1.4 is proving the equivalent result for distributions on HST metric spaces, we focus on that problem in this section. Standard results on low distortion embeddings of metric spaces into HST metric spaces can be used to translate the HST result to $[0, 1]^2$ and to general metric spaces $X$ with $\log |X|$ overhead.

**Definition 1.5** (Hierarchically Separated Tree). *A hierarchically separated tree (HST) is a rooted weighted tree such that the edges between level $\ell$ and $\ell - 1$ all have the same weight (denoted $r_\ell$) and the weights are geometrically decreasing so $r_{\ell+1} = (1/2)r_\ell$. Let $D_T$ be the depth of the tree.*

HSTs can be defined with any geometric scaling but we will only need a factor of 2 in this work. HSTs may also have arbitrary degree. A HST defines a metric on its leaf nodes by defining the distance between any two leaf nodes to be the weight of the minimum weight path between them.

HST metric spaces are particularly well-behaved when working with the Wasserstein distance since the Wasserstein distance on a HST has a simple closed form. A distribution $P$ on the the underlying metric space in a HST induces a function $\mathfrak{G}_P$ on the nodes of the tree where the value of a node $\nu$ is given by the weight in $P$ of the leaf nodes in the subtree rooted at $\nu$. For every level $\ell \in [D_T]$ of the tree, let $P_\ell$ be the distribution induced on the nodes at level $\ell$ where the probability of node $\nu$ is $\mathfrak{G}_P(\nu)$. Thus $P_\ell$ is a discrete distribution on a domain of size $N_\ell$, where $N_\ell$ is the number of nodes in level $\ell$ of the tree.

**Lemma 1.6** (Closed form Wasserstein distance formula). *Given two distributions $P$ and $Q$ defined in a HST metric space, the Wasserstein distance between $P$ and $Q$ has the closed formula:*

$$\mathcal{W}(P, Q) = \frac{1}{2}\sum_\nu r_\nu|\mathfrak{G}_P(\nu) - \mathfrak{G}_Q(\nu)| = \sum_\ell r_\ell \mathrm{TV}(P_\ell, Q_\ell),$$

*where $r_\nu$ is the weight of the edge connecting $\nu$ to its parent, and the sum is over all nodes in the tree.*

We will call a node $\nu$ $\alpha$-*active* under the distribution $P$ if $\mathfrak{G}_P(\nu) \geq \alpha$. Let $\gamma_P(\alpha)$ be the set of $\alpha$-active nodes under $P$ and $\gamma_{P_\ell}(\alpha)$ be the set of $\alpha$-active nodes at level $\ell$. Then there exists an algorithm $\mathcal{A}$ such that given a distribution $P$, $\varepsilon > 0$, and $n \in \mathbb{N}$,

$$\mathcal{R}_{\mathcal{A},n}(P) = \tilde{O}\left(\max_\ell r_\ell \sum_{x \in [N_\ell]} \min\left\{P_\ell(x)(1 - P_\ell(x)), \sqrt{\frac{P_\ell(x)(1 - P_\ell(x))}{n}}\right\} + \sum_{x \notin \gamma_{P_\ell}(2\kappa)} P_\ell(x) + (|\gamma_{P_\ell}(2\kappa)| - 1)\kappa\right),$$

where the max is over all the levels of the tree and $\kappa = \Theta(\frac{\log(n)}{\varepsilon n})$. Further, this bound matches (up to logarithmic factors) the lower bound $\min_{\varepsilon\text{-DP } \mathcal{A}'} \sup_{P':D_\infty(P,P') \leq \ln 2} \mathbb{E}[\mathcal{W}(\mathcal{A}'(\hat{P}'_{n'}), P')]$ where $n' \approx \frac{n}{\mathrm{polylog}\, n}$. The error rate $\mathcal{R}_{\mathcal{A},n}$ does indeed adapt to easy instances as we expected. The error decomposes into three components. The first component is the non-private sampling error; the error that would occur even if privacy was not required. The second component indicates that we can not

privately estimate the value of nodes that have probability less than $\approx 1/(\varepsilon n)$. The third component is the error due to privacy on the active nodes. If $P$ is highly concentrated then we expect most nodes to either be $\kappa$-active or have weight 0, so the first two terms in $\mathcal{R}_{\mathcal{N},n,\varepsilon}(P)$ are small. There should also be few active nodes, making the last term also small. Conversely, if $P$ has a large region of low density then we expect a large number of inactive nodes, as well as non-zero inactive nodes that are at higher levels of the tree and hence contribute more to the final term. Thus, in distributions with high dispersion we expect the right hand side to be large.

**Upper Bounds:** As in the one-dimensional setting, we want to restrict to only privately estimating the density at a small number ($\approx \varepsilon n$) of points. While we could try to mimic the one-dimensional solution by privately estimating a solution to the $\varepsilon n$-median problem, it's not clear how to prove such an approach is instance-optimal. It turns out that a simpler solution more amenable to analysis will suffice. Our algorithm has two stages; first we attempt to find the set of $\kappa$-active nodes, then we estimate the weight of these active nodes. Since these nodes have weight greater than $\frac{\log(n)}{\varepsilon n}$, we can privately estimate them to within constant multiplicative error. Any nodes that are not detected as active, are initially ascribed a weight of 0. The error due to not estimating the non-active nodes is absorbed into the third error term. The final step is to project the noisy density function into the space of distributions on the underlying metric space. The error of the upper bound algorithm is summed over all levels of the tree, although since the depth of the tree is logarithmic in the size of the metric space, this is within a logarithmic factor of the maximum over the levels.

**Lower Bound:** We first observe that in order to estimate the distribution well in Wasserstein distance, an algorithm must estimate each level of the tree well in TV distance. This is derived from Lemma 1.6. This allows us to reduce to the problem of lower bounding the error of density estimation of discrete distributions in TV distance. The main tool we use is a differentially private version of Assouad's method. Similar to how the technique in the previous section allowed us to relate lower bounding estimation rates to simple hypothesis testing, Assouad's lemma allows us to relate lower bounding estimation rates to multiple hypothesis testing. Note that unlike the technique in the previous section, Assouad's lemma allows us to prove lower bounds on the expected error, rather than lower bounds on high probability error bounds. It involves constructing nets of distributions in $\mathcal{N}(P)$ that are pairwise far in the relevant metric of interest (which for us in the TV distance) but the multiple hypothesis testing problem between the distributions is sufficiently hard. For proving the third term belongs in the lower bound, the standard statement of DP Assouad's lemma [ASZ21] suffices, where one builds a set of distributions indexed by a hypercube. For the first and second terms, we need to slightly generalise the statement to allow for sets of distributions indexed by a product of hypercubes. We use the approximate DP version of DP Assouad's so while our upper bounds are for pure differential privacy, our lower bounds hold for both pure and approximate differential privacy.

Let us start with the third term. Suppose the number of active nodes is even (a small tweak is made if there is an odd number of active nodes). We pair up the active nodes and index each pair by a coordinate of the hypercube. For each corner of the hypercube, $(u^0, u^1, \cdots, u^k) \in \{\pm 1\}^k$, for each coordinate $j \in [k]$, if $u^j = +1$, we move $\tilde{O}(\kappa)$ mass from one node in the $j$th pair to the other node. If $u^j = -1$ then we leave the $j$th pair of nodes alone. Since each active node has mass $> \kappa$, it's clear that each resulting distribution belongs in $\mathcal{N}(P)$. We can also show that these distributions form a sufficiently hard multiple hypothesis testing problem. By DP Assouad's (Lemma D.8), this allows us to lower bound the estimation error by $\Omega(k\kappa)$, which is within $\tilde{\Omega}$ of the third term when the number of active nodes is $\geq 2$. We treat the case where there is a single active node separately.

For the second term, we want to pair up the inactive nodes in a similar manner and move half their mass from one node to the other. However, since we want to remain within $\mathcal{N}(P)$, we can't pair any two inactive nodes together. Thus, we divide the inactive nodes into *scales*, where nodes within a certain scale all have weight within a multiplicative factor of two. We then pair up nodes within each scale and have a different hypercube for each scale. Again, it's clear that these distributions are all in $\mathcal{N}(P)$ and we can show that these distributions form a sufficiently hard multiple hypothesis testing problem. The proof for the first term follows similarly.

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

## Organization of Appendices

# A Preliminaries

For all distributions $P$, we will use $f_P$ to denote the density of $P$ (when it exists) and $F_P$ to denote the cumulative distribution function of $P$. Given a space $\mathcal{X}$, let $\Delta(\mathcal{X})$ be the set of distributions on the space $\mathcal{X}$. Given a logical statement $a$, let $\chi_a = 0$ if $a$ is false and 1 if $a$ is true. For example, $\chi_{0=0} = 1$ and $\chi_{0=1} = 0$.

A number of distances between distributions are important in this work. We start by defining the infinity divergence, which is important in the notion of instance optimality we use.

**Definition A.1** ($D_\infty$-divergence). *Given two distributions $P$ and $Q$ with the same support, the $\infty$-Rényi divergence $D_\infty(P,Q) = \ln\sup_t \max\left\{\frac{P(t)}{Q(t)}, \frac{Q(t)}{P(t)}\right\}$, if $P$ and $Q$ are discrete, and $D_\infty(P,Q) = \ln\sup_t \max\left\{\frac{f_P(t)}{f_Q(t)}, \frac{f_Q(t)}{f_P(t)}\right\}$, if $P$ and $Q$ are continuous distributions on $\mathbb{R}$, and have density functions. If $P$ and $Q$ don't have the same support, then $D_\infty(P,Q) = \infty$.*

We will use $\mathrm{KL}(P,Q)$ to denote the KL-divergence, $H^2(P,Q)$ to denote the squared Hellinger divergence and $\mathrm{TV}(P,Q)$ to denote the total variation distance, defined later.

**Wasserstein Distance:** The error metric that we use to judge our performance on the density estimation task is 1-Wasserstein distance (that we will call just Wasserstein distance where it is clear from context). In this subsection, we define Wasserstein distance.

**Definition A.2.** *For any separable metric space $(E, D)$, let $P, Q$ represent Borel measures on $E$. Then, the 1-Wasserstein distance between $P, Q$ is defined as*

$$\mathcal{W}(P,Q) = \inf_\pi \int_E \int_E D(t, t_0)\pi(x, x_0),$$

*where the infimum is over all measures $\pi$ on the product space $E \times E$ with marginals $P$ and $Q$ respectively.*

Finally, for one dimensional real spaces where the metric of interest is $\ell_1$ norm, we will use the following equivalent formulation of Wasserstein distance extensively.

**Lemma A.3** (Wasserstein formula over $\mathbb{R}$). *Let $P, Q$ represent probability distributions on $\mathbb{R}$ with finite expectation. Then, the 1-Wasserstein distance between $P, Q$ is equal to*

$$\mathcal{W}(P,Q) = \int_\infty^\infty |F_P(t) - F_Q(t)|dt,$$

*where the $F(\cdot)$ represents the cumulative distribution function.*

Given an metric space $\mathcal{X}$, the Wasserstein metric is a well-defined metric on the set of the probability distributions over $\mathcal{X}$.

## A.1 Distribution Distances

A number of other distances between distributions are used in this work.

**Definition A.4** ($KL$-divergence). *Given two distributions $P$ and $Q$ with $supp(P) \subseteq supp(Q)$, the KL divergence $KL(P,Q) = \sum_{t \in supp(P)} P(t) \ln \frac{P(t)}{Q(t)}$, if $P$ and $Q$ are discrete, and $KL(P,Q) = \int_{t \in \mathbb{R}: f_P(t) > 0} f_P(t) \ln \frac{f_P(t)}{f_Q(t)} dt$ if $P$ and $Q$ are distributions on $\mathbb{R}$, and have density functions. If $supp(P) \not\subseteq supp(Q)$, then $KL(P,Q) = \infty$.*

**Definition A.5** (Hellinger distance). *Given two distributions $P$ and $Q$, the Hellinger distance $H(P,Q) = \frac{1}{\sqrt{2}}\|\sqrt{P} - \sqrt{Q}\|_2$ (where we think of $P$ and $Q$ as vectors representing the probability masses, and the square root being component-wise.), if $P$ and $Q$ are discrete. If $P$ and $Q$ are distributions on $\mathbb{R}$, and have density functions, then $H(P,Q) = \frac{1}{\sqrt{2}}\sqrt{\int_{t \in \mathbb{R}: f_P(t) > 0} (\sqrt{f_P(t)} - \sqrt{f_Q(t)})^2 dt}$.*

Note that we use $H^2(P,Q)$ to represent the squared Hellinger distance. Next, we define total variation distance, which will come up in our high-dimensional results.

**Definition A.6** (Total Variation distance). *Given two discrete distributions $P$ and $Q$, the Total Variation distance $TV(P, Q) = \frac{1}{2}\|P - Q\|_1$, (where we think of $P$ and $Q$ as vectors representing the probability masses). More generally, for any two probability measures $P$ and $Q$ defined on $(\Omega, \mathcal{F})$, the total variation distance is defined as $\sup_{A \in \mathcal{F}} |P(A) - Q(A)|$ where $P(A)$ represents the probability of $A$ under measure $P$ and likewise for $Q$.*

We use the following relationship between Hellinger distance and KL divergence.

**Lemma A.7.** *For all distributions $P, Q$ such that KL-divergence of $P, Q$ is well defined, we have that*

$$H^2(P, Q) \leq KL(P, Q), \quad H^2(P, Q) \leq TV(P, Q) \tag{2}$$

### A.2 Differential Privacy

**Lemma A.8** (Post-Processing [DMNS17]). *If Algorithm $\mathcal{A} : \mathcal{X}^n \to \mathcal{Y}$ is $(\varepsilon, \delta)$-differentially private, and $\mathcal{B} : \mathcal{Y} \to \mathcal{Z}$ is any randomized function, then the algorithm $\mathcal{B} \circ \mathcal{A}$ is $(\varepsilon, \delta)$-differentially private.*

Secondly, differential privacy is robust to adaptive composition.

**Lemma A.9** (Composition of $(\varepsilon, \delta)$-differential privacy [DMNS17]). *If $\mathcal{A}$ is an adaptive composition of $m$ differentially private algorithms $\mathcal{A}_1, \ldots, \mathcal{A}_m$, where $\mathcal{A}_j$ is $(\varepsilon_j, \delta_j)$ differentially private, then $\mathcal{A}$ is $\left(\sum_j \varepsilon_j, \sum_j \delta_j\right)$-differentially private.*

Finally, we discuss the Laplace mechanism, which we will use in one of our algorithms.

**Definition A.10** ($\ell_1$-Sensitivity). *The $\ell_1$-sensitivity of a function $f : \mathcal{X}^n \to \mathbb{R}^d$ is*

$$\Delta_f = \max_{\substack{\mathbf{x}, \mathbf{x}' \in \mathcal{X}^n \\ d_{ham}(\mathbf{x}, \mathbf{x}') \leq 1}} \|f(\mathbf{x}) - f(\mathbf{x}')\|_1.$$

**Lemma A.11** (Laplace Mechanism). *Let $f : \mathcal{X}^n \to \mathbb{R}^d$ be a function with $\ell_1$-sensitivity $\Delta_f$. Then the Laplacian mechanism is algorithm*

$$\mathcal{A}_f(\mathbf{x}) = f(\mathbf{x}) + (Z_1, \ldots, Z_d),$$

*where $Z_i \sim Lap\left(\frac{\Delta_f}{\varepsilon}\right)$ (and $Z_1, \ldots, Z_d$ are mutually independent). Algorithm $\mathcal{A}_f$ is $\varepsilon$-DP.*

## B On Instance Optimality

In this section, we discuss the notion of instance optimality, and argue that it provides a useful benchmark that captures the idea of *going beyond the worst case*. The notion of instance optimality we propose can be see as a generalisation of the hardest one-dimensional subproblem, or hardest local alternative introduced by [CL15]. Suppose we have a family of distributions $\mathcal{P} \subset \Delta(\mathcal{X})$ on a space $\mathcal{X}$ and our goal is to learn the parameter $\theta : \mathcal{P} \to \mathcal{M}$ where $\mathcal{M}$ is a metric space with metric $d$. Given an estimation algorithm $\mathcal{A} : \mathcal{X}^n \to \mathcal{M}$, we can define the estimation rate[3] of $\mathcal{A}$ to be the function $\mathcal{R}_{\mathcal{A}, n} : \mathcal{P} \to \mathbb{R}_+$ where

$$\mathcal{R}_{\mathcal{A}, n}(P) = \mathbb{E}_{D \sim P^n}[d(\theta(P), \mathcal{A}(D))].$$

Since the estimation rate is a function of the distribution $P$, the estimation rate of an algorithm may be lower at "easy" distributions and larger at "harder" distributions. As a classic example, consider the estimation rate of Bernoulli parameter estimation where $\mathcal{A}$ simply outputs the empirical mean. Then $\mathcal{R}_{\mathcal{A}, n}(\text{Ber}(p)) = \min\{p(1 - p), \sqrt{p(1 - p)/n}\}$, so this algorithm performs better when the Bernoulli parameter is close to 0 or 1, and has it's worst case error when $p = 1/2$.

Cai and Low [CL15] proposed three desiderata that a target estimation rate $\mathcal{R}_n : \Delta(\mathcal{X}) \to \mathbb{R}_+$ should satisfy in order to be a meaningful benchmark;

---

[3]We note that while the estimation rate here is defined in expectation, we will sometimes show results (for e.g. in the one-dimensional case) where estimation rate is defined with probability at least 0.75 over the randomness of the algorithm and the data; see Equation 1.

1. $\mathcal{R}_n(P)$ varies significantly across $\mathcal{P}$

2. $\mathcal{R}_n$ is an achievable estimation rate; there exists an algorithm $\mathcal{A}$ and constant $\alpha$ such that $\mathcal{R}_{\mathcal{A},n}(P) \leq \alpha \mathcal{R}_n(P)$ for all $P \in \mathcal{P}$

3. Outperforming the benchmark $\mathcal{R}_n$ at one distribution leads to worse performance at another distribution.

In this section we will discuss the definition of instance optimality we will use in this work by defining the target estimation rate that will serve as our benchmark estimation rate. The main theorems of this paper establish that our chosen benchmark achieves desiderata 1 and 2 above. It is not immediately obvious that desiderata 3 holds. We will show in Section B.2, through the introduction of a related notion of instance optimality which we call *local minimality*, that desiderata 3 holds in many important settings, including the problem studied in this paper.

## B.1 Local Estimation Rates

We will start by defining a target estimation rate. We'll say an algorithm is $\alpha$-instance optimal if it uniformly achieves this target estimation rate up to a multiplicative $\alpha$ factor. For each distribution $P \in \mathcal{P}$, we define a neighbourhood $\mathcal{N}(P)$.

**Definition B.1.** *Given a function $\mathcal{N} : \mathcal{P} \to \mathfrak{P}(\mathcal{P})$, where $\mathfrak{P}(\mathcal{P})$ is the power set of $\mathcal{P}$, we define the optimal estimation rate with respect to $\mathcal{N}$ to be:*

$$\mathcal{R}_{\mathcal{N},n}(P) = \min_{\mathcal{A}} \sup_{Q \in \mathcal{N}(P)} \mathcal{R}_{\mathcal{A},n}(Q). \tag{3}$$

*An algorithm $\mathcal{A}$ is $\alpha$-instance optimal with respect to $\mathcal{N}$ if for all $P \in \mathcal{P}$,*

$$\mathcal{R}_{\mathcal{A},n}(P) \leq \alpha \mathcal{R}_{\mathcal{N},n}(P)$$

If an algorithm $\mathcal{A}$ uniformly achieves the optimal estimation rate wrt a function $\mathcal{N}$, then this implies that for all distributions $P$, the error of the algorithm $\mathcal{A}$ on $P$ is competitive with an algorithm that is told the additional information that the distribution is in $\mathcal{N}(P)$. Given a function $\mathcal{N}$, it is possible that there does not exist an algorithm that uniformly achieves $\mathcal{R}_{\mathcal{N},n}$. For example, as discussed in the introduction, if $\mathcal{N}(P) = \{P\}$, then $\mathcal{R}_{\mathcal{N},n}$ is not uniformly achievable. Conversely, if $\mathcal{N}(P)$ is not chosen carefully, then the estimation rate $\mathcal{R}_{\mathcal{N},n}$ may not define a meaningful benchmark; e.g. an estimation rate that adapts to easy instances.

A different formalization may be more probabilistic: the algorithm designer may have in mind a distribution $\mathcal{D}$ over distributions that they care about, and their objective may be to minimize $\mathbb{E}_{P \sim \mathcal{D}}[\mathcal{R}_{\mathcal{A},n}(P)]$. Suppose that for the $\mathcal{A}^\star$ chosen by the algorithm designer, and for our neighborhood map $\mathcal{N}$, the function $\mathcal{R}_{\mathcal{A}^\star,n}(P)$ does not vary too much over $\mathcal{N}(P)$ on average. Formally, let

$$disc_{\mathcal{A}^\star}^{\mathcal{N}}(P) = \sup_{P' \in \mathcal{N}(P)} (\mathcal{R}_{\mathcal{A}^\star,n}(P') - \mathcal{R}_{\mathcal{A}^\star,n}(P)$$

and let $\overline{disc}_{\mathcal{A}^\star}^{\mathcal{N}}(P) = \mathbb{E}_{P \sim \mathcal{D}}[disc_{\mathcal{A}^\star}^{\mathcal{N}}(P)]$. Then for any algorithm $\mathcal{A}$ that is $\alpha$-instance optimal with respect to $\mathcal{N}$, we can write

$$\mathbb{E}_{P \sim \mathcal{D}}[\mathcal{R}_{\mathcal{A},n}(P)] \leq \alpha \cdot \mathbb{E}_{P \sim \mathcal{D}}[\sup_{P' \in \mathcal{N}(P)} \mathcal{R}_{\mathcal{A}^\star,n}(P')]$$

$$= \alpha \cdot \mathbb{E}_{P \sim \mathcal{D}}[\mathcal{R}_{\mathcal{A}^\star,n}(P) + disc_{\mathcal{A}^\star}^{\mathcal{N}}(P))]$$

$$= \alpha \cdot \mathbb{E}_{P \sim \mathcal{D}}[\mathcal{R}_{\mathcal{A}^\star,n}(P)] + \alpha \cdot \mathbb{E}_{P \sim \mathcal{D}}[disc_{\mathcal{A}^\star}^{\mathcal{N}}(P))]$$

$$= \alpha \cdot \left( \mathbb{E}_{P \sim \mathcal{D}}[\mathcal{R}_{\mathcal{A}^\star,n}(P)] + \overline{disc}_{\mathcal{A}^\star}^{\mathcal{N}}(P) \right).$$

In other words, as long as the algorithm $\mathcal{A}^\star$'s performance is relatively constant over $\mathcal{N}(P)$ on average over the distribution of interest, the instance optimal algorithm (that is not tailored to $\mathcal{D}$) is competitive with $\mathcal{A}^\star$. A similar result holds for a multiplicative definition of $disc$.

This discussion can help guide the choice of the neighborhood function that is appropriate for a particular application. In the case of density estimation in the Wasserstein distance, we will define $\mathcal{N}(P)$ be a small $D_\infty$ ball around $P$. We believe this captures the kind of domain information an algorithm designer may have. E.g. one may have a small amount of public data samples, in which case the posterior over distributions in a $D_\infty$ ball will be relatively constant. If the algorithm designer's custom algorithm needs to do well for all distributions in this set, an instance-optimal algorithm will be competitive with this custom algorithm.

Previous work in instance optimality has largely focused on two notions of neighborhood. In [FLN01, ABC17, VV16, OS15], where the objects of interest are discrete subsets with no a priori structure, it is natural to ask that the algorithm work well for any permutation of the inputs. For example, if the goal is to compute the set of maximal points from a 2-d point set, the algorithm designer would typically want an algorithm that works well for any permutation of the set of input points. In our setting where the points of interest have a metric structure, this is not an appropriate notion. In fact, even for the discrete case studied in Appendix D.2.2, permutation invariance cannot capture natural prior beliefs that may arise in practice. For example, for power-law distributions that one often sees in private learning applications [ZKM+20, CB22, CCD+23], a small number of samples are sufficient to get a good estimate of the heavy bins, and rule out a large fraction of permutations of the input space.

A second line of work arising from the statistics literature [CL15] has looked at defining instance-optimality with respect to neighborhoods of size 2. While this approach has been very successful for many problems, we find it inappropriate for density estimation (outside of density estimation on $\mathbb{R}$) as neighborhoods of size two are too weak to capture the difficulty of problems of interest. Even in the simple case of discrete distributions, this neighborhood is provably insufficient to get instance-optimality results with any $o(K)$ competitive ratio. Indeed, for any two given distributions on $[K]$ with TV distance $\alpha$, $\tilde{O}(\frac{1}{\alpha^2})$ samples suffice to distinguish them, whereas learning a near uniform distribution on $K$ atoms requires $\Omega(K)$ samples. In the private setting, the need to use multiple distributions to prove lower bounds is well-studied. Our approach shares this similarity of using a multi-instance lower bounding argument with packing lower bounds in privacy, and local Fano's and Le Cam's methods in statistics. Our work shows that some of the same lower bounding techniques can be used to prove instance-optimality results with respect to natural neighborhood maps, going well beyond the the worst-case results those works prove.

In the special case of density estimation in the Wasserstein distance on $\mathbb{R}$, instance optimality with respect to neighborhoods of size 2 is achievable. In the standard version of this benchmark metric, $\mathcal{N}(P) = \{P, Q_P\}$ where $Q_P$ can be *any* distribution and is chosen to maximise $\mathcal{R}_{\mathcal{N},n}(P)$. However, this notion may not be an appropriate notion of instance optimality by itself. To see this, consider a distribution $P$ supported on an interval $[a, b]$. Moving a small amount of mass from one end of the interval to the other would create an indistinguishable distribution that is far from $P$ in Wasserstein distance, and a hypothesis testing argument can be used to show that the target estimation rate defined above (for the hardest one-d sub problem) depends on the interval size $b - a$. This implies that the adaptivity of algorithms to support size of the distribution (crucial in Wasserstein estimation) is not captured by this notion of instance optimality. Instead, we add a further restriction to the definition to make it more appropriate for our setting; we only consider distributions $Q$ that are in a small $D_\infty$ ball around $P$ ($D_\infty(P, Q) \le \ln 2$), and ask that an algorithm is competitive with an algorithm that is told the additional information that $P \in \{P, Q\}$ (in the worst case over distributions $Q$ that are in this $D_\infty$ ball). That is, we define the benchmark estimation rate to be

$$\mathcal{R}_{\mathrm{loc},n}(P) = \sup_{Q:D_\infty(P,Q)\le\ln 2} \min_{\mathcal{A}}\{\mathcal{R}_{\mathcal{A},n}(Q), \mathcal{R}_{\mathcal{A},n}(P)\}. \tag{4}$$

Note that all such distributions $Q$ have the same support as $P$, which allows us to capture the adaptivity of algorithms to the support size of the distribution. Specifically, we define the following target estimation rate in the one-dimensional setting. In the case of estimating distributions on a bounded subset of $\mathbb{R}$, we will show that this error rate is achievable, up to logarithmic factors.

We also note that our notion of instance optimality more naturally captures the accuracy of algorithms even for basic tasks. Note that for the Bernoulli case, our technique achieves a bound of $\frac{\sqrt{p(1-p)}}{\sqrt{n}} + \min\{p, 1-p, \frac{1}{\varepsilon n}\}$ which also appear to be better than the instance-optimal lower bounds in [MSU22], which take the form $\frac{\sqrt{p(1-p)}}{\sqrt{n}} + \frac{1}{\varepsilon n}$. This apparent contradiction can be explained by

the the use in [MSU22] of the hardest-one dimensional sub-problem to define the instance-optimal rate, i.e., $\mathcal{N}(P)$ is $\{P, Q\}$ for a worst-case Bernoulli $Q$. On the other hand, the notion of instance-optimality we use would only consider Bernoullis $Q$ such that $D_\infty(P, Q) \leq \ln 2$. When $p$ is close to 0, the lower bound in [MSU22] would on this instance consider $Q$ to be $Bern(p + \frac{1}{\varepsilon n})$, which can have a large $D_\infty$-distance from $P$, and so isn't in the neighborhood used in our notion of instance-optimality. Hence, the target rate one would obtain from our definition is smaller when $p$ is close to 0. Our algorithm can achieve this improved rate, as it is likely to output 0 as an estimate of $p$ in this case, pushing small counts down to zero.

Recent differentially private algorithms such as those in [HLY21, DKSS23] have shown instance-optimality for problems such as mean estimation. Relatedly, other works have designed algorithms that adapt to the local/smooth/deletion sensitivity of the underlying function. An instance in these works in a dataset rather than a distribution, and it is not clear how to extend the corresponding notion of neighborhood to our setting. Our neighborhood notion perhaps comes closest to the deletion neighborhoods considered in some of these works.

Finally, we remark that while we have stated our results as being competitive with the worst-case instance in $\mathcal{N}(P)$, they apply for the average case over a specific distribution over $\mathcal{N}(P)$. Since that specific distribution is adversarial, we don't view this version as more natural than the worst case.

Given that we are focusing on private estimation, we will use use $\mathcal{R}_{\mathrm{loc}, n, \varepsilon}$ to denote the version of Eqn 4 where the minimum is taken over all $\varepsilon$-DP mechanisms, and $\mathcal{R}_{\mathcal{N}, n, \varepsilon}$ to define the optimal $\varepsilon$-DP estimation rate, i.e. Eqn 3 where the minimum is taken over all $\varepsilon$-DP mechanisms.

## B.2   Locally Minimal Algorithms

In this section we address the third desiderata of [CL15]. An important concept in statistics is that of efficiency of an estimator, which informally compares the rate of convergence of the estimator with a benchmark that in general is not beatable. This idea has been used to argue that for some fundamental estimation problems, the Maximum Likelihood Estimator (MLE) is the best possible. Hodge showed an example of a *superefficient* estimator that is asymptotically as good as the MLE everywhere, but beats the MLE on a certain set of inputs. The statistics community has argued in multiple ways that these superefficient estimators do not limit our ability to argue that MLE is "optimal". We refer the reader to [vdV97, Wol65, Vov09] for a discussion of superefficiency. One of the more compelling arguments here is a result saying that the set of points where superefficiency is achieved has Lebesgue measure zero. This in particular implies that in a small neighborhood around any point, there is a point (in fact many points) where the superefficient estimator does no better than the MLE. In the partial order on estimators, the MLE is thus minimal and this is true even when looking at the performance of the estimator only on a small neighborhood around a given point.

This motivates a slightly different notion capturing the goodness of the algorithm locally.

**Definition B.2.** *Let $\mathcal{M}$ be a class of algorithms. We say that an algorithm $\mathcal{A}$ is $\alpha$-locally minimal with respect to a neighborhood map $\mathcal{N}$, if for all instance $P$, and all $\mathcal{A}' \in \mathcal{M}$, there is a $Q \in \mathcal{N}(P)$ such that $\mathcal{R}_{\mathcal{A}, n}(Q) \leq \alpha \cdot \mathcal{R}_{\mathcal{A}', n}(Q)$.*

In words, local minimality says that for any other $\mathcal{A}'$, the algorithm $\mathcal{A}$ is competitive with $\mathcal{A}'$ for some instance in the neighborhood of $P$. Put differently, no $\mathcal{A}'$ can be uniformly much better than $\mathcal{A}$ on the neighborhood, even one that knows $P$.

We show that in general, this notion is incomparable to our notion of instance optimality. Nevertheless, under reasonable assumptions, the two notions are closely related.

**Example B.3** (Local Minimality $\not\Rightarrow$ Instance Optimality)**.** *Consider a pair of instances $\{P, Q\}$ with $\mathcal{N}(P) = \mathcal{N}(Q) = \{P, Q\}$. Let $\mathcal{M}$ contain two algorithms $\mathcal{A}$, and $\mathcal{A}^\star$ with*

$$\mathcal{R}_{\mathcal{A}, n}(P) = 1; \quad \mathcal{R}_{\mathcal{A}, n}(Q) = 0;$$
$$\mathcal{R}_{\mathcal{A}^\star, n}(P) = 0; \quad \mathcal{R}_{\mathcal{A}^\star, n}(Q) = 0;$$

*Then one can verify that $\mathcal{A}$ is (1-)locally minimal in $\mathcal{M}$. However, it is not $\alpha$-instance optimal for any finite $\alpha$ as it fails to satisfy the definition at $P$.*

**Example B.4** (Instance Optimality $\not\Rightarrow$ Local Minimality)**.** *Consider a set of instances $\{P_1, P_2, P_3\}$ with $\mathcal{N}(P_1) = \{P_1, P_2\}, \mathcal{N}(P_2) = \{P_1, P_2, P_3\}, \mathcal{N}(P_3) = \{P_2, P_3\}$. Let $\mathcal{M}$ contain algorithms*

$\mathcal{A}, \mathcal{A}^\star$ with

$$\mathcal{R}_{\mathcal{A}^\star, n}(P_1) = 1; \quad \mathcal{R}_{\mathcal{A}, n}(P_1) = 2\alpha;$$
$$\mathcal{R}_{\mathcal{A}^\star, n}(P_2) = 2\alpha; \quad \mathcal{R}_{\mathcal{A}, n}(P_2) = 4\alpha^2;$$
$$\mathcal{R}_{\mathcal{A}^\star, n}(P_3) = 4\alpha^2; \quad \mathcal{R}_{\mathcal{A}, n}(P_3) = 4\alpha^2.$$

*Then one can verify that $\mathcal{A}$ is (1-)instance optimal in $\mathcal{M}$. However, it is not $\alpha$-locally minimal at $P_1$.*

Under smoothness assumptions on $\mathcal{A}$ with respect to $\mathcal{N}$, one can argue that the two notions are essentially equivalent.

**Proposition B.5.** *Let $\mathcal{A}$ be such that for all instances $P$ and for all $Q \in \mathcal{N}(P)$, $\mathcal{R}_{\mathcal{A}, n}(Q) \leq \beta \cdot \mathcal{R}_{\mathcal{A}, n}(P)$. Further, suppose that $\mathcal{N}(P)$ is compact for any $P$. If $\mathcal{A}$ is $\alpha$-instance optimal in $\mathcal{M}$ with respect to $\mathcal{N}$, then it is $\alpha\beta$-locally minimal.*

*Proof.* Let $P$ be an instance and let $\mathcal{A}'$ be a competing algorithm. By definition of $\alpha$-instance optimality,

$$\mathcal{R}_{\mathcal{A}, n}(P) \leq \alpha \cdot \sup_{Q \in \mathcal{N}(P)} \mathcal{R}_{\mathcal{A}', n}(Q).$$

By compactness, this implies that there is a $Q$ achieving the supremum. In other words, there exists $Q^\star \in \mathcal{N}(P)$ such that

$$\mathcal{R}_{\mathcal{A}, n}(P) \leq \alpha \cdot \mathcal{R}_{\mathcal{A}', n}(Q^\star).$$

Since $Q^\star \in \mathcal{N}(P)$, our smoothness assumption implies that

$$\mathcal{R}_{\mathcal{A}, n}(Q^\star) \leq \beta \cdot \mathcal{R}_{\mathcal{A}, n}(P).$$

Combining the last two inequalities, this $Q^\star$ satisfies

$$\mathcal{R}_{\mathcal{A}, n}(Q^\star) \leq \alpha\beta \cdot \mathcal{R}_{\mathcal{A}', n}(Q^\star).$$

Since $P$ and $\mathcal{A}'$ were arbitrary, this implies that $\mathcal{A}$ is $\alpha\beta$-locally minimal. $\qquad \square$

**Proposition B.6.** *Let $\mathcal{A}$ be such that for all instances $P$ and for all $Q \in \mathcal{N}(P)$, $\mathcal{R}_{\mathcal{A}, n}(Q) \geq \beta^{-1} \cdot \mathcal{R}_{\mathcal{A}, n}(P)$. If $\mathcal{A}$ is $\alpha$-locally minimal in $\mathcal{M}$ with respect to $\mathcal{N}$, then it is $\alpha\beta$-instance optimal.*

*Proof.* Let $P$ be an instance and let $\mathcal{A}'$ be a competing algorithm. By definition of $\alpha$-local minimality, there is a $Q^\star \in \mathcal{N}(P)$ such that

$$\mathcal{R}_{\mathcal{A}, n}(Q^\star) \leq \alpha \cdot \mathcal{R}_{\mathcal{A}', n}(Q^\star).$$

Since $Q^\star \in \mathcal{N}(P)$, our smoothness assumption implies that

$$\mathcal{R}_{\mathcal{A}, n}(P) \leq \beta \cdot \mathcal{R}_{\mathcal{A}, n}(Q^\star).$$

Combining the last two inequalities, this $Q^\star$ satisfies

$$\mathcal{R}_{\mathcal{A}, n}(P) \leq \alpha\beta \cdot \mathcal{R}_{\mathcal{A}', n}(Q^\star)$$
$$\leq \alpha\beta \cdot \sup_{Q \in \mathcal{N}(P)} cost(\mathcal{A}'(Q), Q).$$

Since $P$ and $\mathcal{A}'$ were arbitrary, this implies that $\mathcal{A}$ is $\alpha\beta$-instance optimal. $\qquad \square$

A similar pair of results hold when the comparator algorithm $\mathcal{A}'$ is smooth with respect to the neighborhood map.

## B.3 Relaxed Definitions

We finish by noting relaxations of the above definitions that share the same semantic meaning (our algorithms will achieve these relaxed notions).

**Definition B.7.** *Given a function $\mathcal{N} : \mathcal{P} \to \mathfrak{P}(\mathcal{P})$, where $\mathfrak{P}(\mathcal{P})$ is the power set of $\mathcal{P}$, we define the optimal estimation rate with respect to $\mathcal{N}$ to be:*

$$\mathcal{R}_{\mathcal{N},n}(P) = \min_{\mathcal{A}} \sup_{Q \in \mathcal{N}(P)} \mathcal{R}_{\mathcal{A},n}(Q). \tag{5}$$

*An algorithm $\mathcal{A}$ is $(\alpha, \beta, \gamma)$-instance optimal with respect to $\mathcal{N}$ if for all $P \in \mathcal{P}$,*

$$\mathcal{R}_{\mathcal{A},n}(P) \leq \alpha \mathcal{R}_{\mathcal{N},\beta n}(P) + \gamma$$

**Definition B.8.** *Let $\mathcal{M}$ be a class of algorithms. We say that an algorithm $\mathcal{A}$ is $(\alpha, \beta, \gamma)$-locally minimal with respect to a neighborhood map $\mathcal{N}$, if for all instance $P$, and all $\mathcal{A}' \in \mathcal{M}$, there is a $Q \in \mathcal{N}(P)$ such that $\mathcal{R}_{\mathcal{A},n}(Q) \leq \alpha \cdot \mathcal{R}_{\mathcal{A}',\beta n}(Q) + \gamma$.*

Note that we think of $\beta \in (0, 1]$ and $\gamma$ as non-negative. The reason these are relaxed definitions is because we allow for an additive approximation factor in addition to a multiplicative factor, and also compare to a benchmark rate that depends on a potentially smaller number of samples (and is hence easier to achieve). The original definition of instance optimality (Definition B.1) can be obtained by setting $\beta = 1$ and $\gamma = 0$.

In our work, for most settings of interest, we roughly achieve $\beta = 1/(\log n)^{O(1)}$ and $\gamma$ to be an arbitrarily small polynomial in the inverse of the number of samples $1/n$ at a $\log(1/\gamma)$ cost to the multiplicative factor. We don't view this as a significant issue since we expect the benchmark rate with $\tilde{O}(n/\log n)$ samples to behave asymptotically similarly to that with $n$ samples in most cases. We leave it as an open question as to whether the original definition of instance optimality can be achieved.

# C Additional Related Work

**Instance Optimality for Differentially Private Statistics:** Several recent works have focused on formulating and giving 'instance optimal' differentially private algorithms for various statistical tasks. The work of McMillan, Smith and Ullman [MSU22] is most directly related to our work; they gave locally minimax optimal algorithms for parameter estimation for one-dimensional exponential families in the central model of differential privacy. The work of Duchi and Ruan [DR18] also gives locally-minimax optimal algorithms for various one-dimensional parameter estimation problems under the stronger constraint of local differential privacy. The notion of local minimax optimality both these papers use is based on the *hardest one-dimensional sub-problem* described in Section B.1. While our results for density estimation in $\mathbb{R}^1$ satisfy this notion, they also satisfy a stronger notion described in Section B.1. Additionally, as discussed in [MSU22], this definition is provably unsuitable for higher dimensions; we instead suggest a looser definition of instance optimality that is more promising in higher dimensions. More importantly, our paper is primarily focused on the non-parametric setting, and hence our techniques are different than the ones used in those papers, which focused primarily on parameter estimation.

**Other Beyond Worse-Case Results in Central Differential Privacy:** Several additional works in the differential privacy literature study algorithms with accuracy that varies with the input dataset. Nearly all of them look at the *empirical setting* where we are concerned with the specific input dataset, rather than a distribution it may be drawn from. While initial algorithms in differential privacy added noise based on a worse case notion of *global sensitivity*, these works give various algorithmic frameworks that help develop algorithms with guarantees that adapt to the hardness of the input dataset. These include algorithms based on smooth sensitivity [NRS07, BS19], the propose-test-release framework [DL09, BA20], Lipschitz extensions [BBDS13, KNRS13, CZ13, RS16], and sensitivity pre-processing [CD20]. However, none of these works study a formal notion of instance optimality.

In contrast, some more recent work do study definitions of instance optimality in the empirical setting. A work of Asi and Duchi [AD20] studies two notions of instance optimality: one by comparing

the performance of an algorithm on a dataset against the performance of the best unbiased algorithm on that dataset, and another based on an analogue of the 'hardest one-dimensional sub-problem' for the empirical setting (they compare the performance of an algorithm on a dataset with all benchmark algorithms that know that the input dataset is either of two possible datasets but whose performance is evaluated as the worse over the two datasets). They give a general mechanism known as the inverse sensitivity mechanism that they show is nearly instance optimal under these definitions for various problems such as median and mean estimation. Our work is focused on population quantities as opposed to empirical quantities—while these are related, they can be very different. For example, as pointed out in McMillan, Smith and Ullman [MSU22], using the inverse sensitivity mechanism in [AD20] to estimate the mean of a Gaussian (by using a locally minimax optimal algorithm for empirical mean) will result in infinite mean squared error, whereas other approaches that reason directly about the population quantities can get much better error.

In [DKSS23] and [HLY21], different notions of instance optimality are defined. Roughly, they compare the performance of an algorithm on a dataset with a benchmark algorithm that knows the input dataset but whose performance is evaluated as the worst-case performance over large subsets of the input dataset. While the details of the definitions in these papers vary slightly, both papers give instance-optimal algorithms for mean estimation under their respective definitions. For one-dimensional distributions, our algorithmic technique at a high level shares ideas with these algorithms—the algorithms in their papers try to adapt to the range of values in the dataset, whereas we try to adapt to the level of concentration of the distribution. However, the details of how this is done and the associated analyses vary. Our algorithm for general metric spaces uses different techniques. Our work differs from these works in a few other prominent ways: firstly, they are primarily concerned with estimating functionals of the underlying dataset, whereas we are concerned with density estimation in Wasserstein distance—these are problems with different output types and different error metrics. Finally, it is not clear if notions such as subset-based instance optimality that are well defined in the empirical setting transfer meaningfully to the distributional setting.

**Instance-Optimal Statistical Estimation without Privacy Constraints:** Donoho and Liu [DL91] formulated the notion of the 'hardest one-dimensional sub-problem' as a way of capturing instance optimality for statistical estimation and gave non-private instance optimal algorithms for some one-dimensional parameter estimation problems. Cai and Low [CL15] formulated an instance-optimality type definition for non-parameteric estimation problems. Our results for Wasserstein density estimation over $\mathbb{R}$ use a stronger version of this notion of instance optimality. In higher dimensions, this notion is provably unachievable, and so we define a different notion.

The other line of work most related to ours is on instance-optimal learning of discrete distributions [OS15, VV16, HO19]. In their setting, instance optimality is defined by comparing the performance of an algorithm on a discrete distribution $P$ to the minimax error of any algorithm on the class of discrete distributions with probability vectors that are permutations of the probability vector of $P$. We note that this notion is not well suited to many metric spaces, because permutations may not preserve properties such as concentration of the distribution, and hence this notion of instance optimality may provide an overly pessimistic view of the performance of an algorithm. Our notion of instance optimality (in terms of $D_\infty$ neighborhood) compares against algorithms with a different type of prior knowledge- i.e., the location of where the distribution concentrates, and approximate values of the probabilities at each point. We note that these are technically incomparable, and may be useful in different settings. For estimation in Wasserstein distance, knowledge of where the distribution is concentrated could be very useful in algorithm design, and so comparing to algorithms with this type of knowledge is more appropriate. See Appendix B for more discussion.

Finally, there is another line of work on getting similar instance optimal guarantees for other statistical problems [ADJ+11, ADJ+12, AJOS13b, AJOS13a]. For the closeness testing problem (given two sequences, determine if they are produced by the same distribution, or different distributions), Acharya, Das, Jafarpour, Orlitsky, Pan and Suresh [ADJ+11, ADJ+12] developed a test (without any knowledge about the generating distributions) that achieves the same error with $O(n^{3/2})$ samples that an optimal label-invariant test that knows the distributions $p$ and $q$ would achieve with $n$ samples.

**Other work on Differentially Private Statistics:** There is a lot of other work on private statistical estimation, and we survey the most relevant parts of the literature here. There is a long

line of work on minimax parameter/distribution estimation on various parametric distribution families: product distributions [BUV18, KLSU19, ASZ20, CWZ19, Sin23], Gaussian, sub-Gaussian distributions (and more generally exponential families) [KV18, KLSU19, AAK21, BGS+21, KMS+22b, KMS22a, HKM22, KMV22, AL22, LKO22, TCK+22, HKMN23, AKT+23, BHS23, KDH23], mixtures of Gaussian distributions [KSSU19, AAL23b, AAL23a], heavy-tailed distributions [KSU20, Nar23], discrete distributions with finite support [DHS15, ASZ20], distributions with finite covers [BKSW21] and more. This line of work focuses on minimax guarantees in the parametric setting, i.e. optimizing the worst-case error of an algorithm over the entire class of distributions. Our work, on the other hand works in the non-parametric setting where we do not make assumptions about the distribution the dataset is drawn from, but instead give 'instance-optimal' algorithms that adapt to the hardness of the distribution the input dataset is drawn from.

There is also a line of work on differentially private CDF estimation [DNPR10, CSS11, BNS16, BNSV15, ALMM19, KLM+20, CLN+23], and quantile estimation [KSS22, GJK21, ASSU24]. Our algorithm for density estimation over $\mathbb{R}$ uses a quantile estimation algorithm (based on a CDF estimator) as a subroutine. Finally, there is a line of work on differentially private testing [ASZ17, CDK17, CKM+19], and the work characterizing the sample complexity of simple hypothesis tests forms an important part of our analysis of the instance-optimal rate for distributions over $\mathbb{R}$.

**Work on Estimation in Wasserstein Distance:** In addition to the recent works [BSV22, HVZ23] on private Wasserstein learning on $[0,1]^d$, there is a plethora of works studying it in the non-private setting.

One line of work studies the convergence in Wasserstein distance of the empirical measure (on $n$ samples) to the true measure, as a function of the measure and the number of samples $n$ [Dud69, DY95, CR12, DSS11, BG14, FG15, BL19, WB19, Lei20, Fou23]. Some of the later works above can be viewed as studying this problem from a beyond worst-case analysis viewpoint. They give upper and lower bounds for the expected value of this quantity, in terms of various notions of 'dimension' of the underlying measure, such as the covering number of the support of the distribution, the upper and lower 'Wasserstein dimensions' of the measure, and others. Our work shows that the empirical measure, appropriately massaged, is approximately instance-optimal for density estimation without privacy constraints (for the notions of instance optimality we consider), and hence these works give us a handle on the instance-optimal rate as a function of the distribution and sample size $n$. Some more recent work studies minimax estimation in Wasserstein distance [SP19, NWB19], and show that without additional assumptions on the distribution, the empirical measure is minimax optimal. Our work extends this result to show that in the general non-parametric setting, the empirical measure is also approximately instance-optimal; to the best of our knowledge, instance optimal estimation in Wasserstein distance (even without privacy constraints) has not been previously studied.

# D  Distribution Estimation on Hierarchically Separated Trees

Let us now turn to distribution estimation on arbitrary finite metric spaces. We will use the fact that any metric on a finite space can be embedding in a hierarchically separated tree (HST) metric to reduce the problem of density estimation in Wasserstein distance on an arbitrary metric space to density estimation in Wasserstein distance on an HST. In Section D.2 we'll characterise the target estimation rate $\mathcal{R}_{\mathcal{N},n}$. In Section D.3, we'll then provide an $\varepsilon$-DP algorithm and prove that it achieves this target estimation rate up to logarithmic factors.

## D.1  Preliminaries on Hierarchically Separated Trees

A key component of our proof strategy is the reduction to Hierarchically Separated Trees (HSTs). HSTs are special class of tree metrics that are able to embed arbitrary metric spaces with low distortion. They are particularly well-behaved when working with the Wasserstein distance since the Wasserstein distance on an HST has a simple closed form.

**Definition D.1** (Hierarchically Separated Tree)**.** *A hierarchically separated tree (HST) is a rooted weighted tree such that the edges between level $\ell$ and $\ell - 1$ all have the same weight (denoted $r_\ell$) and the weights are geometrically decreasing so $r_{\ell+1} = (1/2)r_\ell$. Let $D_T$ be the depth of the tree.*

An HST defines a metric on its leaf nodes by defining the distance between any two leaf nodes to be the weight of the minimum weight path between the two nodes. We will rely on two main facts about HSTs in this work.

**Lemma D.2** (Low distortion metric embeddings [FRT03]). *Let $(V, d)$ be a metric space with $M$ points. There exists a randomized, polynomial time algorithm that produces an HST where the leaf nodes of the tree correspond to the elements of the metric space and the induced tree metric $d_T$ is such that for all $u, v \in V$*

- $d(u, v) \leq d_T(u, v)$

- $\mathbb{E}[d_T(u, v)] \leq O(\log M) \cdot d(u, v)$

*The depth of the HST is logarithmic in the size of the metric space, $D_T = \log M$.*

An immediate consequence of the $O(\log M)$ metric distortion in Lemma D.2 is that the Wasserstein distance in the original metric space is also preserved up to a $O(\log M)$ factor in expectation. Thus, Lemma D.2 allows us to translate the problem of learning densities on an arbitrary metric space in Wasserstein distance to learning densities in Wasserstein distance on an HST. This is a useful tool since HST metrics are generally easier to work with and, as we'll see below, the Wasserstein distance is particularly well-behaved on an HST. In order to use Lemma D.2 to translate the problem of density estimation on a bounded ball in $\mathbb{R}^d$ into density estimation on an HST, one discretizes the metric, paying a small additive term.

**Corollary D.3.** *Given $\alpha > 0$, there is a probabilistic embedding $f$ of $[0, 1]^d$ into an HST such that for all $x, y \in [0, 1]^d$:*

- $d(x, y) - \alpha \leq d_T(f(x), f(y))$

- $\mathbb{E}[d_T(f(x), f(y)] \leq O(d \cdot \log \frac{1}{\alpha}) \cdot (d(x, y) + \alpha)$

The distortion is logarithmic in $\frac{1}{\alpha}$, so taking $\alpha$ to be polynomially small, one gets the distortion to be $O(d \log n)$. It is easy to see that this implies that the Wasserstein distance is preserved in both directions up to $O(d \log \frac{1}{\alpha})$, up to an $\alpha$ additive error.

A distribution $P$ on the the underlying metric space in an HST induces a function $\mathfrak{G}_P$ on the nodes of the tree where the value of a node $\nu$ is given by the weight in $P$ of the leaf nodes in the subtree rooted at $\nu$. For every level $\ell \in [D_T]$ of the tree, let $P_\ell$ be the distribution induced on the nodes at level $\ell$ where the probability of node $\nu$ is $\mathfrak{G}_P(\nu)$. Thus $P_\ell$ is a discrete distribution on a domain of size $N_\ell$, where $N_\ell$ is the number of nodes in level $\ell$ of the tree.

**Lemma D.4** (Closed form Wasserstein distance formula). *Given two distributions $P$ and $Q$ defined on an HST metric space, the Wasserstein distance between $P$ and $Q$ has the closed formula:*

$$\mathcal{W}(P, Q) = \frac{1}{2} \sum_\nu r_\nu |\mathfrak{G}_P(\nu) - \mathfrak{G}_Q(\nu)| = \sum_\ell r_\ell \mathrm{TV}(P_\ell, Q_\ell),$$

*where $r_\nu$ is the length of the edge connecting $\nu$ to its parent, and the sum is over all nodes in the tree.*

## D.2 The Target Estimation Rate

Recall the definition of our neighbourhood.

$$\mathcal{N}(P) = \{Q \in \mathcal{P} \mid D_\infty(P, Q) \leq \ln 2\}$$

We will call a node $\nu$, $\alpha$-*active node under the distribution* $P$ if the weight in $P$ of the sub-tree rooted at $\nu$ is greater than $\alpha$. Let $\gamma_P(\alpha)$ be the set of $\alpha$-active nodes under $P$ and $\gamma_{P_\ell}(\alpha)$ be the $\alpha$-active nodes at level $\ell$.

**Theorem D.5.** *Given a distribution $P$ on $[N]$, $\varepsilon > 0$, $\delta \in [0, 1]$, and $n \in \mathbb{N}$, let $\kappa = \frac{1}{10\varepsilon n} \min\{W\left(\frac{0.45\varepsilon}{\delta}\right), 0.6\}$ where $W(x)$ is the Lambert W function so $W(x)e^{W(x)} = x$, then*

$$\mathcal{R}_{\mathcal{N}, n, \varepsilon}(P) = \Omega \left( \max_\ell r_\ell \sum_{x \in [N_\ell]} \min \left\{ P_\ell(x)(1 - P_\ell(x)), \sqrt{\frac{P_\ell(x)(1 - P_\ell(x))}{n}} \right\} + \sum_{x \notin \gamma_{P_\ell}(2\kappa)} P_\ell(x) + (|\gamma_{P_\ell}(2\kappa)| - 1)\kappa \right),$$

*where the max is over all the levels of the tree.*

Note that $\kappa \approx \frac{1}{\varepsilon n} \min\{\log(1/\delta), 1\}$ so the dependence on $\varepsilon$ and $n$ in Theorem D.5 matches the upper bound in Theorem D.13. The error rate $\mathcal{R}_{\mathcal{N},n,\varepsilon}$ does indeed adapt to easy instances as we expected. The error decomposes into three components. The first component is the non-private sampling error; the error that would occur even if privacy was not required. The second component indicates that we can not estimate the value of nodes that have probability less than $1/(\varepsilon n)$. The third component is the error due to privacy on the active nodes. If $P$ is highly concentrated then we expect most nodes to either be $\frac{1}{\varepsilon n}$-active or have weight 0, so the first two terms in $\mathcal{R}_{\mathcal{N},n,\varepsilon}(P)$ are small. There should also be few active nodes, making the last term smaller as well. Conversely, if $P$ has a large region of low density then we expect a large number of inactive nodes, as well as non-zero inactive nodes that are at higher levels of the tree and hence contribute more to the final term. Thus, in distributions with high dispersion we expect the right hand side to be large.

The proof of Theorem 4.1 will involve two main steps. First, we will reduce the lower bound on the HST to a lower bound on a star metric, or equivalently estimation of a discrete distribution in TV distance. We'll then use a variant of Assouad's inequality to prove the lower bounds on estimating discrete distributions in TV distance.

### D.2.1 Reduction to Estimation in TV distance of Discrete Distributions

The key observation is that in order to estimate the distribution well in Wasserstein distance, an algorithm must estimate each level of the tree well in TV distance. Any estimate of $P$ also induces an estimate of $P_\ell$; let $\hat{P}$ be an estimate of the distribution $P$ and $\hat{P}_\ell$ be the induced estimate of the distribution at level $\ell$. Then for any distribution $P$

$$\mathcal{W}(P, \hat{P}) = \sum_{\ell \in [D_T]} r_\ell TV(P_\ell, \hat{P}_\ell).$$

The following observation ensures that our notions of instance optimality in both the Wasserstein metric and the per-level TV distance are compatible at every level $\ell$.

**Theorem D.6.** *For every level $\ell \in [D_T]$, define the neighborhood of $P_\ell$ as $\mathcal{N}_\ell : \Delta([N_\ell]) \to \mathfrak{P}(\Delta([N_\ell]))$ by $\mathcal{N}_\ell(P_\ell) = \{Q_\ell \mid D_\infty(P_\ell, Q_\ell) \leq \ln 2\}$. Then,*
$$\mathcal{R}_{\mathcal{N},n,\varepsilon}(P) \geq \max_{\ell \in [D_T]} r_\ell \cdot \mathcal{R}_{\mathcal{N}_\ell,n,\varepsilon}(P_\ell),$$

*where the error of $P$ is measured in the Wasserstein distance and $P_\ell$ is measured in the TV distance.*

Recall that $\mathcal{R}_{\mathcal{N}_\ell,n,\varepsilon}(P_\ell)$ is the optimal estimation rate with respect to $\mathcal{N}_\ell$ where the error is measured with respect to the total variation error. The proof of Theorem D.6 can be found in Appendix G.

### D.2.2 Characterizing Target Estimation Rate for Discrete Distributions

In light of Theorem D.6, we will focus on characterizing the difficulty of estimating the distribution at a single level of the tree for the remainder of this section. Since this is fundamentally a statement about estimating discrete distributions in TV distance, we will state everything in this section in terms of general discrete distributions. Let $N \in \mathbb{N}$, and let $P$ be a distribution on $[N]$. Define $\mathcal{N}(P) = \{Q \mid D_\infty(P, Q) \leq \ln 2\}$. Our goal is to give a lower bound for $\mathcal{R}_{f,n,\varepsilon}(P)$, where the metric is the TV distance.

**Theorem D.7.** *Given $\varepsilon > 0$ and $\delta \in [0,1]$, let $\kappa = \frac{1}{10\varepsilon n} \min\{W\left(\frac{0.45\varepsilon}{\delta}\right), 0.6\}$ where $W(x)$ is the Lambert W function so $W(x)e^{W(x)} = x$. Given a distribution $P$,*

$$\mathcal{R}_{\mathcal{N},n,\varepsilon}(P) = \Omega\left(\sum_{x \in [N]} \min\left\{P(x)(1-P(x)), \sqrt{\frac{P(x)(1-P(x))}{n}}\right\} + \sum_{x \notin \gamma_P(2\kappa)} P(x) + (|\gamma_P(2\kappa)| - 1)\kappa\right)$$

Theorem D.5 follows immediately from Theorem D.6 and Theorem D.7. The main tool we will use is a differentially private version of Assouad's method. This gives us a method for lower bounding the error by constructing nets of distributions that are pairwise far in the relevant metric of interest, which for us in the TV distance. The following is a slight variant on the differentially private variant of Assouad's lemma given in [ASZ21]. Rather than building a set of distributions indexed by a hypercube, we will build a set of distributions over a product of hypercubes. Since this is an extension of the version that appears in [ASZ21], we include a proof in Appendix G for completeness.

**Lemma D.8.** *[A extension of $(\varepsilon, \delta)$-DP Assouad's method [ASZ21]] Let $k_0, k_1, \cdots$ be a sequence of natural numbers such that $\sum_s k_s < \infty$, $\varepsilon > 0$ and $\delta \in [0, 1]$. Given a family of distributions $\mathcal{P} \subset \Delta(\mathcal{X})$ on a space $\mathcal{X}$, a parameter $\theta : \mathcal{P} \to \mathcal{M}$ where $\mathcal{M}$ is a metric space with metric $d$, suppose that there exists a set $\mathcal{V} \subset \mathcal{P}$ of distributions indexed by the product of hypercubes $\mathcal{E}_{k_0} \times \mathcal{E}_{k_1} \times \cdots$ where $\mathcal{E}_k := \{\pm 1\}^k$ such that for a sequence $\tau_0, \tau_1, \cdots$,*

$$\forall (u^0, u^1, \cdots), (v^0, v^1, \cdots) \in \mathcal{E}_{k_0} \times \mathcal{E}_{k_1} \times \cdots, \quad d(\theta(p_u), \theta(p_v)) \geq 2 \sum_s \tau_s \sum_{j=1}^{k_s} \chi_{u_j^s \neq v_j^s}. \quad (6)$$

*For each coordinate $s \in \mathbb{N}$, $j \in [k_s]$, consider the mixture distributions obtained by averaging over all distributions with a fixed value at the $(s, j)$th coordinate:*

$$p_{+(s,j)} = \frac{2}{|\mathcal{E}_{k_0} \times \mathcal{E}_{k_1} \times \cdots|} \sum_{u \in \mathcal{E}_{k_0} \times \mathcal{E}_{k_1} \times \cdots : u_j^s = +1} p_u, \quad p_{-(s,j)} = \frac{2}{|\mathcal{E}_{k_0} \times \mathcal{E}_{k_1} \times \cdots|} \sum_{u \in \mathcal{E}_{k_0} \times \mathcal{E}_{k_1} \times \cdots : u_j^s = -1} p_u,$$

*and let $\phi_{s,j} : \mathcal{X}^n \to \{-1, +1\}$ be a binary classifier. Then*

$$\min_{\mathcal{A} \text{ is } (\varepsilon, \delta)\text{-DP}} \max_{p \in \mathcal{V}} \mathcal{R}_{\mathcal{A}, n}(p) \geq \frac{1}{2} \sum_s \tau_s \sum_{j=1}^{k_s} \min_{\phi_{s,j} \text{ is } (\varepsilon, \delta)\text{-DP}} \left( \Pr_{X \sim p_{+(s,j)}^n} (\phi_{s,j}(X) \neq 1) + \Pr_{X \sim p_{-(s,j)}^n} (\phi_{s,j}(X) \neq -1) \right),$$

*where the min on the LHS is over all $(\varepsilon, \delta)$-DP mechanisms, and on the right hand side is over all $(\varepsilon, \delta)$-DP binary classifiers. Moreover, if for all $s \in \mathbb{N}$, $j \in [k_s]$, there exists a coupling $(X, Y)$ between $p_{+(s,j)}^n$ and $p_{-(s,j)}^n$ with $\mathbb{E}[d_{Ham}(X, Y)] \leq D_s$, then*

$$\min_{\mathcal{A} \text{ is } (\varepsilon, \delta)\text{-DP}} \max_{p \in \mathcal{V}} \mathcal{R}_{\mathcal{A}, n}(p) \geq \sum_s \frac{k_s \tau_s}{2} (0.9 e^{-10 \varepsilon D_s} - 10 D_s \delta)$$

Note that an upper bound on $TV(P_i, P_j) \leq \gamma$ implies there exists a coupling $(X, Y)$ between $P_i^n$ and $P_j^n$ such that $\mathbb{E}[d_{Ham}(X, Y)] \leq n\gamma$.

We will separately prove that each of the three terms in Theorem D.7 belong in the lower bound. Each proof will follow the same underlying structure. Given a distribution $P$, the main technical step is carefully designing a family of distributions in $\mathcal{N}(P)$ that satisfy the conditions of Lemma D.8. Lemma D.9 and Lemma D.10 give lower bounds on the noise due to privacy. Lemma D.11 gives lower bounds based on the error due to sampling.

Let

$$\kappa = \frac{1}{10\varepsilon} \min\{W\left(\frac{0.45\varepsilon}{\delta}\right), 0.6\},$$

where $W(x) \approx \ln x - (1 - o(1)) \ln \ln x$ is the Lambert W function satisfying $W(x) e^{W(x)} = x$. In both lemma proofs we will use the inequality that if $D \leq \kappa$, then

$$0.9 e^{-10 \varepsilon D} - 10 D \delta \geq e^{-10 \varepsilon D} \left(0.9 - W\left(\frac{0.45\varepsilon}{\delta}\right) e^{W\left(\frac{0.45\varepsilon}{\delta}\right)} \frac{\delta}{\varepsilon}\right) = e^{-10 \varepsilon D} \left(0.9 - \frac{0.45\varepsilon}{\delta} \frac{\delta}{\varepsilon}\right) \geq e^{-10 \varepsilon D} 0.45 \geq 0.2 \quad (7)$$

**Lemma D.9.** *Given a distribution $P$, $\varepsilon > 0$, $\delta \in [0, 1]$ and $n \in \mathbb{N}$,*

$$\mathcal{R}_{\mathcal{N}, n, \varepsilon}(P) \geq 0.1 \left( \left| \gamma_P \left(\frac{2\kappa}{n}\right) \right| - 1 \right) \frac{\kappa}{n}.$$

*Proof.* Let $L = |\gamma_P(2\kappa/n)|$ be the number of active nodes. If $L = 1$ then the RHS is 0 and so we are done. Otherwise, assume $L > 1$ and let $k = \lfloor L/2 \rfloor \geq 1$. Using the notation from Lemma D.8, let $k_0 = k$ and $k_s = 0$ for all $s > 0$. We will drop the reference to $s$ in the notation since only $s = 0$ is significant.

Pair up the active nodes to form $k$ pairs of active nodes denoted by $(a_1^+, a_1^-), \cdots, (a_k^+, a_k^-)$. Given $u \in \mathcal{E}_k$, define the distribution $p_u$ as follows: for all $a_j^b \in \gamma_P(2\kappa/n)$, $P_u(a_j^+) = P(a_j^+) + (\kappa/n)$ and $P_u(a_j^-) = P(a_j^-) - (\kappa/n)$ if $u_j = +1$ and $P_u(a_j^+) = P(a_j^+) - (\kappa/n)$ and $P_u(a_j^-) = P(a_j^-) + (\kappa/n)$

if $u_j = -1$. For all other $x$, $P_u(x) = P(x)$. It is immediate that for all $u$, $P_u \in \mathcal{N}(P)$. For any pair $u, v$, $d(\theta(p_u), \theta(p_v)) = \mathrm{TV}(p_u, p_v) = d_{Ham}(u, v)(\kappa/n)$, so that Equation (6) is satisfied with $\tau = \frac{1}{2}(\kappa/n)$. Further, given $j \in [k]$, $p_{+j}$ and $p_{-j}$ only differ on the probability of $a_j^+$ and $a_j^-$, so $D/n = \max_j \mathrm{TV}(p_{+j}, p_{-j}) = \kappa/n$ and by Equation (7), $0.9 e^{-10\varepsilon D} - 10D\delta \geq 0.2$. Noting that $k \geq (1/2)(\gamma_P(2\kappa/n) - 1)$ completes the proof. $\qquad\square$

**Lemma D.10.** *For all $\varepsilon > 0$, $\delta \in [0, 1]$, $n \in \mathbb{N}$ and distributions $P$ on $[N]$, if $\kappa < n/2$, then*

$$\mathcal{R}_{\mathcal{N}, n, \varepsilon}(P) \geq \Omega\left(\sum_{x \notin \gamma_P(2\kappa)} P(x)\right).$$

Since $\kappa \leq \frac{1}{\varepsilon n}$, the condition that $\kappa < n/2$ is a mild condition. For example, it is satisfied whenever $\varepsilon > 2/n^2$.

Similar to the proof of Lemma D.9, we are going to pair up the coordinates and move mass between the coordinates to create the distributions indexed by the product of hypercubes. Since we want all the distributions we create to be in $\mathcal{N}(P)$, we will divide the space into scales such that all elements in the same scale have approximately the same probability of occurring. We'll then move mass within these scales. For $s \in \mathbb{N}$, let $\mathcal{S}_s = \{x \in [N] \mid P(x) \in (2^{-s-1}, 2^{-s}]\}$.

*Proof.* Given $s \in \mathbb{N}$, let $\mathcal{S}'_s = \mathcal{S}_s \cap \{x \mid P(x) \leq 2\kappa/n\}$ and $d_s = |\mathcal{S}'_s|$.

Let us first consider the case that there exists a scale $s^*$ with $d_{s^*} = 1$ and $P(x^*) \geq \frac{1}{8} \sum_{x \notin \gamma_P(2\kappa)} P(x)$ where $x^*$ is the element in $\mathcal{S}'_{s^*}$. Define $P'$ by $P'(x^*) = (1/2)P(x^*)$ and for all $x \neq x^*$, $P'(x) = \frac{1-(1/2)P(x^*)}{1-P(x^*)} P(x)$. Since $(1/2)P(x^*) \leq 2\kappa/n \leq 1/2$, $P' \in \mathcal{N}(P)$. In this case we will use Lemma D.8 with $k_0 = 1$, $k_s = 0$ otherwise, and $\mathcal{E}_{k_0}$ corresponds to the set of distributions $\{P, P'\}$. Then noting that $\mathrm{TV}(P, P') = (1/2)P(x^*)$ and using eqn (7) we have that $\tau = \frac{1}{4}P(x^*)$ and $D = (1/2)P(x^*)n \leq \kappa$ so that $\mathcal{R}_{\mathcal{N}, n, \varepsilon}(P) \geq (1/8)P(x^*)(0.2) = \Omega\left(\sum_{x \notin \gamma_P(2\kappa)} P(x)\right)$ and we are done.

Next suppose that for all scales $s$ such that $d_s = 1$ we have $P(x^*) < \frac{1}{8} \sum_{x \notin \gamma_P(2\kappa)} P(x)$. Let $s^*$ be the smallest $s$ such that $d_s = 1$. Since the scales $2^{-s-1}$ are geometrically decreasing,

$$\sum_{s:d_s=1} \sum_{x \in \mathcal{S}_s \cap \{x \mid P(x) \leq 2\kappa/n\}} P(x) \leq 2 \sum_{s:d_s=1} 2^{-s-1} \leq 4 \cdot 2^{-s^*-1} \leq \frac{1}{2} \sum_{x \notin \gamma_P(2\kappa)} P(x).$$

It follows that $\sum_{s:d_s>1} \sum_{x \in \mathcal{S}_s \cap \{x \mid P(x) < 2\kappa/n\}} P(x) \geq (1/2) \sum_{x \notin \gamma_P(2\kappa)} P(x)$. Further,

$$\sum_{x \notin \gamma_P(2\kappa)} P(x) \leq 2 \sum_{s:d_s>1} \sum_{x \in \mathcal{S}_s \cap \{x \mid P(x) < 2\kappa/n\}} P(x) \leq 4 \sum_{s:d_s>1} 2^{-s-1} d_s \leq 16 \sum_{s:d_s>1} 2^{-s-1} \lfloor d_s/2 \rfloor.$$

Thus we can (up to constants) ignore scales such that $d_s \leq 1$ and assume that $d_s$ is even for all scales.

Let $k_s = \lfloor d_s/2 \rfloor$. Now, within each scale $\mathcal{S}_s$, pair the elements to form $k_s$ distinct pairs $(a_{s,j}^+, a_{s,j}^-)$. Given $(u^0, u^1, \cdots) \in \mathcal{E}_{k_0} \times \mathcal{E}_{k_1} \times \cdots$, define $p_u$ by $p_u(a_{s,j}^+) = p(a_{s,j}) + 2^{-s-2}$ and $p_u(a_{s,j}^-) = p(a_{s,j}) - 2^{-s-2}$ if $u_j^s = +1$ and $p_u(a_{s,j}^+) = p(a_{s,j}) - 2^{-s-2}$ and $p_u(a_{s,j}^-) = p(a_{s,j}) + 2^{-s-2}$ if $u_j^s = -1$. For all other elements, $p_u(x) = p(x)$. Then, it is easy to see that for all $u$, $p_u \in \mathcal{N}(P)$. Further, using notation from Lemma D.8, Equation (6) is satisfied with $\tau_s = \frac{1}{2} 2^{-s-2}$ since $d(\theta(p_u), \theta(p_v)) = \mathrm{TV}(p_u, p_v) = 2\sum_s \tau_s d_{Ham}(u^s, v^s)$ and $D_s/n = \max_j \mathrm{TV}(p_{+(s,j)}, p_{-(s,j)}) = 2^{-s-2}$, which is less than $\kappa$ whenever $k_s > 0$. By eqn (7), $0.9 e^{-10\varepsilon D_s} - 10 D_s \delta \geq 0.2$ whenever $k_s > 0$ so by Lemma D.8 we have

$$\mathcal{R}_{\mathcal{N}, n, \varepsilon}(P) \geq \sum_s \frac{1}{2} 2^{-s-2} k_s (0.2) = \frac{0.2}{4} \sum_{s:d_s>1} 2^{-s-1} \lfloor d_s/2 \rfloor \geq \frac{0.2}{4 \times 16} \sum_{x \notin \gamma_P(2\kappa)} P(x)$$

which completes the proof. $\qquad\square$

Next we lower bound the statistical term.

**Lemma D.11.** *For all $n \in \mathbb{N}$, $\varepsilon > 0$, $\delta \in [0,1]$ and distributions $P$, if $n \geq 2$ and $\varepsilon > 2/n$, then*

$$\mathcal{R}_{\mathcal{N},n,\varepsilon}(P) \geq \mathcal{R}_{\mathcal{N},n}(P) \geq \Omega\left(\sum_{x \in [N]} \min\left\{P(x)(1-P(x)), \sqrt{\frac{P(x)(1-P(x))}{n}}\right\}\right).$$

To streamline the notation, we will use $L(x)$ to denote $\min\left\{x(1-x), \sqrt{\frac{x(1-x)}{n}}\right\}$. In order to prove Lemma D.11, we will need the following standard result from the statistics literature which allows us to lower bound the performance of any simple classifier distinguishing two distributions $P$ and $Q$ by the KL divergence between $P$ and $Q$. We give a specific result for distinguishing Bernoulli random variables since we'll use this in the proof of Lemma D.11.

**Lemma D.12.** *Given any pair of distributions $P$ and $Q$ on the same domain,*

$$\min_\phi \left( \Pr_{X \sim P^n} (\phi(X) = 1) + \Pr_{X \sim Q^n} (\phi(X) = -1) \right) \geq \frac{1}{2}(1 - \sqrt{n\mathrm{KL}(P,Q)}),$$

*where the minimum is over all binary classifiers. In particular, if $P = $ `Bernoulli`$(p - \alpha)$ and $Q = $ `Bernoulli`$(p + \alpha)$ where $0 \leq \alpha \leq \frac{1}{2}L(p)$ then*

$$\min_\phi \left( \Pr_{X \sim P^n} (\phi(X) = 1) + \Pr_{X \sim Q^n} (\phi(X) = -1) \right) \geq 1/4,$$

*where again the minimum is over all binary classifiers.*

The proof of Lemma D.12 can be found in Appendix G

*Proof of Lemma D.11.* As in the proof of Lemma D.10, first suppose there exists a scale $s^*$ with $d_{s^*} = 1$ and there exists $x^* \in \mathcal{S}_{s^*}$ such that

$$\frac{1}{2}L(P(x^*)) \geq \frac{1}{60} \sum_{x \in [N]} L(P(x)).$$

Then define a distribution $P'$ by $P'(x^*) = P(x^*) - \frac{1}{2}L(P(x^*))$ and for all $x \neq x^*$, $P'(x) = \frac{1-P(x^*)+\frac{1}{2}L(P(x^*))}{1-P(x^*)}P(x)$. Then $P' \in \mathcal{N}(P)$ since $\frac{1}{2}L(P(x^*)) < \frac{1}{2}\min\{P(x^*), (1-P(x^*))\}$. Then we will use Lemma D.8 with $k_0 = 1$ and $k_s = 0$ for $s > 0$, and $\mathcal{E}_{k_0}$ corresponds to $\{P, P'\}$. Now,

$$\mathrm{KL}(P',P) = (P(x^*) - \frac{1}{2}L(P(x^*))) \ln \frac{P(x^*) - \frac{1}{2}L(P(x^*))}{P(x^*)} + (1 - P(x^*) + \frac{1}{2}L(P(x^*))) \ln \frac{1 - P(x^*) + \frac{1}{2}L(P(x^*))}{1 - P(x^*)}$$

$$\leq \frac{1}{4n}$$

(for more detail on the proof of this inequality see the proof of Lemma D.12) so

$$\min_\phi \left( \Pr_{X \sim P^n} (\phi(X) = 1) + \Pr_{X \sim P'^n} (\phi(X) = -1) \right) \geq \frac{1}{2}(1 - \sqrt{n\mathrm{KL}(P,P')}) \geq 1/4$$

and $\tau_0 = \mathrm{TV}(P,P') = \frac{1}{2}L(P(x^*))$. Thus by Lemma D.8,

$$\mathcal{R}_{\mathcal{N},n,\varepsilon}(P) \geq \mathcal{R}_{\mathcal{N},n}(P) \geq \frac{1}{2}L(P(x^*))\frac{1}{4} \geq \frac{1}{480}\sum_{x \in [N]}\frac{1}{2}L(P(x)),$$

and we are done.

On the other hand, suppose that for all scales $s$ such that $d_s = 1$ we have

$$L(P(x_s)) \leq \frac{1}{30}\sum_{x \in [N]} L(P(x)),$$

where $\mathcal{S}_s = \{x_s\}$. As in the proof of Lemma D.10, we will argue that we can ignore any singleton scales, and assume that $d_s$ is even for all scales. Let $s^* = \min\{s > 0 \mid d_s = 1\}$ so

$$
\sum_{s:d_s=1} L(P(x_s)) \leq \chi_{d_0=1} L(P(x_0)) + \sum_{s>0:d_s=1} \min\left\{2^{-s}, \sqrt{\frac{2^{-s}}{n}}\right\}
$$

$$
\leq \chi_{d_0=1} L(P(x_0)) + (2+\sqrt{2})\min\left\{2^{-s^*}, \sqrt{\frac{2^{-s^*}}{n}}\right\}
$$

$$
\leq \chi_{d_0=1} L(P(x_0)) + 2(2+\sqrt{2})\min\left\{P(x_{s^*}), \sqrt{\frac{P(x_{s^*})}{n}}\right\}
$$

$$
\leq \chi_{d_0=1} L(P(x_0)) + 4(2+\sqrt{2})L(P(x_{s^*}))
$$

$$
\leq \frac{1 + 4(2+\sqrt{2})}{30} \sum_{x\in[N]} L(P(x)).
$$

Therefore, $\sum_{s:d_s>1}\sum_{x\in\mathcal{S}_s} L(P(x)) \geq (1/2)\sum_{x\in[N]} L(P(x))$ and so

$$
\sum_s L(2^{-s-1})\lfloor d_s/2\rfloor \geq \sum_{s:d_s>1} L(2^{-s-1})\frac{1}{3}d_s
$$

$$
\geq \sum_{s:d_s>1}\sum_{x\in\mathcal{S}_s} L(2^{-s-1})\frac{1}{3}
$$

$$
\geq \frac{1}{3\sqrt{2}} \sum_{x\in[N]} L(P(x)) \tag{8}
$$

where the first inequality follows from $\lfloor d_s/2\rfloor \geq (1/3)d_s$ whenever $d_s > 1$, and the second follows because $2^{-s-1} \leq P(x) \leq 1/2$ for all $x \in \mathcal{S}_s$ such that $d_s > 1$.

Assume that $d_s$ is even for all $s$. Within each scale $\mathcal{S}_s$, pair the elements to form $k_s = d_s/2$ distinct pairs $(a_{s,j}^+, a_{s,j}^-)$ per scale. For all $s \in \mathbb{N}$, let $\alpha_s = \frac{1}{2}L(2^{-s-1})$, and note that for all $x \in \mathcal{S}_s$ and $s > 0$, $\alpha_s \leq \frac{1}{2}L(P(x))$. Given $(u^0, u^1, \cdots) \in \mathcal{E}_{k_0}\times\mathcal{E}_{k_1}\times\cdots$, define $p_u$ by $p_u(a_{s,j}^+) = p(a_{s,j}^+) + \alpha_s$ and $p_u(a_{s,j}^-) = p(a_{s,j}^-) - \alpha_s$ if $u_j^s = +1$ and $p_u(a_{s,j}^+) = p(a_{s,j}^+) - \alpha_s$ and $p_u(a_{s,j}^-) = p(a_{s,j}^-) + \alpha_s$ if $u_j^s = -1$. For all other elements, $p_u(x) = p(x)$. Then, for all $u$, $p_u \in \mathcal{N}(P)$. Further, using notation from Lemma D.8, we have $\tau_s = \alpha_s$. Also, for any $(s,j)$, $p_{+(s,j)}$ and $p_{-(s,j)}$ only differ on $a_{s,j}^+$ and $a_{s,j}^-$ where $p_{+(s,j)}(a_{s,j}^+) = P(a_{s,j}^+) + \alpha_s$ and $p_{-(s,j)}(a_{s,j}^+) = P(a_{s,j}^+) - \alpha_s$. Therefore, by Lemma D.12, and the post-processing inequality,

$$
\min_\phi \left(\Pr_{X\sim p_{+(s,j)}^n}(\phi(X)=1) + \Pr_{X\sim p_{-(s,j)}^n}(\phi(X)=-1)\right) \geq 1/4.
$$

Lemma D.8 then implies the result. $\qquad\square$

Theorem D.7 follows immediately from Lemma D.9, Lemma D.10 and Lemma D.11.

## D.3 An $\varepsilon$-DP Distribution Estimation Algorithm

Now, let us return to HSTs and designing an estimation algorithm that achieves the target estimation rate, up to logarithmic factors. As in the one-dimensional setting, we want to restrict to only privately estimating the density at a small number ($\approx \varepsilon n$) of points. While we could try to mimic the one-dimensional solution by privately estimating a solution to the $\varepsilon n$-median problem, it's not clear how to prove that such an approach is instance-optimal. It turns out that a simpler solution more amenable to analysis will suffice. Our algorithm has two stages; first we attempt to find the set of $\frac{\log(1/\delta)}{\varepsilon n}$-active nodes, then we estimate the weight of these active nodes. Since these nodes have weight greater than $\frac{\log(1/\delta)}{\varepsilon n}$, we can privately estimate them to within constant multiplicative error.

Let $\mathcal{X}$ be the underlying metric space so $P \in \Delta(\mathcal{X})$. For any set $S$ of nodes and a function $F$ defined on the nodes, define the function $F|_S$ as $F|_S(\nu) = F(\nu)$ if $\nu \in S$ and $F|_S(\nu) = 0$ otherwise. Given two functions $F$ and $G$ defined on the nodes, we define

$$\mathfrak{W}(F, G) = \sum_\nu r_\nu |F(\nu) - G(\nu)|,$$

where $r_\nu$ is the length of the edge connecting $\nu$ to its parent, and the sum is over all nodes in the tree. So by Lemma D.4, $\mathcal{W}(P, Q) = \mathfrak{W}(\mathfrak{G}_P, \mathfrak{G}_Q)$. Note that $\mathfrak{W}$ satisfies the triangle inequality.

---

**Algorithm 1** `PrivDensityEstTree`

---

1: **Input:** $D \in \mathcal{X}^n, \varepsilon$
2: $\widehat{\mathfrak{G}_P} = \texttt{EmpDist}(D)$ ▷ Compute empirical distribution.
3: $\hat{\gamma}_\varepsilon = \texttt{LocateActiveNodes}(\widehat{\mathfrak{G}_P}; \varepsilon)$ ▷ Privately approximate set of active nodes.
4: Define $\widetilde{\mathfrak{G}_{\hat{P}_n, \hat{\gamma}_\varepsilon}}$ by $\widetilde{\mathfrak{G}_{\hat{P}_n, \hat{\gamma}_\varepsilon}}(x) = \begin{cases} 0 & \text{if } x \notin \hat{\gamma}_\varepsilon \\ \widehat{\mathfrak{G}_P}(x) + \mathsf{Lap}(\frac{1}{\varepsilon n})) & \text{otherwise.} \end{cases}$ ▷ Approximate densities.
5: $\hat{P}_{n,\varepsilon} = \texttt{Projection}(\widetilde{\mathfrak{G}_{\hat{P}_n, \hat{\gamma}_\varepsilon}})$ ▷ Project noisy densities onto space of distributions.
6: **return** $\hat{P}_{n,\varepsilon}$

---

A high-level outline of the proposed algorithm is given in Algorithm 1. Now, we state the main theorem of this section.

**Theorem D.13.** *Given any $\varepsilon > 0$, `PrivDensityEstTree` is $(D_T + 1)\varepsilon$-DP. Given a distribution P, with probability $1 - (D_T \log n + 4D_T \varepsilon n)\beta$,*

$$\mathcal{W}(P, \hat{P}_\varepsilon) = O\Bigg( \sum_{\ell \in [D_T]} \sum_{x \in [N_\ell]} \min\left\{ P_\ell(x), 1 - P_\ell(x)\sqrt{\frac{P_\ell(x)\log(n/\beta)}{n}} \right\}$$
$$+ \sum_{\nu \notin \gamma_P\left(\max\{\frac{2}{\varepsilon n} + 2\frac{\log(2/\beta)}{\varepsilon n}, \frac{192\log(n/\beta)}{\varepsilon n}\}\right)} \mathfrak{G}_P(\nu) + \frac{|\gamma_{P_\ell}\left(\frac{1}{2\varepsilon n}\right) - 1|\log(1/\beta)}{\varepsilon n} \Bigg)$$

This bound has the same three terms as our lower bound on $\mathcal{R}_{\mathcal{N},n,\varepsilon}$ in Theorem D.5 corresponding again to the empirical error (the error inherent even in the absence of a privacy requirement), the error from the private algorithm not being able to estimate the probability of events that occur with probability less than $\approx \log(1/\delta)/\varepsilon n$, and the error due to the noise added to the active nodes. The maximum over the levels that appeared in the lower bound is replaced with a sum over the levels in the upper bound, so, up to logarithmic factors, the upper bound is within a factor of $D_T$ of the lower bound. Since we can not hope to locate the set of $\log(1/\delta)/(\varepsilon n)$-active nodes exactly with a private algorithm, we find a set $\hat{\gamma}_n$ that is guaranteed to satisfy

$$\gamma_P\left(\max\left\{ \frac{2}{\varepsilon n} + 2\frac{\log(2/\beta)}{n}, \frac{192\log(n/\beta)}{n} \right\}\right) \subset \hat{\gamma}_\varepsilon \subset \gamma_P\left(\frac{1}{2\varepsilon n}\right).$$

Note that $\max\left\{\frac{2}{\varepsilon n} + 2\frac{\log(2/\beta)}{n}, \frac{192\log(n/\beta)}{n}\right\} \leq \frac{C\log(n/\beta)}{\varepsilon n}$ so the error introduced here by not estimating $\gamma_P\left(\frac{1}{\varepsilon n}\right)$ perfectly is at most a logarithmic multiplicative factor.

The first step of our algorithm is to estimate the empirical distribution. We use a truncated version of the standard empirical distribution. This allows us to achieve an error rate of $\min\{P(x), \sqrt{P(x)/n}\}$ even when $P(x)$ is small.

The proof of the following lemma is contained in Appendix G.

**Lemma D.14.** *For any distribution P, if $\log(n/\beta) > 1$ then with probability $1 - 3D_T\beta$,*

$$\mathfrak{W}(\widehat{\mathfrak{G}_P}, \mathfrak{G}_P) \leq \sum_{\ell \in [D_T]} \sum_{x \in [N_\ell]} \min\left\{ P_\ell(x)(1 - P_\ell(x)), 4\sqrt{3\frac{P_\ell(x)(1 - P_\ell(x))\log(n/\beta)}{n}} \right\}$$

---

**Algorithm 2** `EmpDist`

---

1: **Input:** $D \in \mathcal{X}^n, A$
2: Let $\hat{P}_n$ be the empirical distribution.
3: **for all** node $\nu$ **do**
4: $\quad \widehat{\mathfrak{G}_P}(\nu) = \begin{cases} 0 & \mathfrak{G}_{\hat{P}_n}(\nu) < \frac{\sqrt{\log(n/\beta)}}{n} \\ 1 & \mathfrak{G}_{\hat{P}_n}(\nu) > 1 - \frac{\sqrt{\log n/\beta)}}{n} \\ \mathfrak{G}_{\hat{P}_n}(\nu) & \text{otherwise} \end{cases}$

---

---

**Algorithm 3** `LocateActiveNodes`

---

1: **Input:** $\widehat{\mathfrak{G}_P}, \varepsilon$
2: Let $\ell = 0$ and $\hat{\gamma}_{\varepsilon,0} = \{\nu\}$ where $\nu$ is the root node.
3: **while** $\hat{\gamma}_{\varepsilon,\ell} \neq \varnothing$ and $\ell < D_T$ **do**
4: $\quad \hat{\gamma}_{\varepsilon,\ell+1} = \varnothing$
5: $\quad$ **for all** $\nu \in \hat{\gamma}_{\varepsilon,\ell}$ **do**
6: $\quad\quad$ **for all** children $\nu'$ of $\nu$ **do**
7: $\quad\quad\quad$ **if** $\widehat{\mathfrak{G}_P}(\nu') + \mathsf{Lap}(\frac{1}{\varepsilon n}) > 2\kappa + \frac{\log(2/\beta)}{\varepsilon n}$ **then**
8: $\quad\quad\quad\quad \hat{\gamma}_{\varepsilon,\ell+1} = \hat{\gamma}_{\varepsilon,\ell+1} + \{\nu'\}$
9: $\quad \ell = \ell + 1$
10: **return** $\cup\hat{\gamma}_{\varepsilon,\ell}$

---

The goal of Algorithm 3 is to estimate the set of $1/(\varepsilon n)$-active nodes.

The next lemma allows us to bound how close to the goal we get. The proof is contained in Appendix G.

**Lemma D.15.** *Let $\hat{\gamma}_\varepsilon$ be the set of active nodes found in Algorithm 1. Then with probability $1 - D_T(\log n + 4\varepsilon n)\beta$,*

$$\gamma_P\left(\max\left\{\frac{2}{\varepsilon n} + 4\frac{\log(2/\beta)}{\varepsilon n}, \frac{192\log(n/\beta)}{n}\right\}\right) \subset \hat{\gamma}_\varepsilon \subset \gamma_P\left(\frac{1}{2\varepsilon n}\right).$$

We also prove the following lemma relating the error due to estimating the active nodes to a quantity depending on the true active nodes.

**Lemma D.16.** *If $\gamma_P\left(\max\{\frac{2}{\varepsilon n} + 4\frac{\log(2/\beta)}{\varepsilon n}, \frac{192\log(n/\beta)}{n}\}\right) \subset \hat{\gamma}_\varepsilon$ then*

$$\mathfrak{W}(\widehat{\mathfrak{G}_P}, \widehat{\mathfrak{G}_P}|_{\hat{\gamma}_\varepsilon}) \leq \mathfrak{W}(\mathfrak{G}_P, \widehat{\mathfrak{G}_P}) + \mathfrak{W}(\mathfrak{G}_P, \mathfrak{G}_P|_{\gamma_P(\max\{\frac{2}{\varepsilon n} + 4\frac{\log(2/\beta)}{\varepsilon n}, \frac{192\log(n/\beta)}{n}\})})$$

The key component of this proof is that any discrepancy between the weight of the nodes on $P$ and that assigned by $\widehat{\mathfrak{G}_P}$ was already paid for in $\mathcal{W}(P, \widehat{\mathfrak{G}_P})$. The final step in Algorithm 1 is to project the noisy function $\widetilde{\mathfrak{G}_{\hat{P}_n, \hat{\gamma}_\varepsilon}}$ into the space of distributions on the underlying metric space. We'd like to do this in a way that preserves, up to a constant, the $\mathfrak{W}$ distance between $P$ and $\widetilde{\mathfrak{G}_{\hat{P}_n, \hat{\gamma}_\varepsilon}}$. We will do this iteratively starting from the root node, by ensuring that the sum of each node's children add up to it's assigned value. Since we know the root node has value 1, this results in a valid distribution. We start from the top of the tree since errors in higher nodes of the contribute more to the Wasserstein distance. While errors in higher nodes of the tree propagate can propagate to lower levels, the predominant influence on the overall error is retained at the top level due to the geometric nature of the edge weights.

**Lemma D.17.** *For any real-valued function $\mathfrak{G}$ on the nodes of the HST such that $\mathfrak{G}(\nu_0) = 1$ where $\nu_0$ is the root node and given any distribution $P$,*

$$\mathcal{W}(P, \texttt{Projection}(\mathfrak{G})) \leq 4\mathfrak{W}(\mathfrak{G}_P, \mathfrak{G}).$$

Combining the above lemmas appropriately gives the proof of Theorem D.13 (see Appendix G).

---

**Algorithm 4** Projection

1: **Input:** $\mathfrak{G}$, a real-valued function on the nodes of the HST such that $\mathfrak{G}(\nu_0) = 1$ where $\nu_0$ is the root node.
2: $\bar{\mathfrak{G}} = \mathfrak{G}$
3: **for** $\ell = 0 : D_T - 1$ **do**
4:     **for all** nodes $\nu$ at level $\ell$ **do**
5:         Let $A_\nu = \sum \mathfrak{G}(\nu')$ where the sum is over the children of $\nu$.
6:         Let $d_\nu$ be the number of children of $\nu$
7:         **if** $A_\nu = 0$ **then**
8:             **for all** children $\nu'$ of $\nu$ **do**
9:                 $\bar{\mathfrak{G}}(\nu') = \frac{1}{d_\nu} \bar{\mathfrak{G}}(\nu)$
10:         **else**
11:             **for all** children $\nu'$ of $\nu$ **do**
12:                 $\bar{\mathfrak{G}}(\nu') = \frac{\mathfrak{G}(\nu)}{A_\nu} \mathfrak{G}(\nu')$
13: **return** $\bar{\mathfrak{G}}$

---

*Proof of Theorem D.13.* The privacy follows from the fact that each user contributes to at most $D_T$ queries in LocateActiveNodes and at most one coordinate in the computation of $\widetilde{\mathfrak{G}_{\hat{P}_n, \hat{\gamma}_n}}$ in line 4 in PrivDensityEstTree.

For the utility, we will consider each level $\ell$ individually. First suppose that $|\gamma_{P_\ell} (1/(2\varepsilon n))| > 1$.

$$
\begin{aligned}
\mathcal{W}(P_\ell, (\hat{P}_\varepsilon)_\ell) &\leq 2\mathfrak{W}((\mathfrak{G}_P)_\ell, (\widetilde{\mathfrak{G}_{\hat{P}_n, \hat{\gamma}_\varepsilon}})_\ell) \\
&\leq 2\left( \mathfrak{W}((\mathfrak{G}_P)_\ell, (\widehat{\mathfrak{G}_P})_\ell) + \mathfrak{W}((\widehat{\mathfrak{G}_P})_\ell, (\widehat{\mathfrak{G}_P}|\hat{\gamma}_\varepsilon)_\ell) + \mathfrak{W}((\widehat{\mathfrak{G}_P}|\hat{\gamma}_\varepsilon)_\ell, (\widetilde{\mathfrak{G}_{\hat{P}_n, \hat{\gamma}_\varepsilon}})_\ell) \right) \quad (9) \\
&\leq 2\left( 2\mathfrak{W}((\mathfrak{G}_P)_\ell, (\widehat{\mathfrak{G}_P})_\ell) + \mathfrak{W}((\mathfrak{G}_P)_\ell, (\mathfrak{G}_P|_{\gamma_P\left(\max\{\frac{2}{\varepsilon n} + 2\frac{\log(2/\beta)}{n}, \frac{\log(n/\beta)}{n}\}\right)})_\ell) + \mathfrak{W}((\widehat{\mathfrak{G}_P}|\hat{\gamma}_\varepsilon)_\ell, (\widetilde{\mathfrak{G}_{\hat{P}_n, \hat{\gamma}_\varepsilon}})_\ell) \right)
\end{aligned}
$$

where the first inequality follow from Lemma D.17, the second inequality follows from the triangle inequality and Lemma D.4, and the third follows from Lemma D.16 and Lemma D.15. Finally,

$$
\mathfrak{W}((\widehat{\mathfrak{G}_P}|\hat{\gamma}_\varepsilon)_\ell, (\widetilde{\mathfrak{G}_{\hat{P}_n, \hat{\gamma}_\varepsilon}})_\ell) \leq \sum_{\nu \in \gamma_P\left(\frac{1}{2\varepsilon n}\right)} r_\nu |\mathsf{Lap}(\frac{1}{\varepsilon n})| \leq \frac{1}{2} \sum_{\nu \in \gamma_P\left(\frac{1}{2\varepsilon n}\right)} r_\nu |\mathsf{Lap}(\frac{1}{\varepsilon n})|
$$

The final statement then follows from Lemma D.14 and basic concentration bounds on the Laplacian distribution.

If $|\gamma_{P_\ell} (1/(2\varepsilon n))| = 1$, then the proof goes through for all except the final term related to the noise due to privacy. We consider two cases. Let $x \in \gamma_{P_\ell} (1/(2\varepsilon n))$. First suppose that $P_\ell(x) > 1 - \frac{1}{2\varepsilon n}$ then no node that is in a level above $x$, but is not a direct ancestor of $x$ is in $\gamma_{P_\ell} (1/(2\varepsilon n))$. Therefore, since the projection algorithm is top-down, $(\hat{P}_{n,\varepsilon})_\ell$ will be concentrated on $x$. Therefore, the error of level $\ell$ is simply $(1 - P(x))$, which can be charged to the first term plus the sum of the weight of the inactive nodes, which is in the second term. Next, suppose that $P_\ell(x) < 1 - \frac{1}{2\varepsilon n}$ then sum of the inactive nodes (in term two) dominates the error due to adding noise to $P(x)$ □

## E  Instance Optimal Density Estimation on $\mathbb{R}$ in Wasserstein distance

Let us now consider the setting of estimating distributions $P$ on $\mathcal{X} = \mathbb{R}$. In this setting, the target estimation rate is that of an algorithm that knows that the distribution is either $P$ or $Q_P$ for a distribution $Q_P$ such that $D_\infty(P, Q_P) \leq \ln 2$. This definition of instance-optimality strengthens that corresponding to the so-called *hardest-one dimensional subproblem* [DL91], since this is a harder estimation rate to achieve. A formal description of the target estimation rate is given in Appendix B.1 and Appendix B.3. In Appendix E.1, we lower bound this estimation rate using hypothesis testing techniques. Then, in Appendix E.2, we give an algorithm that up to polylogarithmic factors, uniformly achieves the lower bound, and hence approximately achieves the instance-optimal estimation rate. Our instance optimality results apply to all continuous distributions in a bounded interval with

density functions (though it is likely that they apply more generally). All omitted proofs can be found in Appendix J.

## E.1 General Lower Bound

To state the main theorem in this section, we will introduce some notation. We start by defining the restriction of a distribution.

**Definition E.1.** *For any distribution $P$ over $\mathbb{R}$ with a density function, the restriction $P|_{u,v}$ of $P$ with respect to $u \leq v \in \mathbb{R}$ is defined as the distribution with the following CDF function F':*

$$F'_{P_{u,v}}(t) = \begin{cases} 0 & t < u \\ F_P(x) & u \leq t < v \\ 1 & t \geq v \end{cases}$$

*If $u = v$, then $F'$ is a step function that goes to $1$ at that point and is $0$ prior to that point.*

Also recall the following definition of quantiles.

**Definition E.2.** *For $0 < \alpha \leq 1$, the $\alpha$-quantile of a distribution $P$ over $\mathbb{R}$ is defined as follows:*

$$q_\alpha(P) = \arg \min_t \{ \Pr_{y \sim P}(y \leq t) \geq \alpha \}.$$

When the distribution $P$ is clear from context, we will sometimes abuse notation and use $q_\alpha$ when we mean $q_\alpha(P)$. The main theorem we will prove in this section is the following:

**Theorem E.3.** *There exists a constant $C$ such that given a continuous distribution $P$ on $\mathbb{R}$ with bounded expectation and $\varepsilon \in (0,1], n \in \mathbb{N}$,*

$$\mathcal{R}_{loc,n,\varepsilon}(P) = \Omega\left( \frac{1}{\varepsilon n} \left( q_{1 - \frac{1}{C\varepsilon n}} - q_{\frac{1}{C\varepsilon n}} \right) + \mathcal{W}(P, P|_{q_{\frac{1}{C\varepsilon n}}, q_{1 - \frac{1}{C\varepsilon n}}}) \right.$$

$$\left. + \frac{1}{\sqrt{\log n}} \mathbb{E}\left[ \mathcal{W}(P|_{q_{\frac{1}{C\varepsilon n}}, q_{1 - \frac{1}{C\varepsilon n}}}, \hat{P}_n|_{q_{\frac{1}{C\varepsilon n}}, q_{1 - \frac{1}{C\varepsilon n}}}) \right] \right),$$

*where $\hat{P}_n$ is the empirical distribution on $n$ samples drawn independently from $P$.*

The same result can be extended to $(\varepsilon, \delta)$-DP algorithms as well for $\delta = o(\frac{1}{n})$

We discuss each of the terms in turn. Note that the final term is related to the expected Wasserstein distance between the empirical distribution and the true distribution. There is now a long line of work characterizing this quantity in terms of the distribution (See Section C), but essentially, if the distribution is more concentrated, this term is smaller. The first term is a very particular inter-quantile distance that is also much smaller for concentrated distributions, and can be large for relatively dispersed distributions. The second term characterizes the length of the tails of the distribution— longer tails make this Wasserstein distance larger. Overall, this rate is significantly lower for more concentrated distributions with small support, and relatively large for more dispersed distributions. We prove this theorem over the following couple of sections; in Section E.1.1 we characterize the cost of private instance optimality, and in Section E.1.2 we characterize the cost of achieving instance optimality without privacy (this non-private characterization is also new to our work, to the best of our knowledge). Combining the theorems in those sections gives the above result.

### E.1.1 The Privacy Term

The main theorem we will prove in this section is the following.

**Theorem E.4.** *Fix $\varepsilon \in (0,1]$, $n \in \mathbb{N}$. For all distributions $P$ over $\mathbb{R}$ that have a density function and finite expectation, there exists another distribution $Q''$ such that $D_\infty(P,Q) \leq 2$, that is indistinguishable from $P$ given $O(n)$ samples such that for all $\varepsilon$-DP algorithms $A : \mathbb{R}^n \to \Delta(\mathbb{R})$, with probability at least $0.25$ over the draws $\mathbf{x} \sim P^n$, $\mathbf{x}' \sim Q''^n$, the following holds for some constant $C$.*

$$\max(\mathcal{W}(P, A(\mathbf{x})), \mathcal{W}(Q'', A(\mathbf{x}'))) \geq \frac{1}{4C\varepsilon n} \left( q_{1 - \frac{1}{C\varepsilon n}} - q_{\frac{1}{C\varepsilon n}} \right) + \frac{1}{4} \mathcal{W}(P, P|_{q_{\frac{1}{C\varepsilon n}}, q_{1 - \frac{1}{C\varepsilon n}}}).$$

We start with some notation. For any distribution $P$ with a density, let $f_P$ denote its density function. Throughout this section, we will use $q_\alpha$ to represent the $\alpha$-quantile of distribution $P$. Let $L(P)$ be the 'starting point' of distribution $P$ (defined as $\inf_{t \in \mathbb{R}}\{t : F_P(t) > 0\}$ if the infimum exists, and $-\infty$ otherwise.

Next, we describe some results on differentially private testing that we will use. We say that a testing algorithm $A_{test}$ distinguishes two distributions $P$ and $Q$ with $n$ samples, if given the promise that a dataset of size $n$ is drawn from either $P^n$ or $Q^n$, with probability at least $\frac{2}{3}$, it outputs $P$ if the dataset was drawn from $P^n$ and $Q$ if it was drawn from $Q^n$. We now state a theorem lower bounding the sample complexity of differentially private hypothesis testing.

**Theorem E.5** ([CKM$^+$19, Theorem 1.2]). *Fix $n \in \mathbb{N}, \varepsilon > 0$. For every pair of distributions $P, Q$ over $\mathbb{R}$, if there exists an $\varepsilon$-DP testing algorithm[4] $A_{test}$ that distinguishes $P$ and $Q$ with $n$ samples, then*

$$n = \Omega\left(\frac{1}{\varepsilon\tau(P,Q) + (1 - \tau(P,Q))H^2(P',Q')}\right),$$

*where*

$$\tau(P,Q) = \max\left\{\int_{\mathbb{R}} \max\{e^\varepsilon f_P(t) - f_Q(t), 0\}dt, \int_{\mathbb{R}} \max\{e^\varepsilon f_Q(t) - f_P(t), 0\}dt\right\},$$

*and $H^2(\cdot, \cdot)$ is the squared Hellinger distance between $P' = \frac{\min(e^\varepsilon Q, P)}{1 - \tau(P,Q)}$, and $Q' = \frac{\min(e^{\varepsilon'} P, Q)}{1 - \tau(P,Q)}$, where $0 \le \varepsilon' \le \varepsilon$ is such that if $\tau(P,Q) = \int_{\mathbb{R}} \max\{f_P(t) - e^\varepsilon f_Q(t), 0\}dt$, then $\varepsilon'$ is the maximum value such that*

$$\tau(P,Q) = \int_{\mathbb{R}} \max\{f_Q(t) - e^{\varepsilon'} f_P(t), 0\}dt,$$

*else $\varepsilon'$ is the maximum value such that*

$$\tau(P,Q) = \int_{\mathbb{R}} \max\{f_P(t) - e^{\varepsilon'} f_Q(t), 0\}dt.$$

We now are ready to start proving our main theorem.

*Proof.* (of Theorem E.4) The idea is to construct $Q$ from $P$ by moving mass from the leftmost quantiles to the rightmost quantile. We do this such that $Q$ is statistically close enough to $P$ such that the two distributions can not be distinguished with $n$ samples, but is also far from $P$ in Wasserstein distance. This produces a lower bound of $(1/2)\mathcal{W}(P,Q)$ on how well an algorithm can simultaneously estimate $P$ and $Q$ since if there was an algorithm that produced good estimates of $P$ and $Q$ in Wasserstein distance with $n$ samples, then we could tell them apart, and this would give a contradiction.

Let $k$ be a quantity to be set later. Formally, we define $Q$ as the distribution with the following density function.

$$f_Q(t) = \left\{\begin{array}{ll} \frac{1}{2}f_P(t), & \text{for } t < q_{1/k} \\ f_P(t), & \text{for } q_{1/k} \le t < q_{1-\frac{1}{k}} \\ \frac{3}{2}f_P(t) & \text{for } q_{1-\frac{1}{k}} \le t \end{array}\right\}$$

Note that by the definition of $Q$, we have that $D_\infty(P,Q) \le 2$.

We will prove that the sample complexity of telling apart $P$ and $Q$ under $(\varepsilon, \delta)$-DP is $\Omega(k/\varepsilon)$, using known results on hypothesis testing. Then, we will argue that the Wasserstein distance between $P$ and $Q$ is sufficiently large. Setting $k$ appropriately will complete the proof.

Define $SC_{\varepsilon,\delta}(P,Q)$ to be the smallest $n$ such that there exists an $(\varepsilon, \delta)$-DP testing algorithm that distinguishes $P$ and $Q$; called the *sample complexity* of privately distinguishing $P$ and $Q$.

**Lemma E.6.** $SC_{\varepsilon,\delta}(P,Q) = \Omega(k/\varepsilon)$.

---

[4]The same bounds (and hence all our results in this subsection) can be extended to $(\varepsilon, \delta)$-DP (with $\delta \le \varepsilon$) by using an equivalence of pure and approximate DP for identity and closeness testing [ASZ17, Lemma 5].

The proof of this lemma is in Appendix J. We next argue that $P$ and $Q$ are sufficiently far away in Wasserstein distance.

**Lemma E.7.** $\mathcal{W}(P,Q) \geq \frac{1}{2k}(q_{1-\frac{1}{k}} - q_{1/k}) + \frac{1}{2}\mathcal{W}(P,P|_{q_{\frac{1}{k}},q_{1-\frac{1}{k}}})$.

The proof of this lemma is also in Appendix J.

Finally, we are ready to prove the theorem. Assume that with probability larger than $0.75$ over the draw of two datasets $\mathbf{x} \sim P^n$, $\mathbf{x}' \sim Q^n$, and the randomness used by invocations of algorithm $A$ we have that $\max(\mathcal{W}(P,A(\mathbf{x})),\mathcal{W}(Q,A(\mathbf{x}'))) < \frac{1}{2}\mathcal{W}(P,Q)$. Then, given a dataset $\mathbf{x}''$ of size $n$, we can perform the following test: run the differentially private algorithm $A$ on the dataset $\mathbf{x}''$ and compute $\mathcal{W}(P,A(\mathbf{x}''))$ and $\mathcal{W}(Q,A(\mathbf{x}''))$ and output the distribution with lower distance. Then, note that $\mathcal{W}(P,Q) \leq \mathcal{W}(P,A(\mathbf{x}'')) + \mathcal{W}(Q,A(\mathbf{x}''))$ which implies that with probability at least $0.75$, $\mathcal{W}(Q,A(\mathbf{x}'')) > \frac{1}{2}\mathcal{W}(P,Q)$ if the dataset $\mathbf{x}''$ was sampled from $P^n$ (by the accuracy guarantee). A similar argument shows that with probability at least $0.75$, $\mathcal{W}(P,A(\mathbf{x}'')) > \frac{1}{2}\mathcal{W}(P,Q)$ if the dataset $\mathbf{x}''$ was sampled from $Q^n$. Hence, with $n$ samples we have defined a test that distinguishes $P$ and $Q$. However, for $k = C\varepsilon n$ for some constant $C$, by Lemma E.6 we get that any differentially private test distinguishing $P$ and $Q$ requires more than $n$ samples, which is a contradiction. Hence, with probability at least $0.25$ over the draw of two datasets $\mathbf{x} \sim P^n$, $\mathbf{x}' \sim Q^n$, and the randomness used by invocations of algorithm $A$ we have that $\max(\mathcal{W}(P,A(\mathbf{x})),\mathcal{W}(Q,A(\mathbf{x}'))) \geq \frac{1}{2}\mathcal{W}(P,Q) \geq \frac{1}{4C\varepsilon n}(q_{1-\frac{1}{C\varepsilon n}} - q_{1/C\varepsilon n}) + \frac{1}{4}\mathcal{W}(P,P|_{q_{\frac{1}{C\varepsilon n}},q_{1-\frac{1}{C\varepsilon n}}})$ ,where the last inequality is by invoking Lemma E.7 with $k = C\varepsilon n$.

$\square$

#### E.1.2 Empirical Term

In this section, we prove the following result.

**Theorem E.8.** *Fix sufficiently large natural numbers $n, k > 0$ and let $C, C' > 0$ be sufficiently small constants. For all algorithms $A : \mathbb{R}^n \to \Delta_\mathbb{R}$, the following holds. For all continuous distributions $P$ over $\mathbb{R}$ with a density and with bounded expectation, there exists another distribution $Q$ (with $D_\infty(P,Q) \leq \ln 2$), that is indistinguishable from $P$ given $O(n)$ samples, such that with probability at least $0.25$ over the draws $\mathbf{x} \sim P^n$, $\mathbf{x}' \sim Q^n$, the following holds.*

$$\max(\mathcal{W}(P,A(\mathbf{x})),\mathcal{W}(Q,A(\mathbf{x}'))) \geq \frac{C'}{\sqrt{\log n}}\mathbb{E}_{\mathbf{x}''\sim P^n}\left[\mathcal{W}\left(P|_{q_{\frac{1}{k}},q_{1-\frac{1}{k}}},\hat{P}_n|_{q_{\frac{1}{k}},q_{1-\frac{1}{k}}}\right)\right],$$

*where $q_\alpha$ is the $\alpha$-quantile of $P$.*

Before going into the proof, we state the following result on the sample complexity of testing. This is a folklore result but for a proof of the lower bound see [BY02] and the upper bound see [Can17].

**Theorem E.9.** *Fix $n \in \mathbb{N}, \varepsilon > 0$. For every pair of distributions $P, Q$ over $\mathbb{R}$, if there exists a testing algorithm $A_{test}$ that distinguishes $P$ and $Q$ with $n$ samples, then*

$$n = \Omega\left(\frac{1}{H^2(P,Q)}\right),$$

*wherer $H^2(\cdot,\cdot)$ represents the squared Hellinger distance between $P$ and $Q$.*

Throughout the proof, we will use $q_\alpha$ to represent the $\alpha$-quantile of distribution $P$.

*Proof of Theorem E.8.* $Q$ is constructed by adding progressively more mass to $P$ up until $q_{1/2}$ and subtracting proportionate amounts of mass from $P$ afterwards. Intuitively, this is done in such a way that to 'change' $P$ to $Q$, for all $i \geq 2$ one has to move roughly $\min\{\frac{1}{\sqrt{2^i n}}, \frac{1}{2^i}\}$ mass from $q_{1/2^i}$ to $q_{1-1/2^i}$. This ensures that the Wasserstein distance between $P$ and $Q$ is larger than the expected Wasserstein distance between $P$ and its empirical distribution on $n$ samples $\hat{P}_n$. This is carefully done to ensure that $P$ is indistinguishable from $Q$.

Formally, consider $i$ in the range $[2, \log n - 1)$. For all $x \in (q_{1/2^i}, q_{1/2^{i-1}}]$, we set $f_Q(t) = f_P(t)\left[1 + \sqrt{\frac{2^i}{n}}\right]$. For all $t \in (q_{1-1/2^{i-1}}, q_{1-1/2^i}]$, we set $f_Q(t) = f_P(t)\left[1 - \sqrt{\frac{2^i}{n}}\right]$. Next,

consider $i$ in the range $[\log n, \infty)$. For all $t \in (q_{1/2^i}, q_{1/2^{i-1}}]$, we set $f_Q(t) = f_P(t)\left[1 + \frac{1}{2}\right]$. For all $t \in (q_{1-1/2^{i-1}}, q_{1-1/2^i}]$, we set $f_Q(t) = f_P(t)\left[1 - \frac{1}{2}\right]$. Note that $P$ has bounded expectation by assumption, and hence, so does $Q$. Additionally, note that $D_\infty(P, Q) \le \ln 2$.

There are two key considerations balanced in the design of $Q$. On one hand, we need $Q$ to be indistinguishable from $P$ given $\tilde{O}(n)$ samples. On the other hand, we need $Q$ to be sufficiently far away from $P$ in Wasserstein distance. This ensures that given an accurate algorithm for estimating the density of the distribution (in Wasserstein distance) given access to $\tilde{O}(n)$ samples from it, we can design a test distinguishing $P$ and $Q$ with that many samples, thereby contradicting their indistinguishability.

Detailed proofs of claims below can be found in Appendix J. First, we show that $P$ is indistinguishable from $Q$.

**Lemma E.10.**
$$KL(P, Q) = O(\log n / n).$$

Next, we establish a lower bound on the Wasserstein distance between $P$ and $Q$.

**Lemma E.11.**
$$\mathcal{W}(P, Q) \ge \frac{1}{4}\left[\sum_{j=2}^{\lceil \log n - 1 \rceil} \frac{1}{\sqrt{2^j n}}\left[q_{1-1/2^j} - q_{1/2^j}\right] + \sum_{j=\log n}^{\infty} \frac{1}{2^j}\left[q_{1-1/2^j} - q_{1/2^j}\right]\right].$$

Next, we upper bound the expected Wasserstein distance between the distribution $P$ and its empirical distribution on $n$ samples.

**Lemma E.12.**
$$\mathbb{E}[\mathcal{W}(P, \hat{P}_n)] \le 8\left[\sum_{i=2}^{\lceil \log n - 1 \rceil} \frac{1}{\sqrt{2^i n}}\left[q_{1-1/2^i} - q_{1/2^i}\right] + \sum_{i=\log n}^{\infty} \frac{1}{2^i}\left[q_{1-1/2^i} - q_{1/2^i}\right]\right]$$

We now prove a simple claim regarding restrictions.

**Claim E.13** (Restrictions preserve Wasserstein distance). *For all datasets $\mathbf{x}$, and any natural number $k > 1$ we have that*
$$\mathcal{W}(P|_{q_{\frac{1}{k}}, q_{1-\frac{1}{k}}}, \hat{P}_n|_{q_{\frac{1}{k}}, q_{1-\frac{1}{k}}}) \le \mathcal{W}(P, \hat{P}_n).$$

Finally, we are ready to put the above lemmas together to prove Theorem E.8. Fix $n' = \frac{n}{C \log n}$. Assume, for sake of contradiction, that with probability larger than $0.75$ over the draw of two datasets $\mathbf{x} \sim P^{n'}$, $\mathbf{x}' \sim Q^{n'}$, and the randomness used by invocations of algorithm $A$ we have that $\max(\mathcal{W}(P, A(\mathbf{x})), \mathcal{W}(Q, A(\mathbf{x}'))) \le \frac{1}{2}W_1(P, Q)$. Then, given a dataset $\mathbf{x}''$ of size $n'$, we perform the following test: run the differentially private algorithm $A$ on the dataset $\mathbf{x}''$ and compute $\mathcal{W}(P, A(\mathbf{x}''))$ and $\mathcal{W}(Q, A(\mathbf{x}''))$ and output the distribution with lower distance. Then, note that $\mathcal{W}(P, Q) \le \mathcal{W}(P, A(\mathbf{x}'')) + \mathcal{W}(Q, A(\mathbf{x}''))$ which implies that with probability at least $0.75$, $\mathcal{W}(Q, A(\mathbf{x}'')) \ge \frac{1}{2}\mathcal{W}(P, Q)$ if $\mathbf{x}'' \sim P^{n'}$ (by the accuracy guarantee). A similar argument shows that with probability at least $0.75$, $\mathcal{W}(P, A(\mathbf{x}'')) \ge \frac{1}{2}\mathcal{W}(P, Q)$ if $\mathbf{x}'' \sim Q^{n'}$. Hence, with $n'$ samples we have defined a test that distinguishes $P$ and $Q$. However, by Lemma E.10 bounding the $KL$ divergence between $P$ and $Q$, Theorem E.9 on sample complexity lower bounds for testing, and Lemma A.7 on the relationship between KL and Hellinger distance, we get that any statistical test distinguishing $P$ and $Q$ requires more than $n'$ samples, which is a contradiction. Hence, with probability at least $0.25$ over the draw of two datasets $\mathbf{x} \sim P^{n'}$, $\mathbf{x}' \sim Q^{n'}$, and the randomness used by invocations of algorithm $A$ we must have that

$$\max(\mathcal{W}(P, A(\mathbf{x})), \mathcal{W}(Q, A(\mathbf{x}'))) \ge \frac{1}{2}\mathcal{W}(P, Q). \tag{10}$$

Next, note that by Lemma E.12 (with value $n'$), we have that

$$\mathbb{E}[\mathcal{W}(P, \hat{P}_{n'})] \leq 8 \left[ \sum_{i=2}^{\log n'-1} \frac{1}{\sqrt{2^i n'}} \left[ q_{1-1/2^i} - q_{1/2^i} \right] + \sum_{i=\log n'}^{\infty} \frac{1}{2^i} \left[ q_{1-1/2^i} - q_{1/2^i} \right] \right]$$

$$= 8 \left[ \sum_{i=2}^{\log \frac{n}{C \log n}-1} \frac{\sqrt{C \log n}}{\sqrt{2^i n}} \left[ q_{1-1/2^i} - q_{1/2^i} \right] + \sum_{i=\log \frac{n}{C \log n}}^{\infty} \frac{1}{2^i} \left[ q_{1-1/2^i} - q_{1/2^i} \right] \right]$$

$$= 8 \left[ \sqrt{C \log n} \sum_{i=2}^{\log n-\log(C \log n)-1} \frac{1}{\sqrt{2^i n}} \left[ q_{1-1/2^i} - q_{1/2^i} \right] + \sum_{i=\log n-\log(C \log n)}^{\log n-1} \frac{1}{2^i} \left[ q_{1-1/2^i} - q_{1/2^i} \right] \right.$$

$$\left. + \sum_{i=\log n}^{\infty} \frac{1}{2^i} \left[ q_{1-1/2^i} - q_{1/2^i} \right] \right]$$

Analyzing the middle term in the above sum, we have that

$$\sum_{i=\log n-\log(C \log n)}^{\log n-1} \frac{1}{2^i} \left[ q_{1-1/2^i} - q_{1/2^i} \right] \leq \sum_{i=\log n-\log(C \log n)}^{\log n-1} \frac{1}{\sqrt{2^i}} \frac{1}{\sqrt{2^{\log n-\log(C \log n)}}} \left[ q_{1-1/2^i} - q_{1/2^i} \right]$$

$$\leq \sum_{i=\log n-\log(C \log n)}^{\log n-1} \frac{1}{\sqrt{2^i}} \frac{\sqrt{C \log n}}{\sqrt{n}} \left[ q_{1-1/2^i} - q_{1/2^i} \right]$$

$$= \sqrt{C \log n} \sum_{i=\log n-\log(C \log n)}^{\log n-1} \frac{1}{\sqrt{2^i n}} \left[ q_{1-1/2^i} - q_{1/2^i} \right]$$

Substituting this back in the previous sum, we have that

$$\mathbb{E}[\mathcal{W}(P, \hat{P}_{n'})] \leq 8 \left[ \sqrt{C \log n} \sum_{i=2}^{\log n-\log(C \log n)-1} \frac{1}{\sqrt{2^i n}} \left[ q_{1-1/2^i} - q_{1/2^i} \right] \right.$$

$$\left. + \sqrt{C \log n} \sum_{i=\log n-\log(C \log n)}^{\log n-1} \frac{1}{\sqrt{2^i n}} \left[ q_{1-1/2^i} - q_{1/2^i} \right] + \sum_{i=\log n}^{\infty} \frac{1}{2^i} \left[ q_{1-1/2^i} - q_{1/2^i} \right] \right]$$

$$\leq 8 \sqrt{C \log n} \left[ \sum_{i=2}^{\log n-1} \frac{1}{\sqrt{2^i n}} \left[ q_{1-1/2^i} - q_{1/2^i} \right] + \sum_{i=\log n}^{\infty} \frac{1}{2^i} \left[ q_{1-1/2^i} - q_{1/2^i} \right] \right]$$

$$\leq 16 \sqrt{C \log n'} \left[ \sum_{i=2}^{\log n-1} \frac{1}{\sqrt{2^i n}} \left[ q_{1-1/2^i} - q_{1/2^i} \right] + \sum_{i=\log n}^{\infty} \frac{1}{2^i} \left[ q_{1-1/2^i} - q_{1/2^i} \right] \right]$$

where in the last inequality we use the fact that $n' \geq \sqrt{n}$. Hence, by Lemma E.11 (which gives a lower bound on $\mathcal{W}(P, Q)$) in conjunction with the above equation, we have that $\mathcal{W}(P, Q) \geq \frac{C'}{\sqrt{\log n'}} \mathbb{E}[P, \hat{P}_{n'}]$ for some sufficiently small constant $C'$. Substituting back in Equation 10, we have that with probability at least 0.25 over the draw of two datasets $\mathbf{x} \sim P^{n'}$, $\mathbf{x}' \sim Q^{n'}$, and the randomness used by invocations of algorithm $A$ we have that

$$\max(\mathcal{W}(P, A(\mathbf{x})), \mathcal{W}(Q, A(\mathbf{x}'))) \geq \frac{1}{2} \frac{C'}{\sqrt{\log n'}} \mathbb{E}[\mathcal{W}(P, \hat{P}_{n'})] \geq \frac{1}{2} \frac{C'}{\sqrt{\log n'}} \mathbb{E}\left[ \mathcal{W}(P|_{q_{\frac{1}{k}}, q_{1-\frac{1}{k}}}, \hat{P}_{n'}|_{q_{\frac{1}{k}}, q_{1-\frac{1}{k}}}) \right],$$

as required.

$\square$

## E.2 Upper Bound

In this section, we describe an algorithm that achieves the instance optimal rate described in the previous section (up to polylogarithmic factors in some of the terms).

We will be looking at distributions $P$ supported on a discrete, ordered interval $\{a, a+\gamma, \ldots, b-\gamma, b\}$. Note that by a simple coupling argument, any continuous distribution $P^{cont}$ on $[a, b]$ is at most $\gamma$ away in Wasserstein distance from a distribution on this grid. The dependence on $\gamma$ in our bounds for discrete distributions will be inverse polylogarithmic (or better), and so our algorithms for estimating distributions $P$ in the interval $\{a, a+\gamma, \ldots, b-\gamma, b\}$ also work to give similar bounds for continuous distributions on $[a, b]$, up to a small additive factor of $\gamma$, which can be set to any inverse polynomial in the dataset size without significantly affecting our bounds.

Formally, we will prove the following theorem (See Theorem E.15 for a more detailed statement).

**Theorem E.14.** *Fix* $\varepsilon, \beta \in (0, 1]$, $a, b \in \mathbb{R}$, *and* $\gamma < b - a \in \mathbb{R}$ *such that* $\frac{b-a}{\gamma}$ *is an integer. Let* $n \in \mathbb{N} > c_2 \frac{\log^4 \frac{b-a}{\beta\gamma}}{\varepsilon}$ *for some sufficiently large constant* $c_2$. *There exists an* $\varepsilon$-*DP algorithm* $A$ *that for any distribution* $P$ *on* $\{a, a + \gamma, a + 2\gamma, \ldots, b - \gamma, b\}$ *satisfies the following. When run with input a random sample* $\mathbf{x} \sim P^n$, $A$ *outputs a distribution* $P^{DP}$ *such that with probability at least* $1 - \beta$ *over the randomness of* $\mathbf{x}$ *and the algorithm,*

$$\mathcal{W}(P, P^{DP}) = O\left(\frac{1}{k}\left(q_{1-\frac{1}{k}} - q_{\frac{1}{k}}\right) + \mathcal{W}(P, P|_{q_{\frac{1}{k}}, q_{1-\frac{1}{k}}}) + \sqrt{\log\frac{n}{\beta}}\mathbb{E}\left[\mathcal{W}\left(P|_{q_{\frac{1}{k}}, q_{1-\frac{1}{k}}}, \hat{P}_n|_{q_{\frac{1}{k}}, q_{1-\frac{1}{k}}}\right)\right]\right),$$

*where* $\hat{P}_n$ *is the empirical distribution on* $n$ *samples drawn independently from* $P$, $q_\alpha$ *represents the* $\alpha$-*quantile of distribution* $P$, *and* $k = \lceil\frac{\varepsilon n}{4c_3 \log^3 \frac{b-a}{\beta\gamma} \log \frac{n}{\beta}}\rceil$ *for a sufficiently large constant* $c_3$.

Since $k \approx \varepsilon n/\log(n)$, this upper bound matches the lower bound in Theorem E.3 in its dependence on $\varepsilon$ and its dependence on $n$ (up to logarithmic factors in $n$). The algorithm that we will analyze proceeds by estimating sufficiently many quantiles from the empirical distribution and distributing mass evenly between the chosen quantiles. The number of quantiles is chosen carefully to ensure that the estimated $\alpha$-quantiles are also approximately $\alpha$-quantiles for the empirical distribution (and hence also approximately for the true distribution), and to ensure that the CDF of the output distribution closely tracks the CDF of the empirical distribution. Through a careful analysis, we are able to leverage these properties to give instance optimality guarantees for the accuracy of the algorithm.

### E.2.1 Algorithm for density estimation

Algorithm 5 is our algorithm for density estimation, and proceeds by differentially privately estimating sufficiently many quantiles of the distribution and placing equal mass on each of them. We argue that a simple CDF based differentially private quantiles estimator $A_{quant}$ satisfies a specific guarantee that will be key to our analysis. See Appendix I for more details about the quantiles algorithm and formal statements and proofs therein.

---

**Algorithm 5** Algorithm $A$ for estimating a distribution on $\mathbb{R}$

> **Input:** $\mathbf{x} = (x_1, \ldots, x_n) \sim P^n$, privacy parameter $\varepsilon$, interval end-points $a, b$, granularity $\gamma$, access to algorithm $A_{quant}$
> **Output:** Distribution $P^{DP}$ on $\mathbb{R}$.
> 1: Let $k$ be set to $\lceil\frac{\varepsilon n}{4c_3 \log^3 \frac{b-a}{\beta\gamma} \log \frac{n}{\beta}}\rceil$ for a sufficiently large constant $c_3$.
> 2: Use Algorithm $A_{quant}$ referenced in Theorem I.2 with inputs interval end points $a, b$, granularity $\gamma$, $\mathbf{x} = (x_1, \ldots, x_n) \in \{a, a + \gamma, \ldots, b - \gamma\}^n$, and desired quantile values $\alpha = \{1/2k, 3/2k, 5/2k, \ldots, (2k-1)/2k\}$, and let the outputs be $\tilde{q}_1 \ldots, \tilde{q}_k$.
> 3: **for** $j \in [k]$ **do**
> 4:      Set $P^{DP}(\tilde{q}_j) = \frac{1}{k}$.
> 5: Output $P^{DP}$.

---

Observe that Algorithm 5 inherits the privacy of $A_{quant}$, since it simply postprocesses the quantiles it receives from that subroutine, and hence is also $\varepsilon$-DP.

Now, we are in a position to state our main theorem, which bounds the Wasserstein distance between the distribution output by our algorithm, and the underlying probability distribution $P$.

**Theorem E.15.** *Fix $\varepsilon, \beta \in (0, 1]$, $a, b \in \mathbb{R}$, and $\gamma < b - a \in \mathbb{R}$ such that $\frac{b-a}{\gamma}$ is an integer. Let $n \in \mathbb{N} > c_2 \frac{\log^4 \frac{b-a}{\gamma \beta \varepsilon}}{\varepsilon}$ for some sufficiently large constant $c_2$. Let $P$ be any distribution supported on $\{a, a + \gamma, a + 2\gamma, \ldots, b - \gamma, b\}$, and $\mathbf{x} \sim P^n$.*

*Then, Algorithm 5, when given inputs $\mathbf{x}$, privacy parameter $\varepsilon$, interval end points $a, b$, and granularity $\gamma$, outputs a distribution $P^{DP}$ such that with probability at least $1 - O(\beta)$ over the randomness of $\mathbf{x}$ and the algorithm,*

$$\mathcal{W}(P, P^{DP}) \leq \sqrt{c \log n} \cdot \mathbb{E} \left[ \mathcal{W}(P|_{q_{\frac{1}{k}}, q_{1 - \frac{1}{k}}}, \hat{P}_n|_{q_{\frac{1}{k}}, q_{1 - \frac{1}{k}}}) \right] + C'' \mathcal{W}(P, P|_{q_{\frac{1}{k}}, q_{1 - \frac{1}{k}}}) + \frac{2}{k} \left( q_{1 - 1/k} - q_{1/k} \right),$$

*where $\hat{P}_n$ is the uniform distribution on $\mathbf{x}$, $q_\alpha$ represents the $\alpha$-quantile of distribution $P$, $c, C''$ are sufficiently large constants, and $k = \lceil \frac{\varepsilon n}{4 c_3 \log^3 \frac{b-a}{\beta \gamma} \log \frac{n}{\beta}} \rceil$, where $c_3$ is a sufficiently large constant.*

We note that using more sophisticated differentially private CDF estimators to estimate quantiles (such as ones in [BNSV15, CLN$^+$23]), we can also obtain a version of the same theorem for approximate differential privacy, with a better dependence on the size of the domain $\frac{b-a}{\gamma}$ (only $\log^*(\frac{b-a}{\gamma})$) as opposed to $poly \log \left( \frac{b-a}{\gamma} \right)$, where $\log^* t$ is the number of times $\log$ has to be applied to $t$ to get it to be $\leq 1$). [5]

To prove Theorem E.15, we first relate the Wasserstein distance of interest (between the true distribution $P$ and the algorithm's output distribution $P^{DP}$ to a quantity related to an appropriately chosen restriction. Let $q_\alpha$ represent the $\alpha$-quantile of $P$ and $\hat{q}_\alpha$ represent the $\alpha$-quantile of $\hat{P}_n$ and $\tilde{q}_\alpha$ represent the $\alpha$-quantiles of $P^{DP}$. We also note that all these distributions (and others that will come up in the proof) are bounded distributions over the real line and so we can freely apply the triangle inequality for Wasserstein distance, and the cumulative distribution formula for Wasserstein distance (Lemma A.3). The proof of the main theorem will follow from the following lemmas (all proved in Appendix J).

**Lemma E.16.** *Let $C'' > 0$ be a sufficiently large constant, and let $n > 0$ be sufficiently large. With probability at least $1 - O(\beta)$ over the randomness in data samples and Algorithm 5,*

$$\mathcal{W}(P, P^{DP}) \leq \mathcal{W}(P|_{q_{\frac{1}{k}}, q_{1 - \frac{1}{k}}}, P^{DP}|_{q_{\frac{1}{k}}, q_{1 - \frac{1}{k}}}) + C'' \mathcal{W}(P, P|_{q_{\frac{1}{k}}, q_{1 - \frac{1}{k}}}).$$

**Lemma E.17** (Wasserstein in terms of quantiles). *For all datasets $\mathbf{x}$ (with data entries in $[a, b]$), with probability at least $1 - \beta$ over the randomness of Algorithm 5, we have that*

$$\mathcal{W}(\hat{P}_n|_{q_{\frac{1}{k}}, q_{1 - \frac{1}{k}}}, P^{DP}|_{q_{\frac{1}{k}}, q_{1 - \frac{1}{k}}}) \leq \frac{2}{k} \left( q_{1 - 1/k} - q_{1/k} \right),$$

*where $\hat{P}_n$ is the uniform distribution over $\mathbf{x}$.*

Now, we argue about the concentration of the Wasserstein distance between restrictions of the empirical distribution and restrictions of the true distribution.

**Claim E.18.** *Fix $\beta \in (0, 1)$ and sufficiently large constants $c_3, c_6$. Let $n > 0$ be sufficiently large such that $n > \log n / \beta$ (as in Theorem E.15). For all $k$ such that $\frac{1}{k} > c_3 \frac{\log \frac{n}{\beta}}{n}$, with probability at least $1 - O(\beta)$ over the randomness in the data,*

$$\mathcal{W}(P|_{q_{\frac{1}{k}}, q_{1 - \frac{1}{k}}}, \hat{P}_n|_{q_{\frac{1}{k}}, q_{1 - \frac{1}{k}}}) \leq \sqrt{c_6 \log \frac{n}{\beta}} \cdot \mathbb{E}[\mathcal{W}(P|_{q_{\frac{1}{k}}, q_{1 - \frac{1}{k}}}, \hat{P}_n|_{q_{\frac{1}{k}}, q_{1 - \frac{1}{k}}})].$$

Now, we give the proof of our main theorem.

---

[5]The theorem would be of the same form as Theorem E.15, except that Algorithm 5 would be $(\varepsilon, \delta)$-DP, with the lower bound on $n$ instead being $n = \Omega \left( \frac{\text{polylog}^* \frac{b-a}{\gamma \varepsilon \delta} \sqrt{\log 1/\delta} \log(1/\beta)}{\varepsilon} \right)$, and $k$ being set instead to $O \left( \frac{\varepsilon n}{\log^* \frac{b-a}{\gamma} \text{polylog} \frac{n}{\beta}} \right)$.

*Theorem E.15.* Using Lemma E.16, Claim E.18 and the triangle inequality, we have that with probability at least $1 - O(\beta)$ over the randomness of the data and the algorithm,

$$\mathcal{W}(P, P^{DP}) \leq \mathcal{W}(P|_{q_{\frac{1}{k}}, q_{1-\frac{1}{k}}}, P^{DP}|_{q_{\frac{1}{k}}, q_{1-\frac{1}{k}}}) + C''\mathcal{W}(P, P|_{q_{\frac{1}{k}}, q_{1-\frac{1}{k}}})$$

$$\leq \mathcal{W}(\hat{P}_n|_{q_{\frac{1}{k}}, q_{1-\frac{1}{k}}}, P^{DP}|_{q_{\frac{1}{k}}, q_{1-\frac{1}{k}}}) + \mathcal{W}(\hat{P}_n|_{q_{\frac{1}{k}}, q_{1-\frac{1}{k}}}, P|_{q_{\frac{1}{k}}, q_{1-\frac{1}{k}}}) + C''\mathcal{W}(P, P|_{q_{\frac{1}{k}}, q_{1-\frac{1}{k}}})$$

$$\leq \mathcal{W}(\hat{P}_n|_{q_{\frac{1}{k}}, q_{1-\frac{1}{k}}}, P^{DP}|_{q_{\frac{1}{k}}, q_{1-\frac{1}{k}}}) + \sqrt{c_6 \log \frac{n}{\beta}} \mathbb{E}\left[\mathcal{W}(\hat{P}_n|_{q_{\frac{1}{k}}, q_{1-\frac{1}{k}}}, P|_{q_{\frac{1}{k}}, q_{1-\frac{1}{k}}})\right] + C''\mathcal{W}(P, P|_{q_{\frac{1}{k}}, q_{1-\frac{1}{k}}})$$

Finally, applying Lemma E.17 and taking a union bound over failure probabilities, we get that with probability at least $1 - O(\beta)$ over the randomness of the data and the algorithm,

$$\mathcal{W}(P, P^{DP}) \leq \frac{2}{k}\left(q_{1-1/k} - q_{1/k}\right) + \sqrt{c_6 \log \frac{n}{\beta}} \mathbb{E}\left[\mathcal{W}(\hat{P}_n|_{q_{\frac{1}{k}}, q_{1-\frac{1}{k}}}, P|_{q_{\frac{1}{k}}, q_{1-\frac{1}{k}}})\right] + C''\mathcal{W}(P, P|_{q_{\frac{1}{k}}, q_{1-\frac{1}{k}}})$$

as required. $\square$

# F   Experiment Details

Below we describe the experiment referenced in the introduction.

**The distribution:** We have taken a distribution on $[0, 999]$, which is concentrated on two points $430$ and $440$, with $p_{430} = \frac{1}{3}$ and $p_{440} = \frac{2}{3}$. These algorithms have been run with $n = 1600$ samples from this distribution.

**Minimax Optimal Algorithm:** The minimax-optimal algorithm here is the algorithm PSMM from [HVZ23] that considers a fixed partitioning of the interval into $\Omega(m^{\frac{1}{d}})$ equal intervals and places the empirical mass in each interval on an arbitrary point in each interval. Here we consider this algorithm with $\varepsilon = \infty$, so that no noise is added. We have run it here with $K = 40$ buckets.

**Instance-optimal Algorithm:** The instance-optimal algorithm finds $k$ quantiles as in Algorithm 5. In this particular implementation, we used the recursive exponential mechanism of [KSS22], but we expect other quantile algorithms would work similarly. In this particular case, we use $k = 10$ quantiles with $\varepsilon = 1$.

# G   Proofs for Section D

**Theorem D.6.** *For every level $\ell \in [D_T]$, define the neighborhood of $P_\ell$ as $\mathcal{N}_\ell : \Delta([N_\ell]) \to \mathfrak{P}(\Delta([N_\ell]))$ by $\mathcal{N}_\ell(P_\ell) = \{Q_\ell \mid D_\infty(P_\ell, Q_\ell) \leq \ln 2\}$. Then,*

$$\mathcal{R}_{\mathcal{N}, n, \varepsilon}(P) \geq \max_{\ell \in [D_T]} r_\ell \cdot \mathcal{R}_{\mathcal{N}_\ell, n, \varepsilon}(P_\ell),$$

*where the error of $P$ is measured in the Wasserstein distance and $P_\ell$ is measured in the TV distance.*

*Proof of Theorem D.6.* Given a distribution $P$, let

$$\mathcal{A}_P^* = \arg \min_{\mathcal{A} \text{ is } \varepsilon\text{-DP}} \max_{Q \in \mathcal{N}(P)} \mathbb{E}_{D \sim P^n}[\mathcal{W}(P, \mathcal{A}(D))]$$

so $\mathcal{R}_{\mathcal{N}, n, \varepsilon}(P) = \max_{Q \in \mathcal{N}(P)} \mathbb{E}_{D \sim Q^n}[\mathcal{W}(P, \mathcal{A}_P^*(D))]$. Let $\ell \in [D_T]$. We want to define an algorithm $\mathcal{A}_{P_\ell}^*$ on the distributions in $\mathcal{N}_\ell(P_\ell)$ that achieves maximum error rate $\frac{1}{r_\ell}\mathcal{R}_{\mathcal{N}, n, \varepsilon}(P)$. Define a randomised function $g_P$ which given a node $\nu_\ell$ at level $\ell$, $g_P(\nu_\ell)$ is sampled from the distribution $P$ restricted to the leaf nodes that are children of $\nu_\ell$. Given a set of nodes at level $\ell$, define $g_P(D)$ to be the set where $g_D$ is applied to each set element individually. Then define $\mathcal{A}_{P_\ell}^*(D) = (\mathcal{A}_P^*(g_P(D)))_\ell$. Since $g_P$ is applied individually to each element in $D$, $\mathcal{A}_{P_\ell}^*$ is $\varepsilon$-DP.

Given a distribution $Q^\ell \in \mathcal{N}_\ell(P_\ell)$, define a distribution $Q$ on the leaves of the tree as follows:

$$Q(\nu) = \frac{Q^\ell(\nu_\ell)}{P_\ell(\nu_\ell)} * P(\nu),$$

where $\nu_\ell$ is the parent node of $\nu$ at level $\ell$. Note $Q \in \mathcal{N}(P)$, $g_P(Q^\ell) = Q$ and $Q_\ell = Q^\ell$. Now,

$$TV(Q^\ell, \mathcal{A}^*_{P_\ell}(D)) = TV(Q_\ell, (\mathcal{A}^*_P(g_P(D))_\ell)$$

$$\leq \frac{1}{r_\ell} \sum_{\ell' \in [D_T]} r_{\ell'} TV(Q_{\ell'}, (\mathcal{A}^*_P(g_P(D))_{\ell'})$$

$$= \frac{1}{r_\ell} \mathcal{W}(Q, \mathcal{A}^*_P(g_P(D)))$$

where the first inequality follows by definition of $\mathcal{A}^*_{P_\ell}$ and the fact $Q_\ell = Q^\ell$. Since $g_P(Q^\ell) = Q$, this implies that for all distributions in $\mathcal{N}_\ell(P_\ell)$,

$$\mathbb{E}_{D \sim Q^\ell} \left[ TV(Q^\ell, \mathcal{A}^*_{P_\ell}(D)) \right] \leq \mathbb{E}_{D \sim Q} \left[ \frac{1}{r_\ell} \mathcal{W}(Q, \mathcal{A}^*_P(D)) \right] \leq \frac{1}{r^\ell} \mathcal{R}_{\mathcal{N},n,\varepsilon}(P),$$

which implies for all levels $\ell$, $\mathcal{R}_{\mathcal{N}_\ell,n,\varepsilon}(P_\ell) \leq \frac{1}{r^\ell} \mathcal{R}_{\mathcal{N},n,\varepsilon}(P)$ and so we are done. $\qquad \square$

**Lemma G.1.** *[A extension of $(\varepsilon, \delta)$-DP Assouad's method [ASZ21]] Let $k_0, k_1, \cdots$ be a sequence of natural numbers such that $\sum_s k_s < \infty$, $\varepsilon > 0$ and $\delta \in [0,1]$. Given a family of distributions $\mathcal{P} \subset \Delta(\mathcal{X})$ on a space $\mathcal{X}$, a parameter $\theta : \mathcal{P} \to \mathcal{M}$ where $\mathcal{M}$ is a metric space with metric $d$, suppose that there exists a set $\mathcal{V} \subset \mathcal{P}$ of distributions indexed by the product of hypercubes $\mathcal{E}_{k_0} \times \mathcal{E}_{k_1} \times \cdots$ where $\mathcal{E}_k := \{\pm 1\}^k$ such that for a sequence $\tau_0, \tau_1, \cdots$,*

$$\forall (u^0, u^1, \cdots), (v^0, v^1, \cdots) \in \mathcal{E}_{k_0} \times \mathcal{E}_{k_1} \times \cdots, \quad d(\theta(p_u), \theta(p_v)) \geq 2 \sum_s \tau_s \sum_{j=1}^{k_s} \chi_{u_j^s \neq v_j^s}. \quad (6)$$

*For each coordinate $s \in \mathbb{N}$, $j \in [k_s]$, consider the mixture distributions obtained by averaging over all distributions with a fixed value at the $(s,j)$th coordinate:*

$$p_{+(s,j)} = \frac{2}{|\mathcal{E}_{k_0} \times \mathcal{E}_{k_1} \times \cdots|} \sum_{u \in \mathcal{E}_{k_0} \times \mathcal{E}_{k_1} \times \cdots : u_j^s = +1} p_u, \quad p_{-(s,j)} = \frac{2}{|\mathcal{E}_{k_0} \times \mathcal{E}_{k_1} \times \cdots|} \sum_{u \in \mathcal{E}_{k_0} \times \mathcal{E}_{k_1} \times \cdots : u_j^s = -1} p_u,$$

*and let $\phi_{s,j} : \mathcal{X}^n \to \{-1, +1\}$ be a binary classifier. Then*

$$\min_{\mathcal{A} \text{ is } (\varepsilon,\delta)\text{-DP}} \max_{p \in \mathcal{V}} \mathcal{R}_{\mathcal{A},n}(p) \geq \frac{1}{2} \sum_s \tau_s \sum_{j=1}^{k_s} \min_{\phi_{s,j} \text{ is } (\varepsilon,\delta)\text{-DP}} \left( \Pr_{X \sim p_{+(s,j)}^n} (\phi_{s,j}(X) \neq 1) + \Pr_{X \sim p_{-(s,j)}^n} (\phi_{s,j}(X) \neq -1) \right),$$

*where the min on the LHS is over all $(\varepsilon, \delta)$-DP mechanisms, and on the right hand side is over all $(\varepsilon, \delta)$-DP binary classifiers. Moreover, if for all $s \in \mathbb{N}$, $j \in [k_s]$, there exists a coupling $(X, Y)$ between $p_{+(s,j)}^n$ and $p_{-(s,j)}^n$ with $\mathbb{E}[d_{Ham}(X,Y)] \leq D_s$, then*

$$\min_{\mathcal{A} \text{ is } (\varepsilon,\delta)\text{-DP}} \max_{p \in \mathcal{V}} \mathcal{R}_{\mathcal{A},n}(p) \geq \sum_s \frac{k_s \tau_s}{2} (0.9 e^{-10 \varepsilon D_s} - 10 D_s \delta)$$

*Proof of Lemma D.8.* We will follow the proof of Theorem 3 in [ASZ21]. Given an estimator $\mathcal{A}$, define a classifier $\mathcal{A}^*$ by projecting on the product of hypercubes so

$$\mathcal{A}^*(X) = \arg \min_{u \in (\mathcal{E}_{k_0} \times \mathcal{E}_{k_1} \times \cdots)} d(\mathcal{A}(X), \theta(p_u)).$$

By the triangle inequality and the definition of $\mathcal{A}^*$, for any $p \in \mathcal{V}$,

$$d(\theta(p_{\mathcal{A}^*(X)}), \theta(p)) \leq d(\mathcal{A}(X), \theta(p_{\mathcal{A}^*(X)})) + d(\mathcal{A}(X), \theta(p)) \leq 2 d(\mathcal{A}(X), \theta(p)).$$

Therefore, we can restrict to a lower bound on the performance of DP classifiers:

$$\min_{\mathcal{A} \text{ is } (\varepsilon,\delta)\text{-DP}} \max_{p \in \mathcal{V}} \mathcal{R}_{\mathcal{A},n}(p) \geq \frac{1}{2} \min_{\mathcal{A}^* \text{ is } (\varepsilon,\delta)\text{-DP}} \max_{p \in \mathcal{V}} \mathbb{E}_{X \sim p^n} [d(\theta(p_{\mathcal{A}^*(X)}), \theta(p))]. \quad (11)$$

Also, for any $(\varepsilon, \delta)$-DP classifier $\mathcal{A}^*$,

$$\max_{p \in \mathcal{V}} \mathbb{E}_{X \sim p^n}[d(\theta(p_{\mathcal{A}^*(X)}), \theta(p))] \geq \frac{1}{|\mathcal{V}|} \sum_{u \in (\mathcal{E}_{k_0} \times \mathcal{E}_{k_1} \times \cdots)} \mathbb{E}_{X \sim p_u^n}[d(\theta(p_{\mathcal{A}^*(X)}), \theta(p_u))]$$

$$\geq \frac{2}{|\mathcal{V}|} \sum_s \tau_s \sum_{j=1}^{k_s} \sum_{u \in \mathcal{E}_{k_0} \times \mathcal{E}_{k_1} \times \cdots} \Pr_{X \sum p_u^n}(\mathcal{A}^*(X)_j^s \neq u_j^s),$$

where the first inequality follows from the fact that the max is greater than the average, and the second follows from assumption (6). For each $(s, j)$ pair, we divide $\mathcal{E}_{k_0} \times \mathcal{E}_{k_1} \times \cdots$ into two groups;

$$\max_{p \in \mathcal{V}} \mathbb{E}_{X \sim p^n}[d(\theta(p_{\mathcal{A}^*(X)}), \theta(p))]$$

$$\geq \frac{2}{|\mathcal{V}|} \sum_s \tau_s \sum_{j=1}^{k_s} \left[ \sum_{u \in (\mathcal{E}_{k_0} \times \mathcal{E}_{k_1} \times \cdots) \mid u_j^s = +1} \Pr_{X \sum p_u^n}(\mathcal{A}^*(X)_j^s \neq u_j^s) + \sum_{u \in (\mathcal{E}_{k_0} \times \mathcal{E}_{k_1} \times \cdots) \mid u_j^s = -1} \Pr_{X \sim p_u^*}(\mathcal{A}^*(X)_j^s \neq u_j^s) \right]$$

$$\geq \frac{2}{|\mathcal{V}|} \sum_s \tau_s \sum_{j=1}^{k_s} \left[ \sum_{u \in (\mathcal{E}_{k_0} \times \mathcal{E}_{k_1} \times \cdots) \mid u_j^s = +1} \Pr_{X \sim p_u^n}(\mathcal{A}^*(X)_j^s \neq u_j^s) + \sum_{u \in (\mathcal{E}_{k_0} \times \mathcal{E}_{k_1} \times \cdots) \mid u_j^s = -1} \Pr_{X \sim p_u^n}(\mathcal{A}^*(X)_j^s \neq u_j^s) \right]$$

$$\geq \sum_s \tau_s \sum_{j=1}^{k_s} (\Pr_{X \sim p_{+(s,j)}^n}(\mathcal{A}^*(X) \neq +1) + \Pr_{X \sim p_{-(s,j)}^n}(\mathcal{A}^*(X) \neq -1))$$

$$\geq \sum_s \tau_s \sum_{j=1}^{k_s} (\Pr_{X \sim p_{+(s,j)}^n}(\phi_{s,j}(X) \neq +1) + \Pr_{X \sim p_{-(s,j)}^n}(\phi_{s,j}(X) \neq -1)).$$

Combining with eqn 11 we have the first statement. Next, since for each pair $(s, j)$, there exists a coupling $(X, Y)$ between $p_{+(s,j)}$ and $p_{-(s,j)}$ such that $\mathbb{E}[d_{Ham}(X, Y)] \leq D_s$, we can use the DP version of Le Cam's method from [ASZ21] to give for any classifier $\phi_{s,j}$,

$$\Pr_{X \sim p_{+(s,j)}^n}(\phi_{s,j}(X) \neq +1) + \Pr_{X \sim p_{-(s,j)}^n}(\phi_{s,j}(X) \neq -1) \geq \frac{1}{2}(0.9 e^{-10 \varepsilon D_s} - 10 D_s \delta),$$

which implies the final result. $\qquad \square$

**Lemma G.2.** *Given any pair of distributions $P$ and $Q$ on the same domain,*

$$\min_\phi \left( \Pr_{X \sim P^n}(\phi(X) = 1) + \Pr_{X \sim Q^n}(\phi(X) = -1) \right) \geq \frac{1}{2}(1 - \sqrt{n \mathrm{KL}(P, Q)}),$$

*where the minimum is over all binary classifiers. In particular, if $P = \texttt{Bernoulli}(p - \alpha)$ and $Q = \texttt{Bernoulli}(p + \alpha)$ where $0 \leq \alpha \leq \frac{1}{2} L(p)$ then*

$$\min_\phi \left( \Pr_{X \sim P^n}(\phi(X) = 1) + \Pr_{X \sim Q^n}(\phi(X) = -1) \right) \geq 1/4,$$

*where again the minimum is over all binary classifiers.*

*Proof of Lemma D.12.* A standard result in the statistics literature states that for any pair of distributions $P$ and $Q$,

$$\min_\phi \left( \Pr_{X \sim P^n}(\phi(X) = 1) + \Pr_{X \sim Q^n}(\phi(X) = -1) \right) = \frac{1}{2}(1 - \mathrm{TV}(P^n, Q^n)) \geq \frac{1}{2}(1 - \sqrt{n \mathrm{KL}(P, Q)}),$$

where the minimum is over all binary classifiers. If $P = \texttt{Bernoulli}(p - \alpha)$ and $Q = \texttt{Bernoulli}(p + \alpha)$ where $0 \le \alpha \le \frac{1}{2}L(p)$ then

$$
\begin{aligned}
\mathrm{KL}(Q, P) &= (p + \alpha) \ln \frac{p + \alpha}{p - \alpha} + (1 - p - \alpha) \ln \frac{1 - p - \alpha}{1 - p + \alpha} \\
&= (p + \alpha) \ln \left( 1 + \frac{2\alpha}{p - \alpha} \right) + (1 - p - \alpha) \ln \left( 1 - \frac{2\alpha}{1 - p + \alpha} \right) \\
&\le (p + \alpha) \frac{2\alpha}{p - \alpha} - (1 - p - \alpha) \frac{2\alpha}{1 - p + \alpha} \\
&= \frac{4\alpha^2}{p - \alpha} + \frac{4\alpha^2}{1 - p + \alpha} \\
&= \frac{\alpha^2}{(p - \alpha)(1 - p + \alpha)} \\
&\le \frac{1}{4n}.
\end{aligned}
$$

where the first inequality holds since $\ln(1 + x) < x$ for $x \in [-1, 1]$ and by assumption $2\alpha/(p - \alpha), 2\alpha/(1 - p + \alpha) \in [0, 1]$ and the second follows again because of the constraint on $\alpha$. $\qquad \square$

**Lemma G.3.** *For any distribution $P$, if $\log(n/\beta) > 1$ then with probability $1 - 3D_T\beta$,*

$$
\mathfrak{W}(\widehat{\mathfrak{G}_P}, \mathfrak{G}_P) \le \sum_{\ell \in [D_T]} \sum_{x \in [N_\ell]} \min \left\{ P_\ell(x)(1 - P_\ell(x)), 4\sqrt{3 \frac{P_\ell(x)(1 - P_\ell(x)) \log(n/\beta)}{n}} \right\}
$$

Lemma D.14 is an immediate corollary of the following lemma.

**Lemma G.4.** *For any distribution $P$, if $\log(n/\beta) > 1$ then with probability $1 - 3D_T\beta$,*

$$
\mathfrak{W}(\widehat{\mathfrak{G}_P}, \mathfrak{G}_P) \le \sum_{\ell \in [D_T]} \sum_{x \in [N_\ell]} \min \left\{ P_\ell(x), 1 - P_\ell(x), 4\sqrt{3 \frac{P_\ell(x) \log(n/\beta)}{n}}, 4\sqrt{3 \frac{(1 - P_\ell(x)) \log(n/\beta)}{n}} \right\}
$$

*Proof of Lemma D.14.* We'll consider each level of the tree individually then use a union bound over all the levels to obtain our final bound. Let $(\hat{P}_\ell)_n$ be the empirical distribution without truncation. The following conditions are sufficient to ensure that the bounds hold for a single level $\ell$:

$$
\sup_{\nu \text{ s.t. } P_\ell(\nu) \le \frac{3 \ln(n/\beta)}{n}} (\hat{P}_\ell)_n(\nu) \le \frac{7 \ln(n/\beta)}{n}
$$

$$
\sup_{\nu \text{ s.t. } P_\ell(\nu) \ge 1 - \frac{3 \ln(n/\beta)}{n}} (\hat{P}_\ell)_n(\nu) \ge 1 - \frac{7 \ln(n/\beta)}{n}
$$

$$
\forall \left( \nu \text{ s.t. } P_\ell(\nu) \in \left[ \frac{3 \ln(n/\beta)}{n}, 1 - \frac{3 \ln(n/\beta)}{n} \right] \right),
$$

$$
|(\hat{P}_\ell)_n(x) - P_\ell(\nu)| \le \min \left\{ \sqrt{\frac{3 P_\ell(\nu) \ln(n/\beta)}{n}}, \sqrt{\frac{3(1 - P_\ell(x)) \ln(n/\beta)}{n}} \right\}
$$

We will begin by showing these conditions are sufficient. If $P_\ell(\nu) \notin [\frac{3 \ln(n/\beta)}{n}, 1 - \frac{3 \ln(n/\beta)}{n}]$ then these conditions imply that the empirical density for node $\nu$ is truncated, and hence the error that that node is either $P_\ell(\nu)$ or $1 - P_\ell(\nu)$ (when $P_\ell(\nu) < 1/2$ and $P_\ell(\nu) > 1/2$, respectively), as required. If $P_\ell(\nu) \in [\frac{3 \ln(n/\beta)}{n}, 1 - \frac{3 \ln(n/\beta)}{n}]$ then either the estimate is not truncated and the error is less than $\min \left\{ \sqrt{\frac{3 P_\ell(\nu) \ln(2n/\beta)}{n}}, \sqrt{\frac{3(1 - P_\ell(x)) \ln(2n/\beta)}{n}} \right\} \le \min\{P_\ell(\nu), 1 - P_\ell(\nu)\}$, as required. Or the estimate is truncated and the error is $\min\{P_\ell(\nu), 1 - P_\ell(\nu)\}$. Under the above conditions, if $P_\ell(\nu) \le 1/2$ then truncation will only occur if

$$
P_\ell(\nu) - \sqrt{\frac{3p \ln(2n/\beta)}{n}} \le \frac{7 \ln(n/\beta)}{n} \le \sqrt{\frac{7 \ln(n/\beta)}{n} \frac{7}{3} \frac{3 \ln(n/\beta)}{n}} \le \sqrt{\frac{7 \ln(n/\beta)}{n} \frac{7}{3} p} = \frac{7}{3} \sqrt{\frac{3 \ln(n/\beta)}{n} p},
$$

in which case $P_\ell(nu) \le 4\sqrt{\frac{3P_\ell(\nu)\ln(n/\beta)}{n}}$, as required. Similarly, if $P_\ell(\nu) > 1/2$ then truncation will only occur if $1 - P_\ell(\nu) \le 4\sqrt{\frac{3(1-P_\ell(\nu))\ln(n/\beta)}{n}}$, as required.

We will now show that these conditions hold simultaneously with probability at least $1 - 3\beta$ for all the nodes at level $\ell$. If $P_\ell(\nu) \le \frac{1}{en}$ then using the multiplicative form of Chernoff bound,

$$
\Pr\left((\hat{P}_\ell)_n(\nu) \ge \frac{3\ln(n/\beta)}{n}\right) = \Pr\left((\hat{P}_\ell)_n(\nu) \ge \left(1 + \frac{3\ln(n/\beta)}{P_\ell(\nu)n} - 1\right)P_\ell(\nu)\right)
$$

$$
\le \left(\frac{e^{\frac{3\ln(n/\beta)}{nP_\ell(\nu)}-1}}{\left(\frac{3\ln(n/\beta)}{nP_\ell(\nu)}\right)^{\frac{3\ln(n/\beta)}{nP_\ell(\nu)}}}\right)^{P_\ell(\nu)n}
$$

$$
\le \left(\frac{enP_\ell(\nu)}{3\ln(n/\beta)}\right)^{3\ln(n/\beta)}
$$

$$
\le P_\ell(\nu)n\left(\frac{e}{3\ln(n/\beta)}\right)^{3\ln(n/\beta)}(nP_\ell(\nu))^{3\ln(n/\beta)-1}
$$

Firstly, since $\ln(n/\beta) \ge 1$, $\left(\frac{e}{3\ln(n/\beta)}\right)^{3\ln(n/\beta)} \le 1$. Further, $nP_\ell(\nu) \le 1/e$ and $3\ln(n/\beta) - 1 \ge \ln(n/\beta)$ so $(nP_\ell(\nu))^{3\ln(n/\beta)-1} \le (1/e)^{\ln(n/\beta)} = \beta/n$. Therefore,

$$
\Pr\left((\hat{P}_\ell)_n(\nu) \ge \frac{3\ln(n/\beta)}{n}\right) \le P_\ell(\nu)\beta. \tag{12}
$$

Let $\mathcal{S} = \{x \in [N_\ell] \mid P_\ell(x) < 1/(en)\}$ then using a union bound and Eqn (12) we have

$$
\Pr\left(\exists x \in \mathcal{S} \text{ s.t. } (\hat{P}_\ell)_n(x) \ge \frac{2\sqrt{2}\log(n/\beta)}{n}\right) \le \sum_{x \in \mathcal{S}} P_\ell(\nu)\beta \le \beta
$$

There exist at most $n$ elements in $[N_\ell]$ that do not belong in $\mathcal{S}$. We will prove that, independently, each of these elements satisfy the required condition with probability $\le 2\beta/n$ then a union bound proves the final result. If $P_\ell(\nu) \in [\frac{3\ln(n/\beta)}{n}, 1 - \frac{3\ln(n/\beta)}{n}]$ then using the multiplicative form of Chernoff bound (If $X_i$ are all i.i.d. and $0 < \delta < 1$, then $\Pr(|\sum_{i=1}^n X_i - n\mathbb{E}[X_1]| \ge \delta n\mathbb{E}[X_1]) \le 2e^{-\delta^2 n\mathbb{E}[X_1]/3}$),

$$
\Pr\left(|(\hat{P}_\ell)_n(x) - P_\ell(x)| \ge \sqrt{\frac{3P_\ell(x)\log(n/\beta)}{n}}\right) = \Pr\left(|(\hat{P}_\ell)_n(x) - P_\ell(x)| \ge \sqrt{\frac{3\log(n/\beta)}{P_\ell(x)n}}P_\ell(x)\right)
$$

$$
\le 2e^{-\left(\frac{3\log(n/\beta)}{P_\ell(x)n}\right)\frac{P_\ell(x)n}{3}}
$$

$$
= 2\beta/n.
$$

Next, if $P_\ell(\nu) \le \frac{3\ln(n/\beta)}{n}$ then using the additive form of Chernoff bound (If $X_i$ are all i.i.d. and $\varepsilon \ge 0$, then $\Pr\left(\frac{1}{n}\sum_{i=1}^n X_i \ge \mathbb{E}[X_1] + \varepsilon\right) \le e^{-\varepsilon^2 n/(2(p+\varepsilon))}$)

$$
\Pr\left((\hat{P}_\ell)_n(\nu) \ge \frac{7\ln(n/\beta)}{n}\right) \le \Pr\left((\hat{P}_\ell)_n(\nu) \ge p + \left(7\frac{\ln(n/\beta)}{n} - p\right)\right)
$$

$$
\le e^{-\frac{(7\frac{\ln(n/\beta)}{n}-p)^2 n}{14\frac{\ln(n/\beta)}{n}}}
$$

$$
\le e^{-\frac{(4\frac{\ln(n/\beta)}{n})^2 n}{14\frac{\ln(n/\beta)}{n}}}
$$

$$
\le e^{-\ln(n/\beta)}
$$

$$
= \beta/n.
$$

By symmetry, if $P_\ell(\nu) \geq 1 - \frac{3\ln(n/\beta)}{n}$ then

$$\Pr\left((\hat{P}_\ell)_n(\nu) \leq 1 - \frac{7\ln(n/\beta)}{n}\right) \leq \beta/n.$$

$\square$

**Lemma G.5.** *Let $\hat{\gamma}_\varepsilon$ be the set of active nodes found in Algorithm 1. Then with probability $1 - D_T(\log n + 4\varepsilon n)\beta$,*

$$\gamma_P\left(\max\left\{\frac{2}{\varepsilon n} + 4\frac{\log(2/\beta)}{\varepsilon n}, \frac{192\log(n/\beta)}{n}\right\}\right) \subset \hat{\gamma}_\varepsilon \subset \gamma_P\left(\frac{1}{2\varepsilon n}\right).$$

*Proof of Lemma D.15.* First notice that if a node $\nu$ is an $\alpha$-active node, then all of it's ancestor nodes are also $\alpha$-active. So, it suffices to show that (with high probability) if at any stage a node makes to it Line 7 of Algorithm 3, then if $\nu \notin \gamma_P(2\kappa)$ then $\widehat{\mathfrak{G}_P}(\nu) + \mathsf{Lap}(\frac{1}{\varepsilon n}) \leq 2\kappa + \frac{\log(2/\beta)}{\varepsilon n}$ and if $\nu \in \gamma_P\left(\max\left\{\frac{2}{\varepsilon n} + 4\frac{\log(2/\beta)}{\varepsilon n}, \frac{\log(n/\beta)}{n}\right\}\right)$ then $\widehat{\mathfrak{G}_P}(\nu) + \mathsf{Lap}(\frac{1}{\varepsilon n})) > 2\kappa + \frac{\log(2/\beta)}{\varepsilon n}$.

By Lemma D.14, with probability $1 - 3D_T\beta$, all nodes $\nu$ satisfy

$$|\widehat{\mathfrak{G}_P}(\nu) - \mathfrak{G}_P(\nu)| \leq \min\left\{\mathfrak{G}_P(\nu)(1 - \mathfrak{G}_P(\nu)), 4\sqrt{\frac{3\mathfrak{G}_P(\nu)(1 - \mathfrak{G}_P(\nu))\log(n/\beta)}{n}}\right\}$$

Further, if one samples $X$ independent samples from $\mathsf{Lap}(\frac{1}{\varepsilon n})$ then with probability $1 - X\beta$,

$$\sup|\mathsf{Lap}(\frac{1}{\varepsilon n})| \leq \frac{\ln(2/\beta)}{\varepsilon n}.$$

So conditioning on both these events if $x \notin \gamma_P\left(\frac{1}{2\varepsilon n}\right)$,

$$\widehat{\mathfrak{G}_P}(\nu) + \mathsf{Lap}(\frac{1}{\varepsilon n}) \leq \mathfrak{G}_P(\nu) + \frac{\ln(2/\beta)}{\varepsilon n} \leq \frac{1}{2\varepsilon n} + \frac{\ln(2/\beta)}{\varepsilon n},$$

so they will not survive Line 7 of Algorithm 3. If $x \in \gamma_P\left(\max\{\frac{2}{\varepsilon n} + 4\frac{\log(2/\beta)}{\varepsilon n}, \frac{192\log(n/\beta)}{n}\}\right)$ then

$$\begin{aligned}
\widehat{\mathfrak{G}_P}(\nu) + \mathsf{Lap}(\frac{1}{\varepsilon n}) &\geq \mathfrak{G}_P(\nu) - 4\sqrt{3\frac{\mathfrak{G}_P(\nu)\log(n/\beta)}{n}} - \frac{\ln(2/\beta)}{\varepsilon n} \\
&\geq \frac{1}{2}\mathfrak{G}_P(\nu) - \frac{\log(2/\beta)}{\varepsilon n} \\
&\geq \frac{1}{\varepsilon n} + \frac{\log(2/\beta)}{n}
\end{aligned}$$

Each level has at most $2\varepsilon n$ in $\gamma_P\left(\frac{1}{2\varepsilon n}\right)$ so we query at most $4\varepsilon n$ nodes in the tree when running LocateActiveNodes since each node has at most 2 children. Therefore, we can set $X = 4\varepsilon n D_T$.

$\square$

**Lemma G.6.** *If $\gamma_P\left(\max\{\frac{2}{\varepsilon n} + 4\frac{\log(2/\beta)}{\varepsilon n}, \frac{192\log(n/\beta)}{n}\}\right) \subset \hat{\gamma}_\varepsilon$ then*

$$\mathfrak{W}(\widehat{\mathfrak{G}_P}, \widehat{\mathfrak{G}_P}|_{\hat{\gamma}_\varepsilon}) \leq \mathfrak{W}(\mathfrak{G}_P, \widehat{\mathfrak{G}_P}) + \mathfrak{W}(\mathfrak{G}_P, \mathfrak{G}_P|_{\gamma_P\left(\max\{\frac{2}{\varepsilon n} + 4\frac{\log(2/\beta)}{\varepsilon n}, \frac{192\log(n/\beta)}{n}\}\right)})$$

*Proof of Lemma D.16.* The key component of this proof is that any discrepancy between the weight of the nodes on $P$ and that assigned by $\widehat{\mathfrak{G}}_P$ was already paid for in $\mathcal{W}(P, \widehat{\mathfrak{G}}_P)$.

$$\mathfrak{W}(\widehat{\mathfrak{G}_P}, \widehat{\mathfrak{G}_P}|_{\hat{\gamma}_\varepsilon}) = \sum_{\nu \notin \hat{\gamma}_\varepsilon} r_\nu |\widehat{\mathfrak{G}_P}(\nu)|$$

$$\leq \sum_{\nu \notin \gamma_P \left( \max\{ \frac{2}{\varepsilon n} + 4 \frac{\log(2/\beta)}{\varepsilon n}, \frac{192 \log(n/\beta)}{n} \} \right)} r_\nu |\widehat{\mathfrak{G}_P}(\nu)|$$

$$= \sum_{\nu \notin \gamma_P \left( \max\{ \frac{2}{\varepsilon n} + 4 \frac{\log(2/\beta)}{\varepsilon n}, \frac{192 \log(n/\beta)}{n} \} \right)} r_\nu |\widehat{\mathfrak{G}_P}(\nu) - \mathfrak{G}_P(\nu) + \mathfrak{G}_P(\nu)|$$

$$\leq \sum_{\nu \notin \gamma_P \left( \max\{ \frac{2}{\varepsilon n} + 4 \frac{\log(2/\beta)}{\varepsilon n}, \frac{192 \log(n/\beta)}{n} \} \right)} r_\nu |\widehat{\mathfrak{G}_P}(\nu) - \mathfrak{G}_P(\nu)|$$

$$+ \sum_{\nu \notin \gamma_P \left( \max\{ \frac{2}{\varepsilon n} + 2 \frac{\log(2/\beta)}{n}, \frac{192 \log(n/\beta)}{n} \} \right)} r_\nu |\mathfrak{G}_P(\nu)|$$

$$\leq \mathfrak{W}(\mathfrak{G}_P, \widehat{\mathfrak{G}_P}) + \mathfrak{W}(\mathfrak{G}_P, \mathfrak{G}_P|_{\gamma_P \left( \max\{ \frac{2}{\varepsilon n} + 4 \frac{\log(2/\beta)}{\varepsilon n}, \frac{192 \log(n/\beta)}{n} \} \right)})$$

as required. $\qquad \square$

**Lemma G.7.** *For any real-valued function $\mathfrak{G}$ on the nodes of the HST such that $\mathfrak{G}(\nu_0) = 1$ where $\nu_0$ is the root node and given any distribution $P$,*

$$\mathcal{W}(P, \texttt{Projection}(\mathfrak{G})) \leq 4\mathfrak{W}(\mathfrak{G}_P, \mathfrak{G}).$$

*Proof of Lemma D.17.* We first note that for any pair of sequences of real values $a_1, \cdots, a_k$ and $b_1, \cdots, b_k$, and constant $A$ such that $\sum_i a_i \neq 0$,

$$\sum |\frac{A}{\sum a_i} a_i - b_i| \leq \sum |\frac{A}{\sum a_i} a_i - a_i| + |a_i - b_i| = |A - \sum a_i| + \sum |a_i - b_i| \leq |A - \sum b_i| + 2 \sum |a_i - b_i|.$$

Also if $\sum a_i = 0$ then

$$\sum |\frac{A}{k} - b_i| \leq \sum |\frac{A}{k} - \frac{\sum_i b_i}{k}| + |\frac{\sum_i b_i}{k} - b_i| = |A - \sum b_i| + 2 \sum |b_i| = |A - \sum b_i| + 2 \sum |a_i - b_i|$$

Let $\bar{\mathfrak{G}}^\ell$ be the function $\bar{\mathfrak{G}}$ after only levels $0, \cdots, \ell$ have been updated. So $\bar{\mathfrak{G}}^\ell$ matches $\bar{\mathfrak{G}}^{\ell-1}$ on all levels except $\ell$. Let $\nu$ be a node in the $\ell$th level of the HST. If we suppose the sum is over the normalised children of a node $\nu$, $A = \bar{\mathfrak{G}}^{\ell-1}(\nu)$, and for all the children $\nu'$ of $\nu$, $a_i = \mathfrak{G}(\nu')$ and $b_i = \mathfrak{G}_P(\nu')$, we can see that the contribution to the Wasserstein distance by the children increases by an additive factor of $|\mathfrak{G}^{\ell-1}(\nu) - \mathfrak{G}_P(\nu)|$. Iterating, we can see that

$$\mathcal{W}(P, \texttt{Projection}(\mathfrak{G})) \leq 2 \sum_{\ell=0}^{D_T} \sum_{\nu \text{ at level } \ell} (r_\ell + r_{\ell+1} \cdots r_{D_T}) |\mathfrak{G}(\nu) - \mathfrak{G}_P(\nu)| \leq 4 \sum_{\ell=0}^{D_T} \sum_{\nu \text{ at level } \ell} r_\ell |\mathfrak{G}(\nu) - \mathfrak{G}_P(\nu)|,$$

which is 4 times the wasserstein distance.

$\qquad \square$

# H   Local Minimality in the High Dimensional Setting

**Theorem H.1.** *Given any $\varepsilon > 0$, and a distribution $P$, and let $n' = \frac{5}{4 \min\{W(\frac{0.45\varepsilon}{\delta}), 0.6\}} n$, then for all $(\varepsilon, \delta)$-DP algorithms $\mathcal{A}'$, there exists a distribution $Q \in \mathcal{N}(P)$ such that with probability $1 - (D_T \log n + 4D_T \varepsilon n)\beta$,*

$$\mathcal{W}(Q, \hat{Q}_{\varepsilon, n'}) \leq \tilde{O}(\mathbb{E}_{X \sim Q^n, \mathcal{A}'}(\mathcal{W}(\mathcal{A}'(X), Q))),$$

*where $\hat{Q}_{\varepsilon, n'}$ is the output of* `PrivDensityEstTree(Q)` *with $n'$ samples.*

*Proof.* First, let us obtain a slightly simpler upper bound on $\mathcal{W}(P, \hat{P}_\varepsilon)$. From eqn (9) in the proof of Theorem D.13 we have that for each level $\ell$,

$$\mathcal{W}(P_\ell, (\hat{P}_\varepsilon)_\ell) \leq 2 \left( \mathfrak{W}((\mathfrak{G}_P)_\ell, (\widehat{\mathfrak{G}_P})_\ell) + \mathfrak{W}((\widehat{\mathfrak{G}_P})_\ell, (\widehat{\mathfrak{G}_P}|\hat{\gamma}_\varepsilon)_\ell) + \mathfrak{W}((\widehat{\mathfrak{G}_P}|\hat{\gamma}_\varepsilon)_\ell, (\widetilde{\mathfrak{G}_{\hat{P}_n,\hat{\gamma}_\varepsilon}})_\ell) \right),$$

from Lemma D.15 we have that with probability $1 - D_T(\log n + 4\varepsilon n)\beta$,

$$\gamma_P \left( \max \left\{ \frac{2}{\varepsilon n} + 4\frac{\log(2/\beta)}{\varepsilon n}, \frac{192\log(n/\beta)}{n} \right\} \right) \subset \hat{\gamma}_\varepsilon \subset \gamma_P \left( \frac{1}{2\varepsilon n} \right),$$

and if one samples $4\varepsilon n D_T$ independent samples from $\mathsf{Lap}(\frac{1}{\varepsilon n})$ then we have that with probability $1 - 4\varepsilon n D_T \beta$,

$$\sup |\mathsf{Lap}(\frac{1}{\varepsilon n})| \leq \frac{\ln(2/\beta)}{\varepsilon n}.$$

Therefore, for all $\nu \notin \hat{\gamma}_\varepsilon$ we have $P_\ell(\nu) \leq \max\left\{ \frac{2}{\varepsilon n} + 4\frac{\log(2/\beta)}{\varepsilon n}, \frac{192\log(n/\beta)}{n} \right\} \leq C\frac{\ln(n/\beta)}{\varepsilon n}$ for some constant $C$ therefore,

$$\mathfrak{W}((\widehat{\mathfrak{G}_P})_\ell, (\widehat{\mathfrak{G}_P}|\hat{\gamma}_\varepsilon)_\ell) + \mathfrak{W}((\widehat{\mathfrak{G}_P}|\hat{\gamma}_\varepsilon)_\ell, (\widetilde{\mathfrak{G}_{\hat{P}_n,\hat{\gamma}_\varepsilon}})_\ell) \leq \sum_{\nu \notin \hat{\gamma}_\varepsilon} P_\ell(\nu) + \sum_{\nu \in \hat{\gamma}_\varepsilon} \frac{\ln(2/\beta)}{\varepsilon n}$$

$$\leq \sum_{\nu \notin \gamma_P\left(\frac{1}{2\varepsilon n}\right)} P_\ell(\nu) + \sum_{\nu \in \gamma_P\left(\frac{1}{2\varepsilon n}\right)\backslash\hat{\gamma}_\varepsilon} C\frac{\ln(n/\beta)}{\varepsilon n} + \sum_{\nu \in \hat{\gamma}_\varepsilon} \frac{\ln(2/\beta)}{\varepsilon n}$$

$$\leq \sum_{\nu \notin \gamma_P\left(\frac{1}{2\varepsilon n}\right)} P_\ell(\nu) + C\ln(n/\beta) \sum_{\nu \in \gamma_P\left(\frac{1}{2\varepsilon n}\right)} \frac{1}{\varepsilon n}.$$

For the same reason as in the proof of Theorem D.13, we can upper bound $\sum_{\nu \in \gamma_P\left(\frac{1}{2\varepsilon n}\right)} \frac{1}{\varepsilon n}$ by $(|\gamma_P\left(\frac{1}{2\varepsilon n}\right)| - 1)\frac{1}{\varepsilon n}$ by dealing with the $|\gamma_P\left(\frac{1}{2\varepsilon n}\right)| = 1$ case separately. Therefore,

$$\mathcal{W}(P, \hat{P}_\varepsilon)$$

$$\leq 2C\ln(n/\beta) \left( \sum_\nu \min\left\{ \mathfrak{G}_P(\nu)(1 - \mathfrak{G}_P(\nu)), \sqrt{\frac{\mathfrak{G}_P(\nu)(1 - \mathfrak{G}_P(\nu))}{n}} \right\} + \sum_{\nu \notin \gamma_P\left(\frac{1}{2\varepsilon n}\right)} \mathfrak{G}_P(\nu) + (|\gamma_P\left(\frac{1}{2\varepsilon n}\right)| - 1)\frac{1}{\varepsilon n} \right),$$

Further, by Theorem D.7 and Theorem D.6, given $\varepsilon > 0$ and $\delta \in [0, 1]$, let $\kappa = \frac{1}{10\varepsilon n}\min\{W\left(\frac{0.45\varepsilon}{\delta}\right), 0.6\}$ where $W(x)$ is the Lambert W function so $W(x)e^{W(x)} = x$. Given a distribution $P$, there exists a constant $C'$ such that

$$\mathcal{R}_{\mathcal{N},n,\varepsilon}(P) \geq \frac{C'}{D_T} \left( \sum_\nu \min\left\{ \mathfrak{G}_P(\nu)(1 - \mathfrak{G}_P(\nu)), \sqrt{\frac{\mathfrak{G}_P(\nu)(1 - \mathfrak{G}_P(\nu))}{n}} \right\} + \sum_{\nu \notin \gamma_P(2\kappa)} \mathfrak{G}_P(\nu) + (|\gamma_P(2\kappa)| - 1)\kappa \right)$$

Let $Q \in \mathcal{N}(P)$, then $\gamma_P\left(\frac{1}{\varepsilon n}\right) \subset \gamma_Q\left(\frac{1}{2\varepsilon n}\right) \subset \gamma_P\left(\frac{1}{4\varepsilon n}\right)$ so

$$\sum_{\nu \notin \gamma_Q\left(\frac{1}{2\varepsilon n}\right)} \mathfrak{G}_Q(\nu) + (|\gamma_Q\left(\frac{1}{2\varepsilon n}\right)| - 1)\frac{1}{\varepsilon n} = \sum_{\nu \notin \gamma_P\left(\frac{1}{4\varepsilon n}\right)} \mathfrak{G}_Q(\nu) + \sum_{\nu \in \gamma_P\left(\frac{1}{4\varepsilon n}\right)\backslash\gamma_Q\left(\frac{1}{2\varepsilon n}\right)} \mathfrak{G}_Q(\nu) + (|\gamma_Q\left(\frac{1}{2\varepsilon n}\right)| - 1)\frac{1}{\varepsilon n}$$

$$\leq \sum_{\nu \notin \gamma_P\left(\frac{1}{4\varepsilon n}\right)} 2\mathfrak{G}_P(\nu) + \sum_{\nu \in \gamma_P\left(\frac{1}{4\varepsilon n}\right)\backslash\gamma_Q\left(\frac{1}{2\varepsilon n}\right)} \frac{1}{\varepsilon n} + (|\gamma_Q\left(\frac{1}{2\varepsilon n}\right)| - 1)\frac{1}{\varepsilon n}$$

$$\leq \sum_{\nu \notin \gamma_P\left(\frac{1}{4\varepsilon n}\right)} 2\mathfrak{G}_P(\nu) + (|\gamma_P\left(\frac{1}{4\varepsilon n}\right)| - 1)\frac{1}{\varepsilon n}.$$

Now, let $n' = \frac{5}{4\min\{W(\frac{0.45\varepsilon}{\delta}),0.6\}}n \geq n$ so for all $Q \in \mathcal{N}(P)$,

$$\mathcal{W}(Q, \hat{Q_{\varepsilon,n'}})$$

$$\leq \tilde{O}\left(\sum_\nu \min\left\{\mathfrak{G}_Q(\nu)(1-\mathfrak{G}_Q(\nu)), \sqrt{\frac{\mathfrak{G}_Q(\nu)(1-\mathfrak{G}_Q(\nu))}{n'}}\right\} + \sum_{\nu \notin \gamma_Q\left(\frac{1}{2\varepsilon n'}\right)} \mathfrak{G}_Q(\nu) + (|\gamma_Q\left(\frac{1}{2\varepsilon n'}\right)|-1)\frac{1}{\varepsilon n'}\right)$$

$$\leq \tilde{O}\left(\sum_\nu 2\min\left\{\mathfrak{G}_P(\nu)(1-\mathfrak{G}_P(\nu)), \sqrt{\frac{\mathfrak{G}_P(\nu)(1-\mathfrak{G}_P(\nu))}{n'}}\right\} + \sum_{\nu \notin \gamma_P\left(\frac{1}{4\varepsilon n'}\right)} 2\mathfrak{G}_P(\nu) + (|\gamma_P\left(\frac{1}{4\varepsilon n'}\right)|-1)\frac{1}{\varepsilon n'}\right)$$

$$= \tilde{O}\left(\sum_\nu 2\min\left\{\mathfrak{G}_P(\nu)(1-\mathfrak{G}_P(\nu)), \sqrt{\frac{\mathfrak{G}_P(\nu)(1-\mathfrak{G}_P(\nu))}{n}}\right\} + \sum_{\nu \notin \gamma_P(2\kappa)} 2\mathfrak{G}_P(\nu) + (|\gamma_P(2\kappa)|-1)\frac{1}{\varepsilon n}\right)$$

$$\leq \tilde{O}\left(\min_{\mathcal{A}'} \max_{Q' \in \mathcal{N}(P)} \mathbb{E}_{X \sim Q'^n, \mathcal{A}'}(\mathcal{W}(\mathcal{A}'(X), Q'))\right).$$

As in Proposition B.5, since $\mathcal{N}(P)$ is compact, for all $\mathcal{A}'$, there exists a specific $Q^* \in \mathcal{N}(P)$ such that

$$\mathcal{W}(Q^*, \hat{Q^*_{\varepsilon,n'}}) \leq \tilde{O}(\mathbb{E}_{X \sim (Q^*)^n, \mathcal{A}'}(\mathcal{W}(\mathcal{A}'(X), Q^*)))$$

$\square$

# I   Differentially Private Quantiles

Estimating appropriately chosen quantiles is the main part of our algorithm for approximating the distribution over $\mathbb{R}$ in Wasserstein distance, and so in this section, we describe some known differentially private algorithms for this task and derive some corollaries that we use extensively in our application. We will use $F$ to represent CDF functions, with $F_P$ representing the CDF of distribution $P$. We start by stating an important theorem on private CDF estimation. This follows from a use of the binary tree mechanism [CSS11, DNPR10]. A version of this theorem for approximate differential privacy is described in a survey by Kamath and Ullman [KU20, Theorem 4.1]. The version presented here for pure differential privacy follows from a very similar argument, except using Laplace Noise instead of Gaussian noise (and basic composition instead of advanced composition to analyze privacy). Their accuracy was also in expectation, but a similar analysis yields a high probability bound, as in the theorem below.

**Theorem I.1.** *[KU20, Theorem 4.1]*

*Let $\varepsilon, \beta \in (0,1]$, let $D$ be an ordered, finite domain, and let $\mathbf{x} \in D^n$ be a dataset. Let $\hat{P}_n$ be the uniform distribution on $\mathbf{x}$. Then, there exists an $\varepsilon$-DP algorithm $A^{CDF}$ that on input $\mathbf{x}$ and the domain $D$ outputs a vector $G$ over $D$ such that with probability at least $1 - \beta$ over the randomness of $A^{CDF}$:*

$$\|G - F_{\hat{P}_n}\|_\infty = O\left(\frac{\log^3 \frac{|D|}{\beta}}{\varepsilon n}\right).$$

CDF estimation is intimately related to quantile estimation, and we use the following quantitative statement that will follow from a simple application of Theorem I.1.

**Theorem I.2.** *Fix any $n > 0$, $\varepsilon, \beta \in (0,1]$, $a, b \in \mathbb{R}$, and $\gamma < b - a \in \mathbb{R}$ such that $\frac{b-a}{\gamma}$ is an integer. Let $C$ be a sufficiently large constant. Then, there exists an $\varepsilon$-DP algorithm $A_{quant}$, that on input interval end points $a, b$, granularity $\gamma$, $\mathbf{x} = (x_1, \ldots, x_n) \in \{a, a + \gamma, \ldots, b - \gamma, b\}^n$, and desired quantile values $\alpha \in (0,1)^k$, outputs quantiles $\tilde{q} \in \{a, a + \gamma, \ldots, b - \gamma, b\}^k$ such that with probability at least $1 - \beta$ over the randomness of $A_{quant}$, for all $r \in [k]$,*

$$\alpha_r - F_{\hat{P}_n}(\tilde{q}_r) \leq C\frac{\log^3 \frac{b-a}{\beta\gamma}}{\varepsilon n},$$

*and*

$$\Pr_{y \sim \hat{P}_n} (y < \tilde{q}_r) < \alpha_r + C \frac{\log^3 \frac{b-a}{\beta\gamma}}{\varepsilon n},$$

*where $\hat{P}_n$ is the uniform distribution on the entries of $\mathbf{x}$.*

*Proof.* Algorithm $A_{quant}$ operates by running the algorithm $A^{CDF}$ referenced in Theorem I.1 on $\mathbf{x}$ and domain $\{a, a + \gamma, \ldots, b - \gamma, b\}$, and postprocessing its outputs to get quantile estimates as follows. For every quantile $\alpha_r$ that we are asked to estimate, $A_{quant}$ simply scans the vector $G$ output by algorithm $A^{CDF}$ in order, and outputs the first domain element whose CDF estimate in $G$ crosses $\alpha_r$. Conditioned on the accuracy of the CDF estimation algorithm $G$, we have that this output $\tilde{q}_r$ satisfies

$$\alpha_r - F_{\hat{P}_n}(\tilde{q}_r) \leq C \frac{\log^3 \frac{b-a}{\beta\gamma}}{\varepsilon n}.$$

Additionally, since $\tilde{q}_r$ is the first domain element whose estimate in $G$ crosses $\alpha_r$, we also have that

$$Pr_{y \sim \hat{P}_n}(y < \tilde{q}_r) < \alpha_r + C \frac{\log^3 \frac{b-a}{\beta\gamma}}{\varepsilon n}.$$

Hence, with probability at least $1 - \beta$, we have this property for all $r \in [k]$. $\qquad\square$

We now state a corollary of this theorem that we will use extensively in our presentation.

**Corollary I.3.** *Fix any $\varepsilon, \beta \in (0,1]$, $a, b \in \mathbb{R}$, and $\gamma < b - a \in \mathbb{R}$ such that $\frac{b-a}{\gamma}$ is an integer. Let $n \in \mathbb{N} > \frac{4c_2 \log^4\left(\frac{b-a}{\beta\gamma\varepsilon}\right)}{\varepsilon}$, such that $k$ set to $\lceil \frac{\varepsilon n}{4c_3 \log^3 \frac{b-a}{\beta\gamma} \log \frac{n}{\beta}} \rceil$ is an integer greater than or equal to 1, where $c_2$ and $c_3$ are sufficiently large constants. [6]*

*Then, there exists an $\varepsilon$-DP algorithm $A_{quant}$ (the same one referenced in Theorem I.2), that on input interval end points $a, b$, granularity $\gamma$, $\mathbf{x} = (x_1, \ldots, x_n) \in \{a, a + \gamma, \ldots, b - \gamma, b\}^n$, and desired quantile values $\alpha = \{1/2k, 3/2k, 5/2k, \ldots, (2k-1)/2k\}$, outputs quantiles $\tilde{q} \in \{a, a+\gamma, \ldots, b - \gamma, b\}^k$ such that with probability at least $1 - \beta$, for all $r \in [k]$,*

$$\hat{q}_{\frac{2r-1}{2k} - \frac{1}{4k}} \leq \tilde{q}_r \leq \hat{q}_{\frac{2r-1}{2k} + \frac{1}{4k}},$$

*where $\hat{P}_n$ is the uniform distribution on the entries of $\mathbf{x}$ and for all $p \in (0,1)$, $\hat{q}_p$ is the $p$-quantile of $\hat{P}_n$.*

*Proof.* First, note that $k$ is set such that $\frac{1}{4k} \geq C \frac{log^3 \frac{b-a}{\beta\gamma}}{\varepsilon n}$.

Hence, by Theorem I.2, we have that with probability at least $0.99$,

for all $r \in [k]$,

$$\frac{2r - 1}{2k} - F_{\hat{P}_n}(\tilde{q}_r) \leq \frac{1}{4k},$$

and

$$\Pr_{y \sim \hat{P}_n}(y < \tilde{q}_r) < \frac{2r - 1}{2k} + \frac{1}{4k}.$$

Condition the event above for the rest of the proof. Note that the first equation implies that for all $r \in [k]$,

$$\Pr_{y \sim \hat{P}_n}(y \leq \tilde{q}_r) \geq \frac{2r - 1}{2k} - \frac{1}{4k},$$

which implies that $\tilde{q}_r \geq \hat{q}_{\frac{2r-1}{2k} - \frac{1}{4k}}$.

---

[6] $k$ is set to be sufficiently small in order to relate the accuracy of the quantiles algorithm to a parameter depending on $k$, and $n$ is set sufficiently large that $k$ is not less than $1$. The dependence on $\beta$ comes up in the proof of Claim E.18.

Next, note that we also have that for all $r \in [k]$,

$$\Pr_{y \sim \hat{P}_n} (y < \tilde{q}_r) < \frac{2r-1}{2k} + \frac{1}{4k}.$$

This implies that for all $r \in [k]$, $\tilde{q}_r \leq \hat{q}_{\frac{2r-1}{2k} + \frac{1}{4k}}$. $\qquad\square$

## J  Proofs in Section E

### J.1  Omitted Proofs in Section E.1.1

*Proof of Lemma E.6.* We evaluate the various terms in Theorem E.5.

We start by evaluating $\tau(P,Q) = \max\{\int_{\mathbb{R}} \max(f_P(t) - e^\varepsilon f_Q(t), 0)dt, \int_{\mathbb{R}} \max(f_Q(t) - e^\varepsilon f_P(t), 0)dt \}$. Consider the first term in the outer maximum. For all $t \in [L(P), q_{1/k})$, we have that $f_Q(t) = \frac{1}{2}f_P(t)$. For all other $t$, one can see that the value of the integrand is $0$. Hence, the value of the first term is $\int_{L(P)}^{q_{1/k}} \max(f_P(t) - \frac{e^\varepsilon}{2}f_P(t), 0)dt = \max\{\left(1 - \frac{e^\varepsilon}{2}\right)\frac{1}{k}, 0\} \leq \frac{1}{2k}$. Now, consider the second term in the outer maximum. For all $t < q_{1-\frac{1}{k}}$, the value of the integrand is $0$. For all $q_{1-\frac{1}{k}} \leq t \leq q_1$, the value of the integrand is $\max\{\left(\frac{3}{2} - e^\varepsilon\right) f_P(t), 0\}$. Hence, the second term is $\max\{\left(\frac{3}{2} - e^\varepsilon\right)\frac{1}{k}, 0\} \leq \frac{1}{2k}$. Put together, we get that $\tau(P,Q) \leq \frac{1}{2k}$.

When $\varepsilon \geq \ln 2$, we have that $1 - \frac{e^\varepsilon}{2} \leq 0$, and so we have that the largest value of $\varepsilon' \in [0, \varepsilon]$ that makes $\int_{\mathbb{R}} \max(f_Q(t) - e^{\varepsilon'} f_P(t), 0)dt = \tau(P,Q) = 0$, is $\varepsilon' = \varepsilon$. When $\varepsilon < \ln 2$, we have that the value of $\varepsilon'$ that makes $\int_{\mathbb{R}} \max(f_Q(t) - e^{\varepsilon'} f_P(t), 0)dt = \max\{\left(\frac{3}{2} - e^\varepsilon\right)\frac{1}{k}, 0\} = \left(1 - \frac{e^\varepsilon}{2}\right)\frac{1}{k}$, is $\varepsilon' = \ln\left(\frac{1+e^\varepsilon}{2}\right)$.

Finally, we describe the distributions $P'$ and $Q'$ and compute the squared Hellinger distance between them. There are two cases, based on the range of $\varepsilon$. First, consider $\varepsilon \geq \ln 2$. First, we calculate $\tilde{P} \equiv \min\{e^\varepsilon Q, P\}$. This value is equal to $\min\{e^\varepsilon/2, 1\}f_P(t) = f_P(t)$ for $t < q_{\frac{1}{k}}(P)$, and is also equal to $f_P(t)$ for $q_{\frac{1}{k}} \leq t \leq q_1$. Similarly, consider $\tilde{Q} \equiv \min\{e^{\varepsilon'} P, Q\} = \min\{e^\varepsilon P, Q\}$; it is equal to $\frac{f_P(t)}{2}$ for $t < q_{\frac{1}{k}}(P)$, and is equal to $f_P(t)$ for $q_{\frac{1}{k}} \leq t \leq q_{1-\frac{1}{k}}$. It is also equal to $\min(e^\varepsilon, \frac{3}{2})f_P(t) = \frac{3}{2}f_P(t)$ for $q_{\frac{1}{k}} \leq t \leq q_1$. Since $\tau(P,Q) = 0$, and by the above calculations, we have that $P' = P$, and $Q' = Q$. Upper bounding the squared Hellinger distance between $P'$ and $Q'$ by the TV distance (See Lemma A.7), we get that $H^2(P', Q') = H^2(P,Q) \leq TV(P,Q) = \frac{1}{2k} \leq \frac{\varepsilon}{2(\ln 2)k}$ (where we have used that $\varepsilon \geq \ln 2$).

Next, consider $\varepsilon < \ln 2$. First, consider $\tilde{P} \equiv \min\{e^\varepsilon Q, P\}$. This value is equal to $\min\{e^\varepsilon/2, 1\}f_P(t) = \frac{e^\varepsilon}{2}f_P(t)$ for $t < q_{\frac{1}{k}}(P)$, and is also equal to $f_P(t)$ for $q_{\frac{1}{k}} \leq t \leq q_1$. Similarly, consider $\tilde{Q} \equiv \min\{e^{\varepsilon'} P, Q\} = \min\{\frac{1+e^\varepsilon}{2}P, Q\}$; it is equal to $\frac{1}{2}f_P(t)$ for $t < q_{\frac{1}{k}}(P)$, and is equal to $f_P(t)$ at $q_{\frac{1}{k}} \leq t \leq q_{1-\frac{1}{k}}$. It is also equal to $\min\{\frac{1+e^\varepsilon}{2}, \frac{3}{2}\}f_P(t) = \frac{1+e^\varepsilon}{2}f_P(t)$ at $q_{1-\frac{1}{k}} \leq t \leq q_1$. Note that $\tau(P,Q) = \left(1 - \frac{e^\varepsilon}{2}\right)\frac{1}{k}$. $P'$ and $Q'$ are the distributions created by normalizing $\tilde{P}$ and $\tilde{Q}$ by dividing by a factor of $1 - \tau(P,Q)$. Now, we upper bound the squared Hellinger distance between $P'$ and $Q'$ by the TV distance (See Lemma A.7), to get that $H^2(P', Q') \leq TV(P', Q') = O(\frac{\varepsilon}{k})$.

Substituting into the lower bound for sample complexity of distinguishing $P$ and $Q$, this tells us that for all $\varepsilon \in (0, 1]$, $SC_\varepsilon(P,Q) = \Omega\left(\frac{1}{\varepsilon \cdot \frac{1}{k}}\right) = \Omega(k/\varepsilon)$.

$\qquad\square$

*Proof of Lemma E.7.* Note that $P$ has bounded expectation (and hence, so does $Q$). Hence, we can use the following form of the Wasserstein distance:

$$\mathcal{W}(P,Q) = \int_{\mathbb{R}} |F_P(t) - F_Q(t)|dt.$$

Now, given the settings of $P$ and $Q$, we can precisely write the forms of their cumulative distribution function as follows. Note that for $L(P) \leq t < q_{1/k}(P)$, we have that $|F_P(t) - F_Q(t)| = \frac{1}{2}F_p(t)$. For $q_{1/k} \leq t \leq q_{1-\frac{1}{k}}$, we have $|F_P(t) - F_Q(t)| = \frac{1}{2k}$. Finally, for $q_{1-\frac{1}{k}} \leq t \leq q_1$, we have that $F_P(t) = 1 - \frac{1}{k} + \int_{q_{1-1/k}}^t f_P(t)dt$ and $F_Q(t) = 1 - \frac{3}{2k} + \frac{3}{2}\int_{q_{1-1/k}}^t f_P(t)dt$, which gives us that $F_P(t) - F_Q(t) = \frac{1}{2k} - \frac{1}{2}\int_{q_{1-1/k}}^t f_P(t)dt = \frac{1}{2}[1 - F_P(t)]$.

Hence, we have that

$$
\begin{aligned}
\mathcal{W}(P,Q) &= \int_{\mathbb{R}} |F_P(t) - F_Q(t)|dt \\
&= \frac{1}{2}\int_{L(P)}^{q_{1/k}} F_P(t)dt + \int_{q_{1/k}}^{q_{1-\frac{1}{k}}} |F_P(t) - F_Q(t)|dt + \int_{q_{1-\frac{1}{k}}}^{q_1} |F_P(t) - F_Q(t)|dt \\
&\geq \frac{1}{2}\int_{L(P)}^{q_{1/k}} F_P(t)dt + \frac{1}{2}\int_{q_{1-\frac{1}{k}}}^{q_1} [1 - F_P(t)]dt + \frac{1}{2k}(q_{1-\frac{1}{k}} - q_{\frac{1}{k}}) \\
&= \frac{1}{2}\int_{q_{1-\frac{1}{k}}}^{q_1} \left|F_P(t) - F_{P|_{q_{\frac{1}{k}},q_{1-\frac{1}{k}}}}(t)\right|dt + \frac{1}{2}\int_{L(P)}^{q_{1/k}} \left|F_P(t) - F_{P|_{q_{\frac{1}{k}},q_{1-\frac{1}{k}}}}(t)\right|dt + \frac{1}{2k}(q_{1-\frac{1}{k}} - q_{\frac{1}{k}}) \\
&= \frac{1}{2k}(q_{1-\frac{1}{k}} - q_{1/k}) + \frac{1}{2}\mathcal{W}(P,P|_{q_{\frac{1}{k}},q_{1-\frac{1}{k}}})
\end{aligned}
$$

$\square$

## J.2 Omitted proofs in Section E.1.2

*Proof of Lemma E.10.* The KL divergence is defined as $\int_{t:f_Q(t)>0} f_P(t)\log f_P(t)/f_Q(t)dt$. This can be broken up into a sum over the dyadic quantiles as:

$$
\begin{aligned}
KL(P,Q) &= \sum_{i=2}^{\log n - 1} \int_{q_{1/2^i}}^{q_{1/2^{i-1}}} f_P(t)\log\frac{f_P(t)}{f_Q(t)}dt + \int_{q_{1-1/2^{i-1}}}^{q_{1-1/2^i}} f_P(t)\log\frac{f_P(t)}{f_Q(t)}dt \\
&\quad + \sum_{i=\log n}^{\infty} \int_{q_{1/2^i}}^{q_{1/2^{i-1}}} f_P(t)\log\frac{f_P(t)}{f_Q(t)}dt + \int_{q_{1-1/2^{i-1}}}^{q_{1-1/2^i}} f_P(t)\log\frac{f_P(t)}{f_Q(t)}dt \\
&= \sum_{i=2}^{\log n - 1} \int_{q_{1/2^i}}^{q_{1/2^{i-1}}} f_P(t)\log\frac{1}{1+\sqrt{\frac{2^i}{n}}}dt + \int_{q_{1-1/2^{i-1}}}^{q_{1-1/2^i}} f_P(t)\log\frac{1}{1-\sqrt{\frac{2^i}{n}}}dt \\
&\quad + \sum_{i=\log n}^{\infty} \int_{q_{1/2^i}}^{q_{1/2^{i-1}}} f_P(t)\log\frac{1}{1+\frac{1}{2}}dt + \int_{q_{1-1/2^{i-1}}}^{q_{1-1/2^i}} f_P(t)\log\frac{1}{1-\frac{1}{2}}dt \\
&= \sum_{i=2}^{\log 4n} \frac{1}{2^i}\left[\log\frac{1}{1+\sqrt{\frac{2^i}{n}}} + \log\frac{1}{1-\sqrt{\frac{2^i}{n}}}\right] + \sum_{i=\log n}^{\infty} \frac{1}{2^i}\left[\log\frac{1}{1+\frac{1}{2}} + \log\frac{1}{1-\frac{1}{2}}\right] \\
&\leq \sum_{i=2}^{\log n - 1} \frac{1}{2^i}\log\frac{1}{1-\frac{2^i}{n}} + O\left(\frac{1}{n}\right) \\
&\leq \sum_{i=2}^{\log n - 1} \frac{1}{2^i}2\frac{2^i}{n} + O\left(\frac{1}{n}\right) \\
&= O\left(\frac{\log n}{n}\right),
\end{aligned}
$$

where the third inequality from last is by the fact that the geometric series $\sum_{i=\log n}^{\infty} \frac{1}{2^i}$ converges to $O(\frac{1}{n})$, the second inequality from last is from the fact that $\frac{2^i}{n} < 1/2$, and $\log(1/(1-y)) < 2y$ for $0 < y < 1/2$. $\square$

*Proof of Lemma E.11.* First, we recall the definition of the 1-Wasserstein distance in terms of the cumulative distribution function.

$$\mathcal{W}(P,Q) = \int_{\mathbb{R}} |F_P(t) - F_Q(t)| dt$$

Fix any $2 \leq i < \log n - 1$. Observe that by construction, for all $t \in [q_{1/2^i}, q_{1-1/2^i})$ and for all $t \in [q_{1-1/2^{i-1}}, q_{1-1/2^i})$, $|F_P(t) - F_Q(t)| \geq \sum_{j=i+1}^{\log n-1} \frac{1}{\sqrt{2^j n}} + \frac{1}{2} \sum_{j=\log n}^{\infty} \frac{1}{2^j}$. Similarly, fix any $\log n - 1 \leq i < \infty$. Observe that for all $t \in [q_{1/2^i}, q_{1-1/2^i})$, and for all $t \in [q_{1-1/2^{i-1}}, q_{1-1/2^i})$, we have that $|F_P(t) - F_Q(t)| \geq \frac{1}{2} \sum_{j=i+1}^{\infty} \frac{1}{2^j}$. Substituting the above bounds in the formula for the Wasserstein distance, we get that

$$\mathcal{W}(P,Q) \geq \sum_{i=2}^{\log n-2} \int_{q_{1/2^i}}^{q_{1/2^{i-1}}} \left[ \sum_{j=i+1}^{\log n-2} \frac{1}{\sqrt{2^j n}} + \frac{1}{2} \sum_{j=\log n-1}^{\infty} \frac{1}{2^j} \right] dt + \int_{q_{1-1/2^{i-1}}}^{q_{1-1/2^i}} \left[ \sum_{j=i+1}^{\log n-2} \frac{1}{\sqrt{2^j n}} + \frac{1}{2} \sum_{j=\log n-1}^{\infty} \frac{1}{2^j} \right] dt$$

$$+ \sum_{i=\log n-1}^{\infty} \int_{q_{1/2^i}}^{q_{1/2^{i-1}}} \frac{1}{2} \sum_{j=i+1}^{\infty} \frac{1}{2^j} dt + \int_{q_{1-1/2^{i-1}}}^{q_{1-1/2^i}} \frac{1}{2} \sum_{j=i+1}^{\infty} \frac{1}{2^j} dt$$

Pulling the summation over $j$ outside the integral and grouping terms,

$$\mathcal{W}(P,Q) \geq \sum_{i=2}^{\log n-2} \left[ \sum_{j=i+1}^{\log n-2} \int_{q_{1/2^i}}^{q_{1/2^{i-1}}} \frac{1}{\sqrt{2^j n}} dt + \int_{q_{1-1/2^{i-1}}}^{q_{1-1/2^i}} \frac{1}{\sqrt{2^j n}} dt + \frac{1}{2} \sum_{j=\log n-1}^{\infty} \int_{q_{1/2^i}}^{q_{1/2^{i-1}}} \frac{1}{2^j} dt + \int_{q_{1-1/2^{i-1}}}^{q_{1-1/2^i}} \frac{1}{2^j} dt \right]$$

$$+ \frac{1}{2} \sum_{i=\log n-1}^{\infty} \sum_{j=i+1}^{\infty} \left[ \int_{q_{1/2^i}}^{q_{1/2^{i-1}}} \frac{1}{2^j} dt + \int_{q_{1-1/2^{i-1}}}^{q_{1-1/2^i}} \frac{1}{2^j} dt \right]$$

$$= \sum_{i=2}^{\log n-2} \left[ (q_{1/2^{i-1}} - q_{1/2^i}) + (q_{1-1/2^i} - q_{1-1/2^{i-1}}) \right] \left[ \sum_{j=i+1}^{\log n-2} \frac{1}{\sqrt{2^j n}} + \frac{1}{2} \sum_{j=\log n-1}^{\infty} \frac{1}{2^j} \right]$$

$$+ \sum_{i=\log n-1}^{\infty} \left[ (q_{1/2^{i-1}} - q_{1/2^i}) + (q_{1-1/2^i} - q_{1-1/2^{i-1}}) \right] \frac{1}{2} \sum_{j=i+1}^{\infty} \frac{1}{2^j}$$

Switching the order of summation (summing over $j$ first), and grouping terms, we get

$$\mathcal{W}(P,Q) \geq \sum_{j=3}^{\log n-2} \frac{1}{\sqrt{2^j n}} \sum_{i=2}^{j-1} \left[ (q_{1/2^{i-1}} - q_{1/2^i} + (q_{1-1/2^i} - q_{1-1/2^{i-1}})) \right]$$

$$+ \frac{1}{2} \sum_{j=\log n-1}^{\infty} \frac{1}{2^j} \sum_{i=2}^{j-1} \left[ (q_{1/2^{i-1}} - q_{1/2^i} + (q_{1-1/2^i} - q_{1-1/2^{i-1}})) \right]$$

Telescoping the inner sums over $i$ we get that

$$\mathcal{W}(P,Q) \geq \sum_{j=3}^{\log n-2} \frac{1}{\sqrt{2^j n}} \left[ q_{1-1/2^{j-1}} - q_{1/2^{j-1}} \right] + \frac{1}{2} \sum_{j=\log n-1}^{\infty} \frac{1}{2^j} \left[ q_{1-1/2^{j-1}} - q_{1/2^{j-1}} \right]$$

A change of variables (where we now set $j$ to $j-1$) then gives

$$\mathcal{W}(P,Q) \geq \frac{1}{\sqrt{2}} \sum_{j=2}^{\log n-3} \frac{1}{\sqrt{2^j n}} \left[ q_{1-1/2^j} - q_{1/2^j} \right] + \frac{1}{4} \sum_{j=\log n-2}^{\infty} \frac{1}{2^j} \left[ q_{1-1/2^j} - q_{1/2^j} \right]$$

$$\geq \frac{1}{4} \sum_{j=2}^{\log n-1} \frac{1}{\sqrt{2^j n}} \left[ q_{1-1/2^j} - q_{1/2^j} \right] + \frac{1}{4} \sum_{j=\log n}^{\infty} \frac{1}{2^j} \left[ q_{1-1/2^j} - q_{1/2^j} \right],$$

where the last inequality is by pulling the first two terms from the summation in second term to the summation in the first term, and using the fact that for $j = \log n - 2, j = \log n - 1$, we have that $\frac{1}{4 \cdot 2^j} \geq \frac{1}{2\sqrt{2}} \frac{1}{\sqrt{2^j n}}$

$\square$

*Proof of Lemma E.12.* We first state a theorem of Bobkov and Ledoux [BL19].

**Theorem J.1** (Theorem 3.5, [BL19]). *There is an absolute constant $c > 0$, such that for all distributions $P$ over $\mathbb{R}$, for every $n \geq 1$,*

$$c(A_n + B_n) \leq \mathbb{E}[\mathcal{W}(P, \hat{P}_n)] \leq A_n + B_n.$$

*where*

$$A_n = 2 \int_{F(t)[1-F(t)] \leq \frac{1}{4n}} F(t)[1 - F(t)]dt,$$

*and*

$$B_n = \frac{1}{\sqrt{n}} \int_{F(t)[1-F(t)] \geq \frac{1}{4n}} \sqrt{F(t)[1 - F(t)]}dt.$$

Now, we are ready to prove the main theorem. Fix natural number $i \geq 2$. Restricted to $t \leq q_{1/2}$, $F_P(t)(1 - F_P(t))$ is an increasing function, and hence for $t \in [q_{1/2^i}, q_{1/2^{i-1}}]$, we have that $F_P(t)(1 - F_P(t)) \leq \frac{1}{2^{i-1}}[1 - \frac{1}{2^{i-1}}]$.

Similarly, restricted to $t > q_{1/2}$, $F_P(t)(1 - F_P(t))$ is a decreasing function, and hence for $t \in [1 - q_{1/2^{i-1}}, q_{1-1/2^i}]$, we have that $F_P(t)(1 - F_P(t)) \leq \frac{1}{2^{i-1}}[1 - \frac{1}{2^{i-1}}]$.

Using this, we can now upper bound the expected Wasserstein distance between $P$ and its empirical distribution using Theorem J.1. Hence, we upper bound the terms $B_n$ and $A_n$. We start by upper bounding $B_n$. Note that for all $t \notin [q_{\frac{1}{4n}}, q_{1-\frac{1}{4n}}]$, we have that $F_P(t)(1 - F_P(t)) \leq \frac{1}{4n}$. Hence,

$$
\begin{aligned}
B_n &= \frac{1}{\sqrt{n}} \int_{F_P(t)[1-F_P(t)] \geq \frac{1}{4n}} \sqrt{F_P(t)[1 - F_P(t)]}dt \\
&\leq \frac{1}{\sqrt{n}} \int_{q_{\frac{1}{4n}}}^{q_{1-\frac{1}{4n}}} \sqrt{F_P(t)[1 - F_P(t)]}dt \\
&\leq \sum_{i=2}^{\log 4n} \frac{1}{\sqrt{n}} \left[ \int_{q_{1/2^i}}^{q_{1/2^{i-1}}} \sqrt{F_P(t)[1 - F_P(t)]}dt + \int_{q_{1-1/2^{i-1}}}^{q_{1-1/2^i}} \sqrt{F_P(t)[1 - F_P(t)]}dt \right] \\
&\leq \sum_{i=2}^{\log 4n} \frac{1}{\sqrt{n}} \int_{q_{1/2^i}}^{q_{1/2^{i-1}}} \sqrt{\frac{1}{2^{i-1}}\left[1 - \frac{1}{2^{i-1}}\right]}dt + \int_{q_{1-1/2^{i-1}}}^{q_{1-1/2^i}} \sqrt{\frac{1}{2^{i-1}}\left[1 - \frac{1}{2^{i-1}}\right]}dt \\
&= \sum_{i=2}^{\log 4n} \frac{1}{\sqrt{n}} \sqrt{\frac{1}{2^{i-1}}\left[1 - \frac{1}{2^{i-1}}\right]} \left[q_{1/2^{i-1}} - q_{1/2^i} + q_{1-1/2^i} - q_{1-1/2^{i-1}}\right] \\
&\leq \sum_{i=2}^{\log 4n} \frac{2}{\sqrt{2^i n}} \left[q_{1-1/2^i} - q_{1/2^i}\right] \\
&= \sum_{i=2}^{\log n-1} \frac{2}{\sqrt{2^i n}} \left[q_{1-1/2^i} - q_{1/2^i}\right] + \sum_{i=\log n}^{\log 4n} \frac{2}{\sqrt{2^i n}} \left[q_{1-1/2^i} - q_{1/2^i}\right] \\
&\leq \sum_{i=2}^{\log n-1} \frac{2}{\sqrt{2^i n}} \left[q_{1-1/2^i} - q_{1/2^i}\right] + \sum_{i=\log n}^{\log 4n} \frac{4}{2^i} \left[q_{1-1/2^i} - q_{1/2^i}\right],
\end{aligned}
$$

where the last inequality is because for $i \leq \log(4n)$, we have that $\frac{1}{n} \leq \frac{4}{2^i}$.

Next, we bound $A_n$. Note that for all $t \geq q_{1/2n}$ and for all $t \leq q_{1-\frac{1}{2n}}$, we have that $F_P(t)(1 - F_P(t)) \not\leq \frac{1}{4n}$. Hence,

$$A_n = 2 \int_{F_P(t)[1-F_P(t)] \le \frac{1}{4n}} F_P(t)[1 - F_P(t)] dt$$

$$\le 2 \left[ \int_{-\infty}^{q_{\frac{1}{2n}}} F_P(t)[1 - F_P(t)] dt + \int_{q_{1-\frac{1}{2n}}}^{\infty} F_P(t)[1 - F_P(t)] dt \right]$$

$$= \sum_{i=1+\log 2n}^{\infty} 2 \left[ \int_{q_{1/2^i}}^{q_{1/2^{i-1}}} F_P(t)[1 - F_P(t)] dt + \int_{q_{1-1/2^{i-1}}}^{q_{1-1/2^i}} F_P(t)[1 - F_P(t)] dt \right]$$

$$\le \sum_{i=1+\log 2n}^{\infty} 2 \left[ \int_{q_{1/2^i}}^{q_{1/2^{i-1}}} \frac{1}{2^{i-1}} \left[ 1 - \frac{1}{2^{i-1}} \right] dt + \int_{q_{1-1/2^{i-1}}}^{q_{1-1/2^i}} \frac{1}{2^{i-1}} \left[ 1 - \frac{1}{2^{i-1}} \right] dt \right]$$

$$= \sum_{i=1+\log 2n}^{\infty} \frac{2}{2^{i-1}} \left[ 1 - \frac{1}{2^{i-1}} \right] \left[ q_{1/2^{i-1}} - q_{1/2^i} + q_{1-1/2^i} - q_{1-1/2^{i-1}} \right]$$

$$\le \sum_{i=1+\log 2n}^{\infty} \frac{4}{2^i} \left[ q_{1-1/2^i} - q_{1/2^i} \right]$$

Then, using the upper bound in Theorem J.1, substituting in the bounds for $A_n$ and $B_n$, and simplifying, we get the claim. □

*Proof of Claim E.13.* By the definition of Wasserstein distance and restrictions of distributions, we have that

$$\mathcal{W}(\hat{P}_n|_{q_{\frac{1}{k}},q_{1-\frac{1}{k}}}, P|_{q_{\frac{1}{k}},q_{1-\frac{1}{k}}}) = \int_a^b \left| F_{\hat{P}_n|_{q_{\frac{1}{k}},q_{1-\frac{1}{k}}}}(t) - F_{P|_{q_{\frac{1}{k}},q_{1-\frac{1}{k}}}}(t) \right| dt$$

$$= \int_{q_{\frac{1}{k}}}^{q_{1-\frac{1}{k}}} \left| F_{\hat{P}_n|_{q_{\frac{1}{k}},q_{1-\frac{1}{k}}}}(t) - F_{P|_{q_{\frac{1}{k}},q_{1-\frac{1}{k}}}}(t) \right| dt$$

$$= \int_{q_{\frac{1}{k}}}^{q_{1-\frac{1}{k}}} \left| F_{\hat{P}_n}(t) - F_P(t) \right| dt \le \mathcal{W}(P, \hat{P}_n)$$

□

### J.3 Omitted Proofs in Section E.2

Before going into the proofs, we state the standard Chernoff concentration bound that we will use multiple times.

**Theorem J.2** (Binomial Concentration). *Let $X \sim Bin(n, p)$ with expectation $\mu = np$, and $0 < \delta < 1$. Then,*

$$\Pr(|X - \mu| \ge \delta\mu) \le 2e^{\frac{-\delta^2 \mu}{3}}.$$

*Proof of Lemma E.16.*

$$\mathcal{W}(P, P^{DP}) = \int_t |F_P(t) - F_{P^{DP}}(t)| dt \tag{13}$$

$$\le \int_{t=a}^{q_{1/k}} |F_P(t) - F_{P^{DP}}(t)| dt + \int_{t=q_{1/k}}^{q_{1-1/k}} |F_P(t) - F_{P^{DP}}(t)| dt + \int_{t=q_{1-1/k}}^{b} |F_P(t) - F_{P^{DP}}(t)| dt \tag{14}$$

Note that for all $t \in [q_{1/k}, q_{1-1/k}]$, we have that the cumulative distribution functions of $P$ and its restricted version are identical and likewise for $P^{DP}$. Additionally, the cumulative density functions

for the restricted versions of the two distributions are identical to each other outside of this interval. Hence, we can simplify the middle term in the RHS of the inequality above as follows:

$$\int_{t=q_{1/k}}^{q_{1-1/k}(P)} |F_P(t) - F_{PDP}(t)|dt = \mathcal{W}(P|_{q_{\frac{1}{k}},q_{1-\frac{1}{k}}}, P^{DP}|_{q_{\frac{1}{k}},q_{1-\frac{1}{k}}})$$

Next, we reason about the remaining terms.

Consider the term $\int_{t=a}^{q_{1/k}} |F_P(t) - F_{PDP}(t)|dt$. First, condition on the event in Theorem I.3 (on the accuracy of the private quantiles for the empirical distribution), which tells us that with probability at least $1 - \beta$, we have for all $r \in [k]$, that

$$\hat{q}_{\frac{2r-1}{2k}-\frac{1}{4k}} \leq \tilde{q}_{\frac{2r-1}{2k}} \leq \hat{q}_{\frac{2r-1}{2k}+\frac{1}{4k}}, \tag{15}$$

which implies in particular that $\hat{q}_{1/4k} \leq \tilde{q}_{1/2k} \leq \hat{q}_{3/4k}$.

Next, we argue that $\hat{q}_{1/4k} \geq q_{1/8k}$ with high probability. By the definition of quantiles, we have that $Pr_{y \sim P}(y < q_{1/8k}) < \frac{1}{8k}$. The number of entries in the dataset $\mathbf{x}$ less than $q_{1/8k}$ is hence a Binomial with mean less than $\frac{n}{8k}$, and hence, we have by Theorem J.2 (with $\delta$ set to 0.9) that with probability at least $1 - \beta$, the number of entries in the dataset less than $q_{1/8k}$ is at most $1.9\frac{n}{8k} < \frac{n}{4k}$, which means the total mass less than $q_{\frac{1}{8k}}$ in the empirical distribution is less than $\frac{1}{4k}$. This implies that $\hat{q}_{1/4k} \geq q_{1/8k}$ by the definition of quantiles.

Additionally, note that for all $t < q_{1/k}$, $F_P(t) < \frac{1}{k}$. The number of entries in the dataset $\mathbf{x}$ that are less than $q_{1/k}$ is hence a Binomial with success probability less than $\frac{1}{k}$. By Theorem J.2, we can again argue that with probability at least $1 - \beta$, there is a constant $c'$ such that the total mass of the empirical distribution on values less than $q_{1/k}$ is less than $\frac{c'}{k}$. Hence, $q_{1/k} \leq \hat{q}_{c'/k}$. This implies by Equation 15, that $q_{1/k} \leq \tilde{q}_{c/k}$ for some constant $c$. Hence, for all $t < q_{1/k}$, we have that $F_{PDP}(t) \leq \frac{c}{k}$.

Hence, taking a union bound, with probability at least $1 - O(\beta)$,

$$\int_{t=a}^{q_{1/k}} |F_P(t) - F_{PDP}(t)|dt = \int_{t=a}^{\tilde{q}_{1/2k}} |F_P(t) - F_{PDP}(t)|dt + \int_{\tilde{q}_{1/2k}}^{q_{1/k}} |F_P(t) - F_{PDP}(t)|dt$$

$$\leq \int_{t=a}^{\tilde{q}_{1/2k}} |F_P(t) - F_{P|_{q_{\frac{1}{k}},q_{1-\frac{1}{k}}}}(t)|dt + \int_{q_{1/8k}}^{q_{1/k}} |F_P(t) - \frac{c}{k}|dt$$

$$\leq \int_{t=a}^{\tilde{q}_{1/2k}} |F_P(x) - F_{P|_{q_{\frac{1}{k}},q_{1-\frac{1}{k}}}}(t)|dt + \int_{q_{1/8k}}^{q_{1/k}} |F_P(t) - 8cF_P(t)|dt$$

$$\leq (1-8c)\left[\int_{t=a}^{\tilde{q}_{1/2k}} |F_P(t) - F_{P|_{q_{\frac{1}{k}},q_{1-\frac{1}{k}}}}(t)|dt + \int_{q_{1/8k}}^{q_{1/k}} |F_P(t)|dt\right]$$

$$\leq (1-8c)\left[\int_{t=a}^{\tilde{q}_{1/2k}} |F_P(t) - F_{P|_{q_{\frac{1}{k}},q_{1-\frac{1}{k}}}}(t)|dt + \int_{q_{1/8k}}^{q_{1/k}} |F_P(t) - F_{P|_{q_{\frac{1}{k}},q_{1-\frac{1}{k}}}}(t)|dt\right]$$

$$\leq 2(1-8c)\mathcal{W}(P, P|_{q_{\frac{1}{k}},q_{1-\frac{1}{k}}})$$

By a symmetric argument, we also have that with probability at least $1 - O(\beta)$,

$$\int_{t=q_{1-1/k}}^{b} |F_P(t) - F_{PDP}(t)|dt \leq 2(1-8c)\mathcal{W}(P, P|_{q_{\frac{1}{k}},q_{1-\frac{1}{k}}}).$$

Taking a union bound to ensure that all terms in Equation 14 are bounded as required, the proof is complete. $\qquad\square$

*Proof of Lemma E.17.* First, we condition on the event in Corollary I.3 (on the accuracy of differentially private quantile estimates) that for all $r \in [k]$,

$$\hat{q}_{\frac{2r-1}{2k}-\frac{1}{4k}} \leq \tilde{q}_r \leq \hat{q}_{\frac{2r-1}{2k}+\frac{1}{4k}},$$

note that this event happens with probability at least $1 - \beta$ over the randomness of the algorithm.

Observe that this implies that $F_{DP}$ increases by $\frac{1}{k}$ somewhere in the range $[\hat{q}_{\frac{2r-1}{2k} - \frac{1}{4k}}, \hat{q}_{\frac{2r-1}{2k} + \frac{1}{4k}}]$ (for all $r \in [k]$) and remains constant outside these intervals.

Now, we show that for all $t \in [a, b]$, we have that $|F_{PDP}(t) - F_{\hat{P}_n}(t)| \leq \frac{2}{k}$.

If there exists $t \in [a, \hat{q}_{\frac{1}{4k}})$, we have that $F_{P_{DP}}(t) = 0$, and $F_{\hat{P}_n}(t) \leq \frac{1}{4k}$, which implies that $|F_{DP}(t) - F_{\hat{P}_n}(t)| \leq \frac{1}{4k}$. If there exists no such $t$, then we have that $a = \hat{q}_{\frac{1}{4k}}$, and the corresponding interval collapses to a single point (which will fall in another interval considered below).

Next, fix any $r \in [k-1]$. Note that if there exists $t \in [\hat{q}_{\frac{2r-1}{2k} - \frac{1}{4k}}, \hat{q}_{\frac{2r+1}{2k} - \frac{1}{4k}})$, we have for all such $t$ that $\frac{r-1}{k} \leq F_{DP}(t) < \frac{r}{k}$, and $\frac{2r-1}{2k} - \frac{1}{4k} \leq F_{\hat{P}_n}(t) \leq \frac{2r+1}{2k} + \frac{1}{4k}$. This implies that for all such $t$, $|F_{DP}(t) - F_{\hat{P}_n}(t)| \leq \frac{2}{k}$. If there exists no such $t$, then we have that $\hat{q}_{\frac{2r-1}{2k} - \frac{1}{4k}} = \hat{q}_{\frac{2r+1}{2k} - \frac{1}{4k}}$, and this $r$ is not relevant since the corresponding interval collapses to a single point (that is considered in another interval).

Finally, for $t \in [\hat{q}_{\frac{2k-1}{2k}}, b]$, we have that $F_{P_{DP}}(t) \geq 1 - \frac{1}{k}$, and $F_{\hat{P}_n}(t) \geq 1 - \frac{1}{2k}$, so we have that $|F_{DP}(t) - F_{\hat{P}_n}(t)| \leq \frac{1}{k}$.

Note that every $t \in [a, b]$ is considered in some interval above and hence we have shown that for all $t \in [a, b]$, we have that $|F_{PDP}(t) - F_{\hat{P}_n}(t)| \leq \frac{2}{k}$.

Finally, using the formula for Wasserstein distance (and the definition of a restriction), we have that

$$\mathcal{W}(\hat{P}_n|_{q_{\frac{1}{k}}, q_{1 - \frac{1}{k}}}, P^{DP}|_{q_{\frac{1}{k}}, q_{1 - \frac{1}{k}}}) = \int_a^b \left| F_{\hat{P}_n|_{q_{\frac{1}{k}}, q_{1 - \frac{1}{k}}}}(t) - F_{P^{DP}|_{q_{\frac{1}{k}}, q_{1 - \frac{1}{k}}}}(t) \right| dt \qquad (16)$$

$$= \int_{q_{\frac{1}{k}}}^{q_{1 - \frac{1}{k}}} |F_P(t) - F_{PDP}(t)| \, dt \qquad (17)$$

$$\leq \int_{q_{\frac{1}{k}}}^{q_{1 - \frac{1}{k}}} \frac{2}{k} dt \qquad (18)$$

$$\leq \frac{2}{k} \left( q_{1 - 1/k} - q_{1/k} \right) \qquad (19)$$

$\square$

Before the proof of Claim E.18, we state the following variance-dependent version of the DKW inequality that uniformly bounds the absolute difference in CDFs between the true and empirical distribution.

**Theorem J.3** (See for example Theorem 1.2 in [BM23]). *Fix $n > 0$. There are absolute constants $c_0, c_1$ such that for all $\Delta \geq \frac{c_0 \log \log n}{n}$,*

$$\Pr \left[ \sup_{t: F_P(t)(1 - F_P(t)) \geq \Delta} \left| F_P(t) - F_{\hat{P}_n}(t) \right| \geq \sqrt{\Delta \cdot F(t)(1 - F(t))} \right] \leq 2e^{-c_1 \Delta n}$$

We also state the following lemma on Binomial random variables, which is a simple consequence of a Lemma by Bobkov and Ledoux [BL19].

**Lemma J.4** (Lemma 3.8 in [BL19]). *Let $S_n = \sum_{i=1}^n \eta_i$ be the sum of $n$ independent Bernoulli random variables with $\Pr[\eta_i = 1] = p$ and $\Pr[\eta_i = 0] = q = 1 - p$ (for all $i$). Also assume $p \in [\frac{1}{n}, 1 - \frac{1}{n}]$. Then, for some sufficiently small constant $c$,*

$$c\sqrt{npq} \leq \mathbb{E}[|S_n - np|] \leq \sqrt{npq}$$

*Proof of Claim E.18.* Now, by the formula for Wasserstein distance, the definition of restriction, and Fubini's theorem, we have that

$$\mathbb{E}[\mathcal{W}(P|_{q_{\frac{1}{k}}, q_{1 - \frac{1}{k}}}, \hat{P}_n|_{q_{\frac{1}{k}}, q_{1 - \frac{1}{k}}})] = \mathbb{E}\left[ \int_{q_{\frac{1}{k}}}^{q_{1 - \frac{1}{k}}} \left| F_P(t) - F_{\hat{P}_n}(t) \right| dt \right] = \int_{q_{\frac{1}{k}}}^{q_{1 - \frac{1}{k}}} \mathbb{E}\left[ \left| F_P(t) - F_{\hat{P}_n}(t) \right| \right] dt$$

By Lemma J.4, using the fact that $F_{\hat{P}_n}(t) = \sum_{i=1}^n 1[x_i \leq t]$, where each term in the sum is an independent Bernoulli random variable with expectation $F_P(t)$, with $q_{\frac{1}{k}} \leq t < q_{1-\frac{1}{k}}$ (ensuring that the conditions of the lemma are met), we get that $\mathbb{E}\left[\left|F_P(t) - F_{\hat{P}_n}(t)\right|\right] \geq c\sqrt{\frac{F_P(t)[1-F_P(t)]}{n}}$, which gives

$$\mathbb{E}[\mathcal{W}(P|_{q_{\frac{1}{k}},q_{1-\frac{1}{k}}}, \hat{P}_n|_{q_{\frac{1}{k}},q_{1-\frac{1}{k}}})] \geq c\int_{q_{\frac{1}{k}}}^{q_{1-\frac{1}{k}}} \sqrt{\frac{F_P(t)[1-F_P(t)]}{n}}dt$$

Now, consider the random variable $\mathcal{W}(P|_{q_{\frac{1}{k}},q_{1-\frac{1}{k}}}, \hat{P}_n|_{q_{\frac{1}{k}},q_{1-\frac{1}{k}}})$. Note that $\frac{1}{k} \geq \frac{c_3 \log \frac{n}{\beta}}{n}$ (for an appropriately chosen $c_3$), and so we are in the regime where we can apply Theorem J.3 for an appropriately chosen $\Delta$.

In particular, we have that for $t \in [q_{\frac{1}{k}}, q_{1-\frac{1}{k}})$, $F_P(t) \in [\frac{1}{k}, 1-\frac{1}{k})$.

Setting $\Delta = \frac{\log \frac{n}{\beta}}{c_1 n}$, we have that $\Delta \geq c_0 \frac{\log \log n}{n}$, and $\Delta \leq \frac{1}{2k}$ (the second inequality for sufficiently large $c_3$). In particular, this implies for $t \in [q_{\frac{1}{k}}, q_{1-\frac{1}{k}})$, $F_P(t) \in [2\Delta, 1-2\Delta)$, which implies that $F_P(t)(1-F_P(t)) \geq \Delta$, as long as $n > c_4 \log \frac{n}{\beta}$ for some sufficiently large constant $c_4$.

Now, using Theorem J.3, we have that with probability at least $1 - 2e^{-c_1 \frac{\log \frac{n}{\beta}}{c_1 n}n} \geq 1 - O(\beta)$,

$$\sup_{t \in [q_{\frac{1}{k}},q_{1-\frac{1}{k}})} \left|F_P(t) - F_{\hat{P}_n}(t)\right| \leq \sqrt{\frac{\log \frac{n}{\beta}}{c_1 n} F_P(t)(1-F_P(t))}$$

Condition on this for the rest of the proof. Then, we can write the following set of equations.

$$\begin{aligned}
\mathcal{W}(P|_{q_{\frac{1}{k}},q_{1-\frac{1}{k}}}, \hat{P}_n|_{q_{\frac{1}{k}},q_{1-\frac{1}{k}}}) &= \int_{q_{\frac{1}{k}}}^{q_{1-\frac{1}{k}}} |F_P(t) - F_{\hat{P}_n}(t)|dt \\
&\leq \int_{q_{\frac{1}{k}}}^{q_{1-\frac{1}{k}}} \sqrt{\frac{\log \frac{n}{\beta}}{c_1 n} F_P(t)(1-F_P(t))}dt \\
&\leq \sqrt{c_5 \log \frac{n}{\beta}} \int_{q_{\frac{1}{k}}}^{q_{1-\frac{1}{k}}} \sqrt{\frac{F_P(t)(1-F_P(t))}{n}}dt \\
&\leq \sqrt{c_6 \log \frac{n}{\beta}} \mathbb{E}[\mathcal{W}(P|_{q_{\frac{1}{k}},q_{1-\frac{1}{k}}}, \hat{P}_n|_{q_{\frac{1}{k}},q_{1-\frac{1}{k}}})]
\end{aligned}$$

as required.

$\square$

## J.4 Local Minimality in the One-Dimensional Setting

In this subsection, we argue that the instance-optimal algorithm discussed in Section E.2 is also locally-minimal (See Appendix B.2 for a discussion of local minimality).

First, we state a corollary of our upper bound for continuous distributions, Theorem E.14. This corollary follows by discretizing the distribution and applying the previous upper bound to the discretized distribution. The parameters of the discretized distribution are related to that of the original distribution via simple coupling arguments.

**Corollary J.5.** *Fix $\varepsilon, \beta \in (0,1]$, $a, b \in \mathbb{R}$, $n \in \mathbb{N}$. Let $P$ be any continuous distribution supported on $[a,b]$. Consider any $\gamma < b - a \in \mathbb{R}$ (such that $\gamma$ divides $b - a$), and let $n > c_2 \frac{\log^4 \frac{b-a}{\gamma \beta \varepsilon}}{\varepsilon}$ for some sufficiently large constant $c_2$. Then, there exists an algorithm, that when given inputs $\mathbf{x} \sim P^n$,*

*privacy parameter $\varepsilon$, interval end points $a, b$, granularity $\gamma$, and access to algorithm $A_{quant}$, outputs a distribution $P^{DP}$ such that with probability at least $1 - O(\beta)$ over the randomness of $\mathbf{x}$ and the algorithm,*

$$\mathcal{W}(P, P^{DP}) = O\left( \sqrt{\log n}\, \mathbb{E}\left[ \mathcal{W}(P|_{q_{\frac{1}{k}}, q_{1 - \frac{1}{k}}}, \hat{P}_n|_{q_{\frac{1}{k}}, q_{1 - \frac{1}{k}}} \right] + \mathcal{W}(P, P|_{q_{\frac{1}{k}}, q_{1 - \frac{1}{k}}}) + \frac{1}{k}\left( q_{1 - 1/k} - q_{1/k} \right) \right) + \gamma$$

*where $\hat{P}_n$ is the uniform distribution on $\mathbf{x}$, $q_\alpha$ represents the $\alpha$-quantile of distribution $P$, and $k = \lceil \frac{\varepsilon n}{4c_3 \log^3 \frac{b-a}{\beta\gamma} \log \frac{n}{\beta}} \rceil$, where $c_3$ is a sufficiently large constant.*

We state a lemma of Ledoux and Bobkov that we will use in the main proof of this section.

**Lemma J.6** (Lemma 3.8 in [BL19]). *Let $S_n = \sum_{i=1}^n \eta_i$ be the sum of $n$ independent Bernoulli random variables with $\Pr[\eta_i = 1] = p$ and $\Pr[\eta_i = 0] = q = 1 - p$ (for all $i$). Then, for some sufficiently small constant $c$,*

$$c \min\{2npq, \sqrt{npq}\} \leq \mathbb{E}[|S_n - np|] \leq \min\{2npq, \sqrt{npq}\}$$

Now, we are ready to state and prove the local minimality result. Note that the statement will reference the rates defined by Equation 1 in the introduction.

**Theorem J.7.** *Let $a, b \in \mathbb{R}$, $\gamma \in \mathbb{R}$. For any continuous distribution $P$ over $[a, b]$ with a density, let $N(P) = \{Q : D_\infty(P, Q) \leq \log 2\}$. Fix $\beta, \gamma, \varepsilon \in (0, 1]$, and let $n = \Omega\left( \frac{\log^4 \frac{b-a}{\gamma\varepsilon}}{\varepsilon} \right)$, with $n' = \frac{n}{c_7 \log n \log^3 \frac{b-a}{\gamma\varepsilon}}$ for some constant $c_7$. There exists an algorithm $\mathcal{A}$ such that for all continuous distributions $P$, for all algorithms $\mathcal{A}'$, there exists a distribution $Q \in N(P)$ such that*

$$R_{\mathcal{A}, n}(Q) \leq O(\text{polylog } n) \cdot \max\{R_{\mathcal{A}', \lceil n' \rceil}(Q), R_{\mathcal{A}', \lfloor n'/4 \rfloor}(Q)\} + \gamma,$$

*Proof.* Let $k = \lceil \frac{\varepsilon n}{4c_3 \log^3 \frac{b-a}{\beta\gamma} \log \frac{n}{\beta}} \rceil$, and set $n' = \frac{2n}{c_4 \log^3 \frac{b-a}{\beta\gamma} \log \frac{n}{\beta}}$ for a sufficiently large constant $c_4$. Then, by Corollary J.5 with appropriately chosen $\beta$ we have that with probability at least 0.95, for any distribution $Q$ (and hence particularly any distribution $Q \in N(P)$),

$$\mathcal{W}(Q, \mathcal{A}(\hat{Q}_n)) = O\left( \frac{1}{\varepsilon n'} \left( q_{1 - \frac{2}{C\varepsilon n'}}(Q) - q_{\frac{2}{C\varepsilon n'}}(Q) \right) + \mathcal{W}(Q, Q|_{q_{\frac{2}{C\varepsilon n'}}(Q), q_{1 - \frac{2}{C\varepsilon n'}}(Q)}) \right.$$

$$\left. + \sqrt{\log n}\, \mathbb{E}\left[ \mathcal{W}\left( Q|_{q_{\frac{2}{C\varepsilon n'}}(Q), q_{1 - \frac{2}{C\varepsilon n'}}(Q)}, \hat{Q}_n|_{q_{\frac{2}{C\varepsilon n'}}(Q), q_{1 - \frac{2}{C\varepsilon n'}}(Q)} \right) \right] \right) + \gamma,$$

where $C$ is the constant referenced in Theorem E.3. We will show that for distribution $P$, each of the corresponding distribution-dependent terms is closely related to the terms for $Q$.

First, consider $\frac{1}{\varepsilon n'} \left( q_{1 - \frac{2}{C\varepsilon n'}}(Q) - q_{\frac{2}{C\varepsilon n'}}(Q) \right)$. Firstly, note that for all $\alpha \in (0, 1)$, $q_\alpha(P) \geq q_{\alpha/2}(Q)$, and $q_\alpha(P) \leq q_{2\alpha}(Q)$, since $D_\infty(P, Q) \leq \ln 2$, which implies that $\frac{1}{2} F_Q(t) \leq F_P(t) \leq 2 F_Q(t)$ for all $t \in \mathbb{R}$. Similarly, note that for all $\alpha \in (0, 1)$, $q_{1-\alpha}(P) \geq q_{1-2\alpha}(Q)$, and $q_{1-\alpha}(P) \leq q_{1 - \frac{1}{2} \cdot \alpha}(Q)$. Hence, we have that

$$\frac{1}{\varepsilon n'} \left( q_{1 - \frac{2}{C\varepsilon n'}}(Q) - q_{\frac{2}{C\varepsilon n'}}(Q) \right) \leq \frac{1}{\varepsilon n'} \left( q_{1 - \frac{1}{C\varepsilon n'}}(P) - q_{\frac{1}{C\varepsilon n'}}(P) \right)$$

Next, consider $\mathcal{W}(P, P|_{q_{\frac{1}{C\varepsilon n}}(P), q_{1-\frac{1}{C\varepsilon n}}(P)})$. Recall that $q_{\frac{1}{C\varepsilon n}}(P) \le q_{\frac{2}{C\varepsilon n}}(Q)$, and $q_{1-\frac{1}{C\varepsilon n}}(P) \ge q_{1-\frac{2}{C\varepsilon n}}(Q)$. Then, (noting that $L(P) = L(Q)$ and $q_1(P) = q_1(Q)$), we have that

$$\mathcal{W}(Q, Q|_{q_{\frac{2}{C\varepsilon n'}}(Q), q_{1-\frac{2}{C\varepsilon n'}}(Q)}) = \int_{L(Q)}^{q_{\frac{2}{C\varepsilon n'}}(Q)} F_Q(t)dt + \int_{q_{1-\frac{2}{C\varepsilon n'}}(Q)}^{q_1(Q)} |1 - F_Q(t)|dt$$

$$\le 2\int_{L(Q)}^{q_{\frac{2}{C\varepsilon n'}}(Q)} F_P(t)dt + 2\int_{q_{1-\frac{1}{C\varepsilon n'}}(Q)}^{q_1(Q)} |1 - F_P(t)|dt$$

$$\le 2\int_{L(P)}^{q_{\frac{4}{C\varepsilon n'}}(P)} F_P(t)dt + 2\int_{q_{1-\frac{1}{4C\varepsilon n'}}(P)}^{q_1(P)} |1 - F_P(t)|dt$$

$$= 2\mathcal{W}(P, P|_{q_{\frac{4}{C\varepsilon n'}}(P), q_{1-\frac{4}{C\varepsilon n}}(P)})$$

Finally, consider $\frac{1}{\sqrt{\log n}}\mathbb{E}\left[\mathcal{W}(P|_{q_{\frac{1}{C\varepsilon n}}(P), q_{1-\frac{1}{C\varepsilon n}}(P)}, \hat{P}_n|_{q_{\frac{1}{C\varepsilon n}}(P), q_{1-\frac{1}{C\varepsilon n}}(P)})\right]$. By Fubini's theorem and applying both inequalities in Lemma J.6, we have that

$$\mathbb{E}\left[\mathcal{W}(Q|_{q_{\frac{2}{C\varepsilon n'}}(Q), q_{1-\frac{2}{C\varepsilon n'}}(Q)}, \hat{Q}_n|_{q_{\frac{2}{C\varepsilon n'}}(Q), q_{1-\frac{2}{C\varepsilon n'}}(Q)})\right]$$

$$= \int_{q_{\frac{2}{C\varepsilon n'}}(Q)}^{q_{1-\frac{2}{C\varepsilon n'}}(Q)} \mathbb{E}[|F_Q(t) - F_{\hat{Q}_n}(t)|]dt$$

$$\le \int_{q_{\frac{1}{C\varepsilon n'}}(P)}^{q_{1-\frac{1}{C\varepsilon n'}}(P)} \mathbb{E}[|F_Q(t) - F_{\hat{Q}_n}(t)|]dt$$

$$\le \int_{q_{\frac{1}{C\varepsilon n'}}(P)}^{q_{1-\frac{1}{C\varepsilon n'}}(P)} \min\left\{2F_Q(t)[1 - F_Q(t)], \sqrt{\frac{F_Q(t)[1 - F_Q(t)]}{n}}\right\} dt$$

$$\le \int_{q_{\frac{1}{C\varepsilon n'}}(P)}^{q_{1-\frac{1}{C\varepsilon n'}}(P)} \min\left\{8F_P(t)[1 - F_P(t)], 2\sqrt{\frac{F_P(t)[1 - F_P(t)]}{n}}\right\} dt$$

$$\le \int_{q_{\frac{1}{C\varepsilon n'}}(P)}^{q_{1-\frac{1}{C\varepsilon n'}}(P)} \min\left\{8F_P(t)[1 - F_P(t)], 2\sqrt{\frac{F_P(t)[1 - F_P(t)]}{\lceil n'\rceil}}\right\} dt$$

$$\le c_5 \int_{q_{\frac{1}{C\varepsilon n'}}(P)}^{q_{1-\frac{1}{C\varepsilon n'}}(P)} \mathbb{E}[|F_P(t) - F_{\hat{P}_{\lceil n'\rceil}}(t)|]dt$$

$$= \mathbb{E}\left[\mathcal{W}(P|_{q_{\frac{1}{C\varepsilon n'}}(P), q_{1-\frac{1}{C\varepsilon n'}}(P)}, \hat{P}_{\lceil n'\rceil}|_{q_{\frac{1}{C\varepsilon n'}}(P), q_{1-\frac{1}{C\varepsilon n'}}(P)})\right],$$

where $c_5$ is a sufficiently large constant and the fourth inequality holds since $\lceil n'\rceil \le n$.

By the above observations connecting the distribution-dependent terms with the corresponding terms for $P$, we have that for all $Q$, with probability at least 0.95,

$$\mathcal{W}(Q, \mathcal{A}(\hat{Q}_n)) = O\left(\frac{1}{\varepsilon n'}\left(q_{1-\frac{1}{C\varepsilon n'}}(P) - q_{\frac{1}{C\varepsilon n'}}(P)\right) + \mathcal{W}(P, P|_{q_{\frac{4}{C\varepsilon n'}}(P), q_{1-\frac{4}{C\varepsilon n}}(P)})\right.$$

$$\left. + \sqrt{\log n}\mathbb{E}\left[\mathcal{W}\left(P|_{q_{\frac{1}{C\varepsilon n'}}(P), q_{1-\frac{1}{C\varepsilon n'}}(P)}, \hat{P}_{\lceil n'\rceil}|_{q_{\frac{1}{C\varepsilon n'}}(P), q_{1-\frac{1}{C\varepsilon n'}}(P)}\right)\right]\right) + \gamma$$

$$= O(\log n)\left(\frac{1}{\varepsilon\lceil n'\rceil}\left(q_{1-\frac{1}{C\varepsilon\lceil n'\rceil}}(P) - q_{\frac{1}{C\varepsilon\lceil n'\rceil}}(P)\right) + \mathcal{W}(P, P|_{q_{\frac{1}{C\varepsilon\lfloor n'/4\rfloor}}(P), q_{1-\frac{1}{C\varepsilon\lfloor n'/4\rfloor}}(P)})\right.$$

$$\tag{20}$$

$$\left. + \frac{1}{\sqrt{\log n}}\mathbb{E}\left[\mathcal{W}\left(P|_{q_{\frac{1}{C\varepsilon\lceil n'\rceil}}(P), q_{1-\frac{1}{C\varepsilon\lceil n'\rceil}}(P)}, \hat{P}_{\lceil n'\rceil}|_{q_{\frac{1}{C\varepsilon\lceil n'\rceil}}(P), q_{1-\frac{1}{C\varepsilon\lceil n'\rceil}}(P)}\right)\right]\right) + \gamma$$

Now, we proceed with the analysis in two cases. Firstly, consider the case when the first and third terms inside the bracket on the RHS of equation 20 are larger than the second term inside the bracket. Then, we have that for all $Q$, with probability at least 0.95,

$$\mathcal{W}(Q, \mathcal{A}(\hat{Q}_n) = O(\log n) \left( \frac{1}{\varepsilon \lceil n' \rceil} \left( q_{1 - \frac{1}{C\varepsilon \lceil n' \rceil}}(P) - q_{\frac{1}{C\varepsilon \lceil n' \rceil}}(P) \right) \right.$$

$$+ \left. \frac{1}{\sqrt{\log n}} \mathbb{E}\left[ \mathcal{W}\left( P|_{q_{\frac{1}{C\varepsilon \lceil n' \rceil}}(P), q_{1 - \frac{1}{C\varepsilon \lceil n' \rceil}}(P)}, \hat{P}_{\lceil n' \rceil}|_{q_{\frac{1}{C\varepsilon \lceil n' \rceil}}(P), q_{1 - \frac{1}{C\varepsilon \lceil n' \rceil}}(P)} \right) \right] \right) + \gamma$$

By Theorem E.3 and the fact that $n' < n$, for all algorithms $\mathcal{A}'$, there exists a distribution $Q \in N(P)$ such that ,

$$R_Q(\mathcal{A}', \lceil n' \rceil) = \Omega\left( \frac{1}{\varepsilon \lceil n' \rceil} \left( q_{1 - \frac{1}{C\varepsilon \lceil n' \rceil}}(P) - q_{\frac{1}{C\varepsilon \lceil n' \rceil}}(P) \right) \right.$$

$$+ \left. \frac{1}{\sqrt{\log n}} \mathbb{E}\left[ \mathcal{W}\left( P|_{q_{\frac{1}{C\varepsilon \lceil n' \rceil}}(P), q_{1 - \frac{1}{C\varepsilon \lceil n' \rceil}}(P)}, \hat{P}_{\lceil n' \rceil}|_{q_{\frac{1}{C\varepsilon \lceil n' \rceil}}(P), q_{1 - \frac{1}{C\varepsilon \lceil n' \rceil}}(P)} \right) \right] \right).$$

Hence, for all algorithms $A'$ and the corresponding distribution $Q$, with probability at least 0.95,

$$\mathcal{W}(Q, \mathcal{A}(\hat{Q}_n) \leq O(\log n) R_Q(A', \lceil n' \rceil) + \gamma.$$

Next, consider the case where the first and third terms inside the bracket on the RHS of equation 20 are smaller than the second term inside the bracket. Then, we have that for all $Q$, with probability at least 0.95,

$$\mathcal{W}(Q, \mathcal{A}(\hat{Q}_n) = O(\log n) \mathcal{W}(P, P|_{q_{\frac{1}{C\varepsilon \lfloor n'/4 \rfloor}}(P), q_{1 - \frac{1}{C\varepsilon \lfloor n'/4 \rfloor}}(P)}) + \gamma.$$

By Theorem E.3, for all algorithms $\mathcal{A}'$, there exists a distribution $Q \in N(P)$ such that

$$R_Q(\mathcal{A}', \lfloor n'/4 \rfloor) = \Omega\left( \mathcal{W}(P, P|_{q_{\frac{1}{C\varepsilon \lfloor n'/4 \rfloor}}(P), q_{1 - \frac{1}{C\varepsilon \lfloor n'/4 \rfloor}}(P)}) \right).$$

Hence, we have that for all algorithms $A'$ and for the corresponding distribution $Q$, with probability at least 0.95,

$$\mathcal{W}(Q, \mathcal{A}(\hat{Q}_n) = O(\log n) R_Q(\mathcal{A}', \lfloor n'/4 \rfloor) + \gamma,$$

as required. This completes the proof. $\qquad \square$

