# OpenReview forum: "Instance-Optimal Private Density Estimation in the Wasserstein Distance"
_NeurIPS.cc/2024/Conference — NeurIPS 2024 poster_

### Official Review · Reviewer_e9RQ · 2024-07-11

**Soundness:** 3
**Presentation:** 4
**Contribution:** 3
**Rating:** 7
**Confidence:** 3

**Summary:**

The paper addresses the problem of differentially private density estimation using the Wasserstein distance. It approaches this problem in a non-parametric manner, meaning it does not assume any specific distribution. Instead of focusing on worst-case error guarantees, the authors aim to design algorithms that achieve instance-optimal error. In other words, the algorithms provide the best possible error within a small "neighborhood" around any given input.

The authors explore various definitions of "neighborhood" in the context of instance optimality for distribution estimation in both $\mathbb{R}$ and $\mathbb{R}^2$. They present algorithms that achieve performance close to the instance-optimal ones, up to multiplicative polylogarithmic factors. Furthermore, the algorithm for $\mathbb{R}^2$ is extended to arbitrary metric spaces, though this generalization results in an extra polylogarithmic factor in the performance guarantee.

**Strengths:**

1. Instance optimal algorithms can achieve significantly better performance on specific inputs. Designing such algorithms and proving their instance optimality is nontrivial.

2. The paper introduces a notion of instance optimality for the problem. An instance optimal algorithm under this notion is quite strong, as its performance is competitive with an algorithm that knows a constant multiplicative approximation of the distribution density (such approximation has a small absolute error on points with low density values).

3. The writing is excellent. The problem is well-motivated, and the paper is carefully structured and easy to follow.

**Weaknesses:**

The multiplicative factors in the instance optimality guarantee can be large.

**Questions:**

In Theorem 2.3, the established error includes a multiplicative factor of $\log |X|$, where $|X|$ represents the size of the metric space. I would like to confirm that if $X$ is a finite $m$-dimensional space, this implies that the error scales with $m$, which can be relatively large.

I did not read the appendices.

**Limitations:**

Same as ''Questions"

---

> ### Author Rebuttal · Authors · 2024-08-07
>
> Thank you to the reviewer for their review.
>
> **“In Theorem 2.3, the established error includes a multiplicative factor of log|X|, where |X| represents the size of the metric space. I would like to confirm that if X  is a finite m -dimensional space, this implies that the error scales with m, which can be relatively large.”**
>
> This is correct, in dimension $m$, the gap between our upper and lower bounds scales with a multiplicative factor of $m$ making the results less useful in high dimensions. We note however that
>
> (a) There is a large class of applications where the dimension is naturally small, for example, building a population heatmap.
>
>  (b) While we have a linear dependence on the dimension, we note that our bounds are significantly better than worst case bounds for Wasserstein learning which have rates that are $n^{O(-1/d)}$ and are thus vacuous already for $d \approx \log n$.
>
> (c) In settings where data lies on a lower-dimensional manifold that we have some access to, we anticipate that this dependence can be improved to the dimensionality of the manifold, rather than the ambient dimension (by building a better HST embedding). This may happen, for e.g., when we are learning a distribution of embeddings of images, and we can sample from the manifold of all possible images (using a generative model) to build the HST embedding. We leave a formal exploration of this to future work.

---

### Official Review · Reviewer_FtyV · 2024-07-13

**Soundness:** 3
**Presentation:** 2
**Contribution:** 2
**Rating:** 3
**Confidence:** 3

**Summary:**

This paper looks at the problem of private instance optimal density estimation in wasserstein distance. The starting point of their research is the recent result that shows that the minimax rate of  eps differentially private density estimation in wasserstein distance scales as (eps*n)^{-1/d}. Therefore, the authors take an instance optimal approach, where given samples from an unknown distribution P the algorithm wishes to achieve a cost of at most alpha times the maximum cost of any other algorithm’s cost in a neighborhood around P, that knows neighborhood. For defining the neighborhood, they consider the class of distributions that are close to P in D infinity distance. Coming up with this notion of instance optimality is claimed to be the main contribution of the paper. They authors provide tight upper and lower bounds for such an instance optimal private density estimation guarantee. The upper bounds mostly use the existing techniques from prior work.

The first result shows that when P is one dimensional and its sample space is [0,1], they can achieve an polygon-factor approximate cost as compared to an algorithm that knows that the unknown distribution is either P or Q, for some distribution Q that exists. Question to the authors, do they mean for every Q or there exists such a Q? If the later is the case, the theorem sounds weak to me.

Similarly, for two dimensional distributions over [0,1]^2, they could get a (log n)^{O(1)} approximation (how is it different from polylog n?) as compared to an algorithm that knows the distribution is ln 2 close to P in D infinity distance.

**Strengths:**

Overall, while I believe the notion of instance optimality introduced has some merit.

**Weaknesses:**

I did not find any theorems that justifies that their polylog n factors are tight or not. I would like to believe the approximation factors are quite pessimistic. Moreover, the results do not apply beyond two dimensions. The choice n’=n/polylog(n) also seems ad-hoc.

I found the results derived in this paper are quite weak for a venue like NeurIPS.

**Questions:**

None.

**Limitations:**

None.

---

> ### Author Rebuttal · Authors · 2024-08-07
>
> Thank you to the reviewer for their review.
>
> **“they can achieve an polylog-factor approximate cost as compared to an algorithm that knows that the unknown distribution is either P or Q, for some distribution Q that exists. Question to the authors, do they mean for every Q or there exists such a Q? If the latter is the case, the theorem sounds weak to me.”**
>
> Please allow us to clarify the definition of instance optimality we use in the one-dimensional setting. As explained in Section B.1, our notion of instance optimality in the one-dimensional setting builds on one suggested by Donoho and Liu in terms of the ‘hardest one-dimensional subproblem’- where the distribution Q to compare to is chosen to be close to P, making the bounds instance specific. Below, we provide additional intuition on this definition and the choice of Q. If Q is chosen so that P and Q are distinguishable (there exists a hypothesis test that can with high probability distinguish samples from P from samples from Q) then the algorithm that is told that the unknown distribution is either P or Q has 0 error (whp) on both P or Q. Thus, we can not hope to be competitive with such an algorithm. If P and Q are indistinguishable  (there does not exist a hypothesis test that can with high probability distinguish samples from P from samples from Q) then there can not exist an estimation algorithm whose error on both P and Q is less than half the distance from P to Q since such an estimation algorithm would induce a hypothesis test. Thus, the only distributions Q such that we can hope to be competitive (on both P and Q) with the estimation rate of an algorithm that is told the unknown distribution is P or Q are distributions Q that lie on the boundary of indistinguishability for P and Q. In fact, the only Q for which we have such a hope are those which are furthest from P on this boundary. Thus, achieving this for any Q is indeed a strong notion of optimality. In this work, we actually study a stronger notion of instance optimality that essentially amounts to not only comparing to an algorithm that is told that the unknown distribution is P or Q, but is also told the support of P.
>
> **“I did not find any theorems that justifies that their polylog n factors are tight or not.”**
>
>  Our upper and lower bounds are tight up to polylog n factors. It is possible that these factors can be tightened, although we suspect this would require new techniques. In particular, in the 2-dimensional setting, a log factor arises from the metric distortion in the HST where the distortion factor is tight. Even with the polylog n factors, our instance specific upper bound is significantly lower that the worst case bound studied in prior work.
>
> **“Moreover, the results do not apply beyond two dimensions.”**
>
>  Our results apply for any metric space by building a HST approximation to that metric space and leveraging our result for general HSTs. This induces a factor of $\log|X|$ gap in the upper and lower bounds resulting from the distortion factor in the HST approximation. We focus on the 2-dimensional case since this factor is small in this case and this setting captures many interesting realistic scenarios (e.g. population densities). Our algorithm works equally well in other low-dimensional metric spaces, as we describe in the introduction. In higher dimensions the gap has a linear dependence on the dimension. While we agree the result is less useful in high dimensions, we note that it gives better bounds than the worst case results  for Wasserstein learning that have rates that are $n^{O(-1/d)}$ and are thus vacuous already for $d \approx \log n$. We also anticipate that in many practical settings, some knowledge of the data distribution (through some public data for example) would allow for the building of better tree embeddings, and then using our algorithm would give better bounds in high dimensions. We leave such an exploration to future work.
>
> **“The choice n’=n/polylog(n) also seems ad-hoc.”**
>
>  Precise formulations of $n’$ are given in Theorem J.7 for the one-dimensional setting and Theorem H.1 for the HST setting. In both settings, we give precise formulations of the upper and lower bounds in terms of n. In order to compare these bounds, we need to compare the upper bound to a lower bound with slightly fewer samples.

---

> > ### Comment · Reviewer_FtyV · 2024-08-14
> > **Read the rebuttal**
> >
> > I have read the rebuttal by the authors. I am afraid that the answers do not add much more clarity or new information to my previous assessment regarding the paper. Therefore, I'll maintain my score.

---

### Official Review · Reviewer_8M6K · 2024-07-25

**Soundness:** 4
**Presentation:** 4
**Contribution:** 4
**Rating:** 8
**Confidence:** 4

**Summary:**

The paper focuses on estimating densities while preserving privacy (differential privacy). Preserving privacy comes with trade-offs in terms of accuracy of their estimates, which is measured using the Wasserstein distance in this paper. One of the main conceptual contribution is introducing a notion of instance optimality for this problem. The authors motivate the choice by comparing the notion against some alternatives from related literature.

Their new definition of instance optimality is similar to the hypothesis testing problem where one has to look at a sequence of samples and decide whether they come from distribution X or Y. Here, instance optimality is defined as the performance of an algorithm that a priori knows that the samples are either from X or Y, and once it successfully determines which, it can return the density. For some real algorithm to be instance optimal, it has to match the performance of this algorithm without the prior knowledge, unto a constant slowdown.  There are some further nuances to the definition that make the definition robust to trivial edge cases.

The paper then considers (DP) estimation with respect to Wasserstein error, which is a well motivated problem in ML. The paper shows improved algorithms for private estimation, that beat the previous baselines wrt to the special case of Discrete distributions and TV metric.

One other technical insight is in a tighter analysis of existing HST algorithms which are a surprisingly good fit for the problem when considered over R^2.

**Strengths:**

--

**Weaknesses:**

Minor Typos —
(250) follow

Perhaps not the main focus of the paper, but it would be nice to have all the constants. It encourages implementation, and also constants seem to make or break practical DP algorithms. And further since algorithms are inspired from practical HST algorithms, aren't a some of the source constants already available?

**Questions:**

--

**Limitations:**

--

---

> ### Author Rebuttal · Authors · 2024-08-07
>
> Thank you to the reviewer for their review.
>
> **“Perhaps not the main focus of the paper, but it would be nice to have all the constants.”**
>
> We agree that constant factors are very important for practical implementations. From the implementation point of view, the constant factors relating algorithm parameters to privacy parameters are important to ensure the correct privacy guarantee is achieved. We have made sure that all constants related to the privacy guarantees are explicit. The constants in the utility bound are left implicit in our work as they are not needed to correctly implement the private algorithms. Our theoretical analysis can likely be tightened and thus we don't expect the constant factors will be useful for practitioners. We further note that the quantile algorithms used in the one-dimensional setting have been used successfully in practice, so we expect this algorithm to be practical. Similarly, as the reviewer notes, there exist practical HST algorithms, so we also expect the algorithms in the 2-dimensional setting to be practical.

---

### Official Review · Reviewer_WXt4 · 2024-08-04

**Soundness:** 3
**Presentation:** 2
**Contribution:** 2
**Rating:** 5
**Confidence:** 3

**Summary:**

The paper studies distribution estimation under Wasserstein distance while requiring the estimator to be differentially private and "instance-optimal," i.e., being competitive against the algorithm optimal on a distribution neighborhood. The derived class of estimators can achieve comparable performance up to polylogarithmic factors and is extendable to other HST (hierarchically separated tree) metric spaces. Besides the theoretical advancement, the primary conceptual contribution is the new formulation of instance optimality in distribution estimation.

**Strengths:**

- The authors put significant effort into motivating the problem and their "instant-optimal" formulation. Relevant sections are well-written and present detailed comparisons to some of the prior results and works.
- The estimators are constructed concretely rather than shown to exist. Their guarantees extend from 1-D and 2-D Rea to HST metric spaces while maintaining privacy.
- Construction of distribution nets and using Assoud's Lemma may be of independent interest in establishing the lower bounds in other settings where (local) distribution geometry matters.

**Weaknesses:**

- It seems that the reference (optimal) estimator evaluates its performance on both the actual underlying distribution and the worst possible distribution, which differs from the actual by a constant factor of 2 in the distribution density ratio. This formulation significantly weakens the power of the reference distribution since, intuitively, it has to produce something in between that could largely deviate from the actual underlying distribution.
- The poly-log multiplicative factors in the upper bounds do not seem optimal. It would be better if the authors could provide some lower bounds to justify this dependence.
- The formulation is much weaker than some prior works in the discrete distribution setting. For example, OS15 shows that their estimator is comparable to all "natural estimators" on each distribution instance, where the best natural estimator essentially knows the actual distribution (without permutation) and only has to output the same probability estimator for symbols that appear the same number of times (a natural requirement).

**Questions:**

Please see the comments in the "weaknesses" section. I appreciate the authors' time and efforts.

**Limitations:**

The authors discussed and compared their results, which seems sufficient given the page limit. Negative societal impact is not applicable here.

---

> ### Author Rebuttal · Authors · 2024-08-07
>
> We thank the reviewer for their comments and address their questions below.
>
> **“It seems the reference…distribution.”:**
>
> a) In Wasserstein estimation (unlike TV estimation, for example), it is important to accurately estimate the support of the distribution- putting even a very small amount of mass far away from the support could result in large error. Our notion of neighborhood ensures that the distributions we consider have to have the same support as the original distribution. The infinity divergence is stronger than this, and approximately captures the concentration properties of the distribution- regions with large density will continue to have significant density, and regions with very small density will continue to have small density. Hence, comparing to algorithms with knowledge of such a neighborhood is indeed a strong guarantee for Wasserstein distance- since it captures whether the algorithm can adapt to the sparsity and concentration of the distribution without knowledge of it. In particular, we believe that our notion of neighborhood is natural when considering points with a metric structure.
>
> b) We note that the constant $2$ that we choose (to bound the infinity divergence in our neighborhood definition) can be replaced by any constant (strictly) between $1$ and $2$ with appropriate adjustments of the constants in the lower bound- we chose $2$ since we believe it captures the essence of the arguments and makes the presentation cleaner.
>
> c) Our fundamental goal in defining our notion of instance optimality is to ensure that the target estimation rate (defined by the lower bound) adapts as much as possible to easy instances of the problem. In our instance optimality notion, this is achieved if all the distributions within the neighborhood of a distribution P are (up to constants) as hard to estimate as P. Our specific definition of neighborhood is based on the intuition that the concentration and sparsity of a distribution are what controls its ease of estimation in the Wasserstein distance. This is backed up by the fact that our rates are controlled by these parameters. This is an indication that indeed all distributions within the neighborhood of P are approximately as hard to estimate as P.
>
> d) In the one dimensional setting, we note that the notion of instance optimality is even stronger- we not only give the algorithm a multiplicative approximation of the density, but also tell it that the distribution is one of two distributions $P$ or $Q$ (where $Q$s density is within a multiplicative factor of $P$).
>
> **“The formulation is much weaker….(a natural requirement).”:**
>
> We believe that a good definition of instance optimality depends on the context and for the problem of discrete distribution estimation, indeed the work of OS15 presents two compelling definitions. However, the definition based on permutations is inappropriate when the domain has a metric structure on it, e.g. the unit line. Here knowing the distribution up to a permutation can allow for a very large Wasserstein radius, and an algorithm that knows the support can be significantly better. Indeed in Fig. 1, we give an example of such a distribution, where our benchmark is a lot smaller than the permutation neighborhood benchmark. Additionally, for the definition based on natural algorithms, when the distribution is continuous, “counts” as used in the definition of natural algorithms are meaningless as all counts will be 0 or 1, and hence this corresponds to competing with a mixture of the empirical distribution and uniform distribution- a very strong restriction on the algorithms we compare to. Additionally, such a notion of instance optimality is unachievable under the constraint of differential privacy (even for discrete distributions).  Thus the notion of instance optimality must be chosen carefully, based on the kind of domain knowledge we imagine an expert designing a custom algorithm for an application may have. In the case of Wasserstein estimation, we believe that a strong but realistic benchmark is an algorithm that knows roughly where the mass is concentrated, i.e. a multiplicative approximation to the pdf of the distribution. Our algorithm competes with the best such algorithm, on every distribution, even without knowing anything about the distribution a priori.
>
> **“The poly-log multiplicative factors in the upper bounds do not seem optimal. It would be better if the authors could provide some lower bounds to justify this dependence.”**
>
> It is possible that these logarithmic factors could be tightened, but we suspect this would require new techniques. In particular, in the 2-dimensional setting, a log factor arises from the metric distortion in the HST where the distortion factor is tight. Even with the polylog factors, our instance-specific differentially private upper bound is significantly lower than the worst case bounds studied in prior work (which become vacuous when the dimension is $\log n$ or larger).

---

### Decision · Program_Chairs · 2024-09-25

**Decision:**

Accept (poster)

**Comment:**

This paper considers the problem of learning potentially arbitrary distributions with respect to the Wasserstein distance under differential privacy constraints. The authors define a natural notion of instance-optimality in this setting and develop a nearly-instance-optimal learning algorithm (within poly-logarithmic factors). While there was a divergence of scores, after reading the reviews and subsequent discussion, my understanding is that this work is technically and conceptually novel, and clearly meets the acceptance bar at NeurIPS.